# Ferric reduction by a CYBDOM protein counteracts increased iron availability in root meristems induced by phosphorus deficiency

Rodolfo A. Maniero [1], Cristiana Picco [2], Anja Hartmann[1], Felipe Engelberger [3], Antonella Gradogna [2], Joachim Scholz-Starke [2], Michael Melzer[1], Georg Künze[3,4,5], Armando Carpaneto [2,6], Nicolaus von Wirén [1] & Ricardo F. H. Giehl [1] ✉

To mobilize sparingly available phosphorus (P) in the rhizosphere, many plant species secrete malate to release P sorbed onto (hydr)oxides of aluminum and iron (Fe). In the presence of Fe, malate can provoke Fe over-accumulation in the root apoplast, triggering a series of events that inhibit root growth. Here, we identified HYPERSENSITIVE TO LOW P1 (HYP1), a CYBDOM protein constituted of a DOMON and a cytochrome *b*561 domain, as critical to maintain cell elongation and meristem integrity under low P. We demonstrate that HYP1 mediates ascorbate-dependent trans-plasma membrane electron transport and can reduce ferric and cupric substrates in *Xenopus laevis* oocytes and *in planta*. *HYP1* expression is up-regulated in response to P deficiency in the proximal zone of the root apical meristem. Disruption of *HYP1* leads to increased Fe and callose accumulation in the root meristem and causes significant transcriptional changes in roots. We further demonstrate that HYP1 activity overcomes malate-induced Fe accumulation, thereby preventing Fe-dependent root growth arrest in response to low P. Collectively, our results uncover an ascorbate-dependent metalloreductase that is critical to protect root meristems of P-deficient plants from increased Fe availability and provide insights into the physiological function of the yet poorly characterized but ubiquitous CYBDOM proteins.

Phosphorus (P) is essential for undisturbed plant growth and development. However, in most soils, total amounts of P are very low or the P that is present is only poorly available to plants. The latter is because a large portion of inorganic phosphate – the P form taken up by plants – is precipitated with Ca or strongly sorbed on iron (Fe) and aluminum (Al) (hydr)oxides[1]. To cope with limiting P availability, plants have evolved a series of mechanisms to recycle internal P sources and to acquire P from the soil. One of these mechanisms is the release of low-

[1]Leibniz Institute of Plant Genetics & Crop Plant Research (IPK) OT Gatersleben, Corrensstr 3, 06466 Seeland, Germany. [2]Institute of Biophysics, National Research Council, Via De Marini 16, 16149 Genoa, Italy. [3]Institute for Drug Discovery, Leipzig University, SAC 04103 Leipzig, Germany. [4]Center for Scalable Data Analytics and Artificial Intelligence, Leipzig University, 04105 Leipzig, Germany. [5]Interdisciplinary Center for Bioinformatics, Leipzig University, 04107 Leipzig, Germany. [6]Department of Earth, Environment and Life Sciences (DISTAV), University of Genoa, Viale Benedetto XV 5, 16132 Genoa, Italy. ✉e-mail: giehl@ipk-gatersleben.de

molecular-weight carboxylates, such as malate, which can form complexes with Fe(III) (ferric Fe) and Al(III), thereby releasing sorbed phosphate[2]. This P-mining strategy likely evolved in plants growing in acidic soils with very low P concentrations[3]. The increased release of malate by P-deficient roots does not only increase the availability of P but also of Fe[4]. The impact of this interaction has been characterized in more detail in *Arabidopsis thaliana*. In this species, root inhibition under low P can be attenuated by removing Fe from the growth medium or by disrupting P deficiency-induced malate release[5–9]. Furthermore, it has been shown that over-accumulation of labile Fe in the apoplast of the apical region of roots triggers ectopic deposition of callose in the root apical meristem (RAM) and increased cell wall stiffening in the elongation zone[5,6,10]. While Fe-dependent primary root attenuation stimulates the development of a shallow and highly branched and root system[11,12] that helps plants to forage P from the topsoil[13], if not counteracted, excess Fe may ultimately disrupt stem-cell niche identity and RAM integrity[5].

Fe uptake in most plant cells and organelles is driven by Fe(III) reduction via chemical reductants and membrane-bound metalloreductases of the ferric reductase oxidase (FRO) family[14]. In *A. thaliana* roots, Fe reduction relies on plasma membrane (PM)-bound FRO2 and the release of redox-active coumarins[15,16]. Interestingly, *FRO2* is not expressed in the root apical zone and is repressed in response to P deficiency elsewhere[17,18], whereas clear *FRO3* expression in the RAM is only detected in Fe-deficient roots[19]. Furthermore, since the promoters of genes encoding the main coumarin biosynthesis enzymes are only active in the root hair zone[16,20], it remains elusive how root tips acquire Fe.

Here, we identify and characterize a PM-bound electron transfer protein that is induced in response to P deficiency in the root apical zone of *A. thaliana* plants. Due to the more severe loss of meristematic integrity and more significantly inhibited cell elongation of a T-DNA insertional mutant specifically under low P conditions, we named the gene *HYPERSENSTIVE TO LOW P1* (*HYP1*). The protein is a member of the yet poorly characterized CYBDOM family and consists of a plasma membrane-embedded cytochrome *b*561 domain fused to an apoplastic dopamine ß-monooxygenase N-terminal (DOMON) domain. We show that HYP1 is present in root tips, mediates ascorbate-dependent trans-PM electron transport, and can reduce ferric chelates and cupric [Cu(II)] ions. Finally, we demonstrate that HYP1-driven ferric reduction is critical to prevent malate-induced Fe overaccumulation in the apoplast and to maintain meristem integrity under low-P conditions.

## Results

### Identification of a CYBDOM mutant with hypersensitive root growth inhibition under low P

To identify genes required for P-dependent root growth responses, we re-analyzed a time-course root transcriptome of *A. thaliana* (Col-0 accession) plants grown on sufficient (625 μM) or low (100 μM) P concentrations. Previously, we used this dataset to specifically assess the expression of genes involved in P deficiency-induced root hair development[21]. Here, we integrated all data and performed an in-depth analysis, including an additional time point (i.e., 6 days after transfer) from the same experiment that was not considered in the previous analysis. Using $\log_2$-fold change (FC) ≥ |1.0| and FDR < 0.05 as cut-off parameters, we identified 2,327 genes that responded significantly to low P in at least one-time point (Fig. 1a; Supplementary Data 1), which were distributed into four major clusters. Among the cluster containing genes down-regulated in response to P deficiency (cluster 1, 901 genes), the gene ontology (GO) terms comprised by the cluster "iron metal homeostasis" were strongly enriched (Fig. 1b; Supplementary Data 2). The expression of major genes involved in root Fe uptake, including *FRO2* and *IRT1*, was indeed down-regulated, as reported earlier[18,22–24]. Since we were interested in genes related with ferric Fe reduction, we first interrogated the expression of genes encoding the

membrane-bound FROs and the enzymes involved with the synthesis of coumarins or ascorbate. The expression of *S8H* and *CYP82C4*, critical for the synthesis of redox-active coumarins, was strongly down-regulated under low P, while only *FRO8* was consistently up-regulated among the *FRO*s (Fig. 1c; Supplementary Data 3). However, FRO8 has been predicted to reside in mitochondrial membranes, a localization partially supported by the analysis of *A. thaliana* mitochondrial proteome[25]. Thus, our results indicated that ferric reduction by PM-bound FRO-type metalloreductases and coumarins is decreased in P-deficient roots. In line with a previous study[7], the ascorbate biosynthesis gene *VITAMINC2* (*VTC2*) and *VTC4* were up-regulated in response to P deficiency, reinforcing a potential role for ascorbate under these conditions.

We then focused our search on other potential ferric reductases among P deficiency-induced genes. By screening for the presence of relevant protein domains, such as "ferric reductase" or "Fe/Cu oxidase", we identified one gene, At5g35735, which was present in cluster 2 and predicted to encode a protein with a "Cytochrome b561/ferric reductase transmembrane" domain according to InterPro (www.ebi.ac.uk/interpro/). We phenotyped one available insertion line, carrying a transfer DNA (T-DNA) insertion in the sole intron of the previously uncharacterized gene At5g35735, and found that the mutant exhibited more severe primary root inhibition than the corresponding wild type (Col-0) when grown on low-P medium (Fig. 1d). Based on the conditional phenotype, we named the mutant hypersensitive to low P 1 (*hyp1*). Since Fe(III) complexed with malate is prone to blue light-induced photoreduction, which increases the risk of Fe(II)-driven hydroxyl radical formation in the apoplast[26], we repeated the phenotypic analysis and shielded the roots from direct light exposure. With this system, only little light reached the roots, and light-induced Fe(III) reduction directly in the agar medium was almost completely prevented (Supplementary Fig. 1a–c). Since we aimed to investigate root responses induced by malate-driven Fe complexation and solubilization, we then provided Fe as the non-chelated form $FeCl_3$. Although the amount of Fe supplied as $FeCl_3$ was higher than that provided as Fe(III)-EDTA in standard Murashige and Skoog medium, only a small fraction was soluble (Supplementary Fig. 1d). When plants were grown under this modified cultivation setup, the severe primary root inhibition of *hyp1* plants under low P supply still persisted (Fig. 1e and Supplementary Fig. 1e). We then used this cultivation setup (i.e., light-shielded roots and non-chelated Fe source) in all further experiments simulating P-limiting conditions in this study.

The expression of *HYP1* in the *hyp1* T-DNA insertion mutant is strongly impaired (Supplementary Fig. 2a), and the severe root inhibition of *hyp1* could be fully rescued by re-introducing a construct containing the full genomic coding sequence of *HYP1*, including a 2.1-kb-long promoter region (Fig. 1e). Since the intron between the two exons of *HYP1* is a putative rolling-circle (*Helitron*) transposon element (Supplementary Fig. 2b), we also expressed *HYP1* complementary DNA (cDNA) under the control of the *HYP1* promoter in the *hyp1* mutant to rule out the possibility that the predicted transposon element was involved with the short-root phenotype. *HYP1* cDNA also complemented the hypersensitive primary root inhibition of *hyp1* under low P (Supplementary Fig. 2c, d). Thus, these results demonstrate that disruption of *HYP1* is causal for the P deficiency-induced hypersensitive root phenotype of the isolated T-DNA line.

An earlier phylogenetic analysis has identified HYP1 as a CYBDOM-type protein[27]. These proteins possess a cytochrome *b*561 fused to a DOMON domain, which was initially identified in dopamine β-hydroxylase, the enzyme that generates norepinephrine from dopamine[28,29]. Among all 11 genes encoding for DOMON-containing proteins in *A. thaliana*, 10 are CYBDOMs while one, *AUXIN-INDUCED IN ROOT CULTURES 12* (*AIR12*), encodes for a protein constituted by a single DOMON associated to a glycosylphosphatidylinositol (GPI) membrane anchor[27]. *HYP1* was the only member of the DOMON superfamily

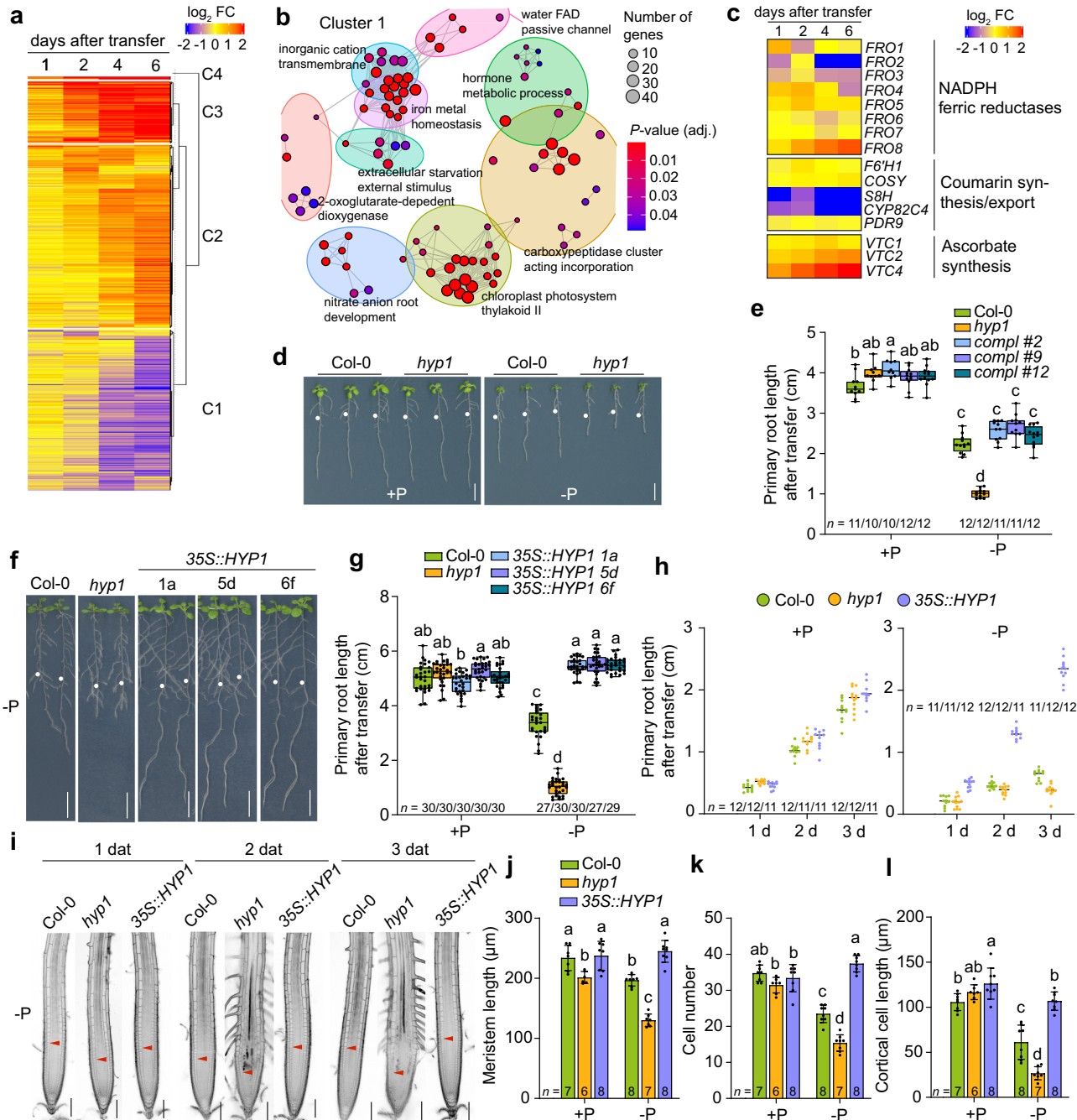

**Fig. 1 | Identification of an *Arabidopsis* mutant hypersensitive to low P. a** Heat map for hierarchically clustered genes significantly regulated ($|\log_2 FC| \geq 1$ -P *versus* +P, FDR < 0.05) under low P in roots (n = 3 independent root pools). **b** GO term enrichment network of significantly enriched terms of cluster 1 (*P*-values, adjusted Benjamini-Hochberg FDR). **c** Heat map showing differential expression ($|\log_2 FC| \geq 1$ -P *versus* +P, FDR < 0.05) of selected genes. **d**, **e** Identification of a mutant with hypersensitive low P-induced primary root inhibition. Visual appearance of wild-type (Col-0) and *hyp1* mutant plants (**d**) and primary root elongation of wild-type (Col-0), *hyp1* mutant, and three independent lines expressing *proHYP1::HYP1* in *hyp1* (*compl*) 6 days after transfer (**e**). In **a**–**e**, seven-day-old seedlings were transferred to fresh medium containing 625 µM P (+P) or 5 µM P (-P) with 100 µM Fe(III)-EDTA. **f**, **g** *HYP1* overexpression improves root growth under low P. Appearance of plants (**f**) and primary root elongation of wild-type (Col-0), *hyp1* mutant, and three independent transgenic lines expressing *35S::HYP1* in Col-0 (**g**). **h**, **i** Time-

dependent changes in primary root length (**h**) and meristem integrity (**i**) after transfer to +P or -P (**h**) or only -P (**i**). **j**–**l**, Quantification of meristem cell length (**j**), meristem cell number (**k**) and mature cortical cell length (**l**). In **f**–**l**, ten-day-old seedlings were transferred to fresh medium containing 625 µM P (+P) or 5 µM P (-P) with 150 µM FeCl₃ and analyzed after 3 days (**j**–**l**), 6 days (**f**, **g**) or at the indicated dates after transfer (**h**, **i**). For the box plots in **e** and **g**, horizontal line, median; edges of boxes, 25th (bottom) and 75th (top) percentiles; whiskers, minimum and maximum values; and dots, individual biological replicates (n = independent roots as indicated). In **h**, dark horizontal lines mark means and in **j**–**l**, bars represent means ± SD. Different letters indicate significant differences (one-way ANOVA followed by post-hoc Tukey's test, *P* < 0.05). The exact *P*-values are provided as a Source Data file. In **d** and **f**, white dots indicate primary root position at the day of transfer, and arrowheads in **i** the boundary between meristem and transition zone. Scale bars, 1 cm (**d**, **f**) and 100 µm (**i**). Source data are provided as a Source Data file.

differentially regulated in response to P deficiency (Supplementary Fig. 3a, b). Nonetheless, we assessed the primary root length of T-DNA insertion mutants available for 9 DOMON-expressing genes. Besides *hyp1*, only T-DNA insertions disrupting the expression of At3g07390 (*AIR12*) and especially At3g25290 (*CRR*) showed significantly shorter primary root length compared to wild type, while most lines showed no or only small differences compared to wild-type plants (Supplementary Fig. 3c, d). The simultaneous disruption of *HYP1* and *CRR* further exacerbated the primary root inhibition under low P (Supplementary Fig. 3e, f), suggesting that the activities of these two CYBDOM proteins act in an additive way to sustain root growth under low-P conditions. In the accompanying manuscript, Clúa et al.[30] provide a comprehensive characterization of CRR. Next, we overexpressed *HYP1* in wild-type plants using the *CaMV 35S* promoter. Ectopic expression of *HYP1* was able to completely prevent the inhibition of primary root elongation of plants grown in low-P medium (Fig. 1f, g). Altogether, our results demonstrate that HYP1 increases the tolerance of roots to low P-induced inhibition of primary root elongation.

## HYP1 is critical for the maintenance of meristem integrity and cell elongation under low P

According to our transcriptome analysis, *HYP1* expression is gradually induced in roots in response to P deficiency (Supplementary Fig. 3a). A time-course analysis of primary root length after transfer to low P revealed that the primary root of *hyp1* plants ceases to elongate from 2 days after transfer onwards (Fig. 1h). A closer inspection of primary root tips revealed that the loss of *HYP1* decreased the meristem size, and inhibited the elongation of mature cells (Fig. 1i–l and Supplementary Fig. 4a–d). A significant decrease in mature cell length was observed already 2 days after transfer to low P, one day before the inhibition of meristem length became significant (Supplementary Fig. 4a–d). Probably as a result of the smaller meristems and shorter cells, and potentially of accelerated cell differentiation, root hairs were detected much closer to the primary root apex of *hyp1* than in wild-type plants under low P (Fig. 1i). To assess stem cell integrity, we then checked the localization of *WOX5* in the RAM. Under low-P conditions, *proWOX5::GFP*-derived fluorescence was not anymore confined to quiescent center (QC) cells but detected in several meristematic cells (Supplementary Fig. 4e), indicating the disintegration of the stem cell niche. Finally, analysis of mitotic activity with *proCYCB1;1::GUS* reporter revealed that loss of HYP1 activity decreased the number of cells undergoing division in the RAM in response to low P (Supplementary Fig. 4f). Taken together, these results show that the hypersensitive primary root phenotype of *hyp1* plants under low P is due to loss of RAM integrity and inhibition of cell expansion in the elongation zone.

## HYP1 is a plasma membrane protein expressed in different root zones

To get insights into the tissue-specific expression of *HYP1*, we generated *proHYP1::GUS* lines. In young seedlings, we detected *proHYP1*-driven GUS activity mainly in the two cotyledons and throughout most parts of primary and lateral roots, including the apical zone (Fig. 2a). In line with the transcriptome analysis, the intensity of GUS staining increased when plants were cultivated on low P, reinforcing that *HYP1* is induced in response to P deficiency. A close-up analysis of root tips revealed that *HYP1* promoter activity was prominent in root cap cells and, under low P, strongly increased in cells surrounding the stem cell niche (Fig. 2b). To assess in more detail where the HYP1 protein is located, we generated a translational fusion carrying GFP in the C-terminal part of HYP1, which was driven by the *HYP1* promoter. Under sufficient P, HYP1:GFP was detected in the vasculature along the mature root hair zone, whereas in the root apex, it was confined to the root cap and mature columella cells (Fig. 2c and Supplementary Fig. 5). Low P availability increased the abundance of HYP1:GFP and, in the RAM, induced its appearance in several cell files above the stem cell

niche while having no effect on HYP1:GFP-derived fluorescence in root cap cells (Fig. 2d and Supplementary Fig. 5). To verify the subcellular localization of HYP1, we used the styryl dye FM4-64 that intercalates into membranes and observed defined colocalization of HYP1:GFP with this dye (Fig. 2e). Moreover, in plasmolyzed root cells, strong HYP1:GFP-derived fluorescence was detected in the PM and in Hechtian strands (Fig. 2f). Thus, these results demonstrate that HYP1 is a PM protein present in distinct domains of the root, including the RAM.

## HYP1 mediates ascorbate-dependent trans-PM electron transport

Cytochrome *b*561-containing proteins have been shown to catalyze electron transport between a cytosolic electron donor and acceptors located on the noncytoplasmic side[31,32]. To explore the transport properties of HYP1, we performed a series of functional assays in *Xenopus laevis* oocytes. Employing a two-electrode voltage clamp, we detected inward currents in oocytes injected with *HYP1* cRNA when ferricyanide was provided as an electron acceptor in the bathing solution (Fig. 2g, h). The currents strongly increased after raising the cytosolic concentrations of ascorbate in the oocytes (Supplementary Fig. 6a, b). The results could be fitted by a Michaelis-Menten equation revealing an apparent affinity constant of 4 mM, which is in line with the approx. 20 mM ascorbate detected in the cytosol of plant cells[33]. Similar to other cytochrome *b*561-containing proteins characterized up to date, we found that ascorbate was a specific cytosolic electron donor for HYP1 with reduced glutathione not being able to enhance HYP1-mediated currents (Supplementary Fig. 6c, d). Current amplitudes were also stimulated by increasing the concentration of the electron acceptor in the bathing solution again following patterns resembling Michaelis-Menten kinetics (Supplementary Fig. 7a–d). To rule out the possibility that the detected currents were associated with ion fluxes, we manipulated the pH and ionic composition of the bathing solutions. Replacing the standard bathing solution with a solution containing only the membrane-impermeable ions 1,3-bis[tris(hydroxymethyl)methyl-amino]propane (BTP) and 2-(N-Morpholino)ethanesulfonic acid (MES) or raising the external pH to 7.5, did not significantly alter the current amplitude detected in HYP1-injected oocytes (Supplementary Fig. 7e, f). Thus, these results demonstrate that HYP1 mediates trans-PM electron transport, an activity that is influenced by the concentration of cytosolic ascorbate and extracytoplasmic electron acceptors.

## Structural features of HYP1 critical for its function in oocytes and *in planta*

While the crystal structure of two cytochrome *b*561 proteins – CYBASC1 (*At*Cytb$_{561}$) of *A. thaliana* and the human duodenal cytochrome *b* (Dcytb) – have been determined[34,35], no structural information is available for CYBDOM proteins. Therefore, we predicted HYP1 structure with AlphaFold[36]. Our model revealed that the cytochrome *b*561 domain is formed by five α-helical trans-membrane segments while the DOMON resembles a sandwiched ß-sheet structure (Fig. 3a). The presence of a secretory signal peptide in the N-terminal part and HYP1 localization in the PM (Fig. 2e, f and Supplementary Fig. 2b) suggest that HYP1's DOMON faces the apoplast. Next, we modeled the putative binding site for ascorbate on the cytosol-facing side using Rosetta ligand docking calculations. For guiding the modeling calculations, we used the structure of ascorbate-bound cytochrome *b*561 (PDB ID: 4O79). The predicted binding model for ascorbate indicates that hydrogen bond interactions with the sidechains of residues arginine 307 (R307) and asparagine 311 (N311) and possibly with the backbone of proline 302 (P302) help to stabilize the binding of ascorbate (Fig. 3b). Furthermore, our structural model revealed that two conserved histidine pairs H245–H314 and H211–H278 are putative coordination sites to two *b*-heme groups, one at the cytoplasmic side and another at the apical side of the cytochrome *b*561 domain,

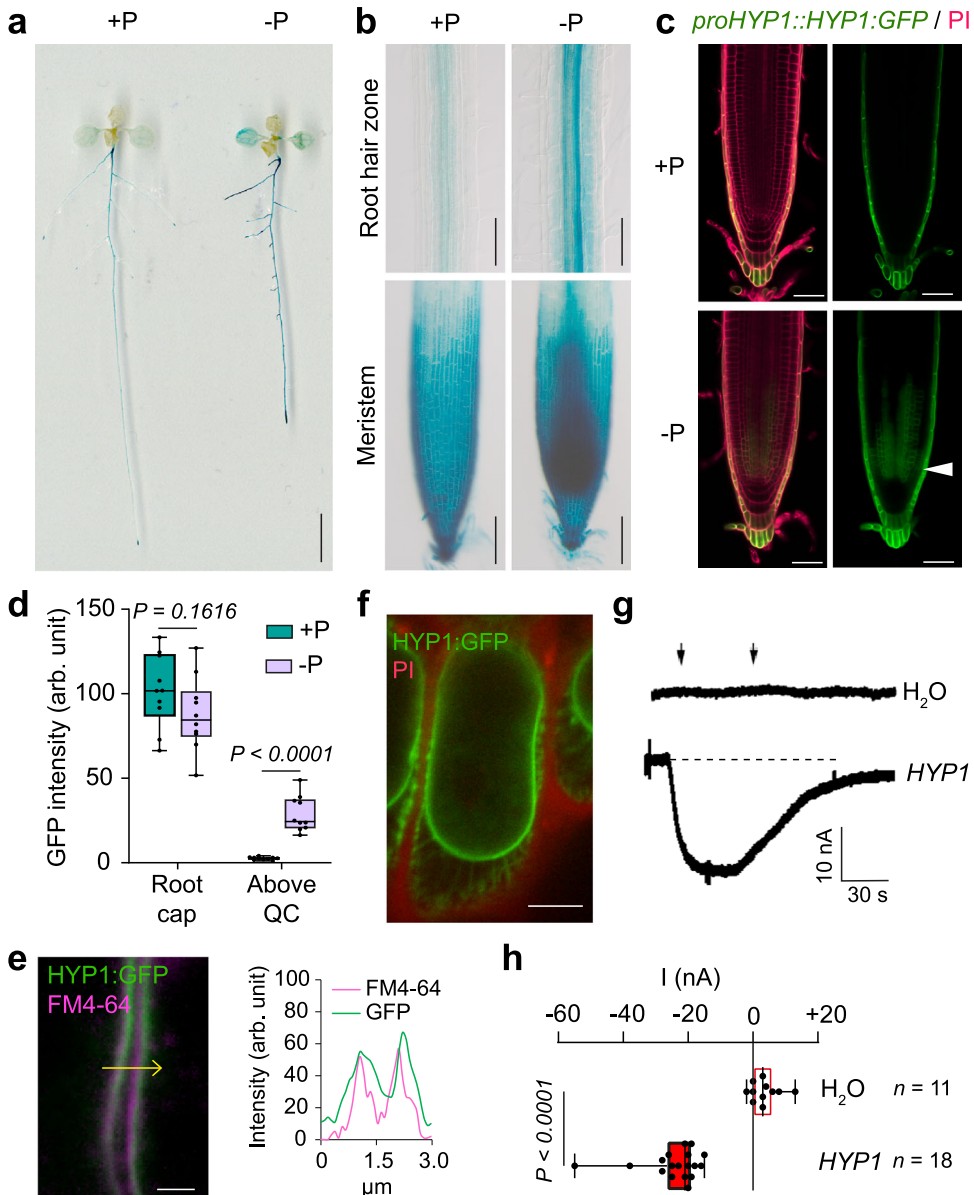

**Fig. 2 | HYP1 resides in the plasma membrane and mediates trans-plasma membrane electron transport. a, b** P-dependent changes in *HYP1* promoter activity in whole seedlings (**a**) and in the indicated zones of the primary root (**b**). **c, d** HYP1:GFP localization in the apical root meristem (**c**) and signal quantification in root cap and in cells located above the QC (*n* = 10 independent roots) (**d**) in response to low P. Cell walls were stained with propidium iodide (PI). The arrowhead indicates the increased abundance of HYP1:GFP in the cells above the stem cell niche. In **a**–**d**, ten-day-old seedlings were transferred to fresh medium containing 625 μM P (+P) or 5 μM P (-P) with 150 μM FeCl₃. GUS staining and HYP1:GFP localization were assessed on plants grown in the indicated treatments for 3 days. **e, f** HYP1:GFP is localized in the plasma membrane. Localization and fluorescence intensity of GFP and the membrane stain FM4-64 (**e**) in the section marked with the yellow arrow, and HYP1:GFP localization in plasmolyzed cells counterstained with

PI (**f**). The experiments were repeated two times with similar results and representative images from one experiment are shown. **g, h** Trans-plasma membrane current recordings (**g**) and calculated mean currents (**h**) recorded in *X. laevis* oocytes injected with water (H₂O) or cRNA of *HYP1* and exposed to 1 mM of the water-soluble electron acceptor [Fe(CN)₆]³⁻ (ferricyanide) in standard bath solution (pH 5.5) at a holding potential of −20 mV (*n* = 11 independent oocytes for H₂O-injected; *n* = 18 independent oocytes for HYP1-injected). For the boxplots in **d** and **h**, horizontal line, median; edges of boxes, 25th (bottom) and 75th (top) percentiles; whiskers, minimum and maximum values; and dots, individual biological replicates. In **d** and **h**, *P*-values according to two-sided Student's *t*-test. In **g**, the left and right arrows indicate the addition and removal of ferricyanide, respectively. Scale bars, 1 cm (**a**), 100 μm (**b**), 50 μm (**c**), 2 μm (**e**) and 5 μm (**f**). Source data are provided as a Source Data file.

respectively (Fig. 3c). According to our model, the Fe-to-Fe distance between the two *b*-heme groups is approx. 23.5 Angstrom (Å), which is comparable to the distances reported for the crystal structures of the cytochrome *b*561 proteins CYBASC1 (*At*Cytb₅₆₁) and Dcytb[34,35]. We confirmed the critical function of the H211–H278 pair for HYP1 function by replacing these positively charged, imidazole-containing histidine residues with non-polar leucine. The concomitant H211L and H278L substitutions completely abolished HYP1-mediated electron transport in oocytes without affecting its PM localization (Fig. 3d, e and

Supplementary Fig. 8c, d, g). The mutated HYP1^{H211L/H278L}:GFP variant also lost the ability to complement the short-root phenotype of *hyp1* in contrast to native HYP1:GFP (Fig. 3f).

Previous studies indicated that DOMON domains can bind *b*-heme or sugars[27,37]. Based on the de novo structure model of HYP1 and compared to cellobiose dehydrogenase templates (PDBs 4QI3B, 1PL3, 4QI7), which contain a heme-binding site in a DOMON-like structure, we identified methionine 90 (M90) as a candidate for *b*-heme coordination. Another potential *b*-heme coordinating

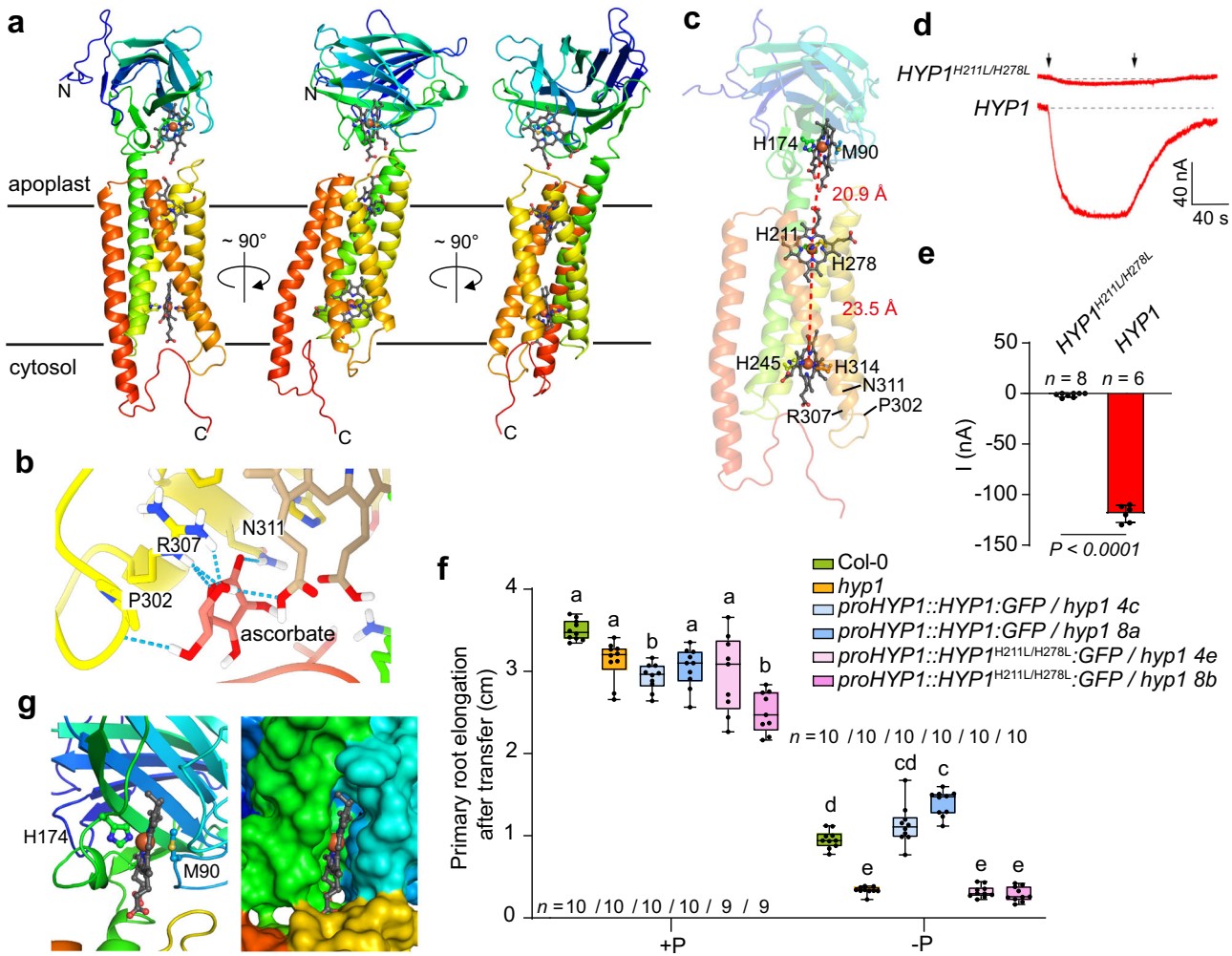

**Fig. 3 | Predicted structural organization of the cytochrome *b*561 and DOMON domains of HYP1 and residues critical for electron transfer. a–c** Structural model of HYP1 predicted by AlphaFold (**a**), docking site of ascorbate in the cytosol-facing side of the cytochrome *b*561 core (**b**), and predicted coordination of two *b*-hemes on the cytochrome *b*561 core and one on the secreted DOMON domain (**c**). The distance between the Fe centers of neighboring *b*-hemes is indicated. **d**, **e** Trans-PM current recordings in *X. laevis* oocytes injected with wild-type *HYP1* cRNA or with a mutated *HYP1* variant carrying substitutions in the *b*-heme coordination sites in the apical part of the cytochrome *b*561 core (*HYP1^H211L/H278L^*). Current traces (**d**) and mean current amplitudes (**e**) recorded with [Fe(CN)$_6$]$^{3-}$ (ferricyanide) after injection of 10 mM ascorbate in standard bath solution (pH 5.5) at a holding potential of -20 mV (*n* = 6 independent oocytes for HYP1; *n* = 8 independent oocytes for *HYP1^H211L/H278L^*). The left and right arrows indicate the addition and removal of ferricyanide, respectively. Bars represent means ± SD. *P*-value according

to two-sided Student's *t*-test. **f** Complementation of *hyp1* with GFP translational fusions of native HYP1 or the indicated mutated variant (*n* = independent roots as indicated). Ten-day-old seedlings were transferred to fresh medium containing 625 μM P (+P) or 5 μM P (-P) with 150 μM FeCl$_3$ and primary root length analyzed after 6 days. For the box plots, horizontal line, median; edges of boxes, 25th (bottom) and 75th (top) percentiles; whiskers, minimum and maximum values; and dots, individual biological replicates. Different letters indicate significant differences (one-way ANOVA followed by post-hoc Tukey's test, *P* < 0.05). The exact *P*-values are provided as a Source Data file. **g** Computational docking of a *b*-heme molecule by a conserved methionine and histidine pair in the DOMON domain, and surface presentation showing a predicted solvent-exposed side. In **a**, **c**, and **g**, red spheres indicate the atom Fe at the center of a *b*-heme. Source data are provided as a Source Data file.

residue is H174, which is at a similar location as H176 in cellobiose dehydrogenase (PDB 4QI3) and in close spatial proximity to M90. In turn, other histidine residues in the DOMON domain, including the conserved histidine at position 54, are unlikely to participate in *b*-heme coordination as they are too distant from M90 or H174. According to our AlphaFold model, one edge of the *b*-heme putatively coordinated by the M90–H174 pair in the DOMON domain is solvent-accessible (Fig. 3g). The distance between two Fe centers of the apical heme of the cytochrome *b*561 domain and the heme in the DOMON is approx. 20.9 Å (Fig. 3c). Single M90L or H174L substitutions resulted in impaired PM localization of HYP1 both in oocytes and plant cells (Supplementary Fig. 8e–g).

Altogether, our results suggest the existence of a continuous electron transfer pathway across three *b*-hemes of HYP1, which links

the cytosolic electron donor ascorbate with electron acceptors located in the apoplast.

## HYP1 can mediate ferric and cupric reduction

Considering the electron transport activity of HYP1, we then explored whether HYP1 is able to reduce ferric-chelates. In oocytes, HYP1-mediated currents were detected when different Fe(III) chelates, including Fe(III)-malate, were supplied in the bathing solution (Fig. 4a, b and Supplementary Fig. 9a). In order to assess the putative role of HYP1 as a ferric reductase *in planta*, we expressed *HYP1:GFP* in the *fro2* mutant using the *FRO2* or CaMV *35S* promoter. Expression of either construct in *fro2* plants significantly increased the root ferric-chelate reductase activity by approx. 1.6 times, allowing these plants to reach up to 60% of the activity detected in wild-type plants (Fig. 4c).

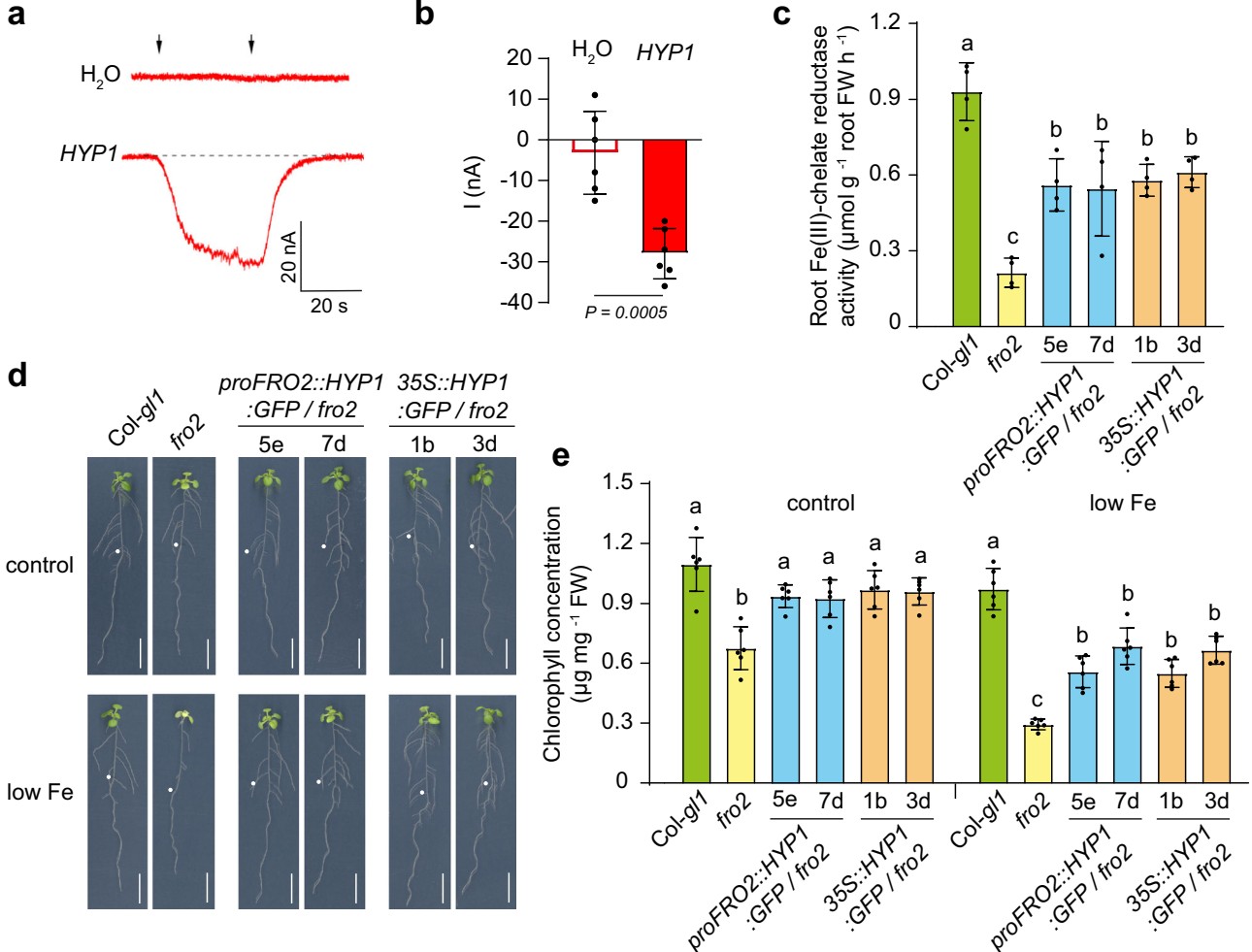

**Fig. 4 | Ferric-chelate reductase activity of HYP1. a, b** Trans-plasma membrane currents (**a**) and mean amplitudes (**b**) elicited by 1 mM Fe(III)-malate in *X. laevis* oocytes injected with water ($H_2O$) or cRNA of *HYP1* (*HYP1*). Oocytes were pre-injected with 10 mM ascorbate and traces recorded at a holding potential of -120 mV. The left and right arrows indicate the addition and removal of Fe(III)-malate, respectively. Bars represent means ± SD (*n* = 6 independent oocytes). *P*-value according to two-sided Student's *t*-test. **c** Ferric-chelate reductase activity of wild-type (Col-*gl1*), *fro2*, and *fro2* plants expressing *HYP1:GFP* under the control of the *FRO2* or CaMV *35S* promoter. Ten-day-old seedlings were transferred to fresh medium without added Fe and assayed after 3 days. Fe(III)-EDTA was used as substrate for the ferric-chelate reductase activity assays. Bars represent means ± SD (*n* = 4 biological replicates constituted of 6 plants each). **d, e** Complementation of the Fe-deficient phenotype of *fro2* by HYP1. Appearance of plants (**d**) and chlorophyll concentration (**e**) of wild-type (Col-*gl1*), *fro2*, and *fro2* plants expressing *HYP1:GFP* under the control of the *FRO2* or CaMV *35S* promoter and grown for 6 days on 75 μM (control) or 25 μM Fe(III)-EDTA (low Fe). Bars represent means ± SD (*n* = 6 biological replicates constituted of 6 plants each). In **c** and **d**, different letters indicate statistical significance (one-way ANOVA followed by post-hoc Tukey's test, *P* < 0.05). The exact *P*-values are provided as a Source Data file. In **d**, white dots indicate primary root position at the day of transfer. Scale bars, 1 cm (**d**). FW fresh weight. Source data are provided as a Source Data file.

Constitutive or Fe deficiency-induced *HYP1* expression was also able to alleviate the Fe deficiency symptoms of *fro2* plants by significantly increasing shoot chlorophyll concentration and shoot Fe contents (Fig. 4d, e and Supplementary Fig. 9c). These results demonstrate that HYP1 can mediate ferric-chelate reductase activity.

Since some members of the FRO family can reduce cupric ions[38,39] and *35S::HYP1* plants accumulated higher Cu concentrations in shoots when grown under low P (Supplementary Fig. 10a), we then assessed whether HYP1 can mediate cupric ion reduction. In *HYP1* cRNA-injected oocytes, trans-PM currents were also detected when the Cu(II) salts $CuSO_4$ and $CuCl_2$ were supplied in the bathing solution (Fig. 5a, b), suggesting that Cu(II) serves as acceptor for HYP1-derived electrons. We then assessed root Cu(II) reductase activity of wild-type, *hyp1*, *hyp1 crr* and *35S::HYP1* plants. While the disruption of *HYP1* did not significantly change the Cu(II) reduction capacity of roots, a small but significant decrease was detected in the *hyp1 crr* double mutant (Fig. 5c). In contrast, *HYP1* overexpression increased root Cu(II) reduction activity by two-fold. Since Cu uptake relies on a reductive

mechanism and excess Cu can inhibit root elongation[38,40], we then phenotyped plants on elevated external Cu(II) concentrations. While the primary root of *hyp1* plants was slightly longer than wild-type plants under high Cu, root growth of plants overexpressing *HYP1* was strongly inhibited (Fig. 5d, e). Closer inspection of roots showed that overexpression of *HYP1* decreased the length and number of cells in the meristem and the length of mature cells when plants were exposed to high Cu (Fig. 5f and Supplementary Fig. 10b). In *hyp1*, meristem size and cell length were less affected by high Cu and remained higher than in wild-type plants (Supplementary Fig. 10b). Altogether, our results demonstrate that HYP1 can function as Fe(III) and Cu(II) reductase and can affect the accumulation of these micronutrients in plant tissues.

### HYP1 prevents Fe overaccumulation in root meristems under low P

The ferric-chelate reduction capacity of HYP1 prompted us to investigate the role of Fe in HYP1-mediated root growth in response to P deficiency. Omitting Fe from the growth medium fully rescued the

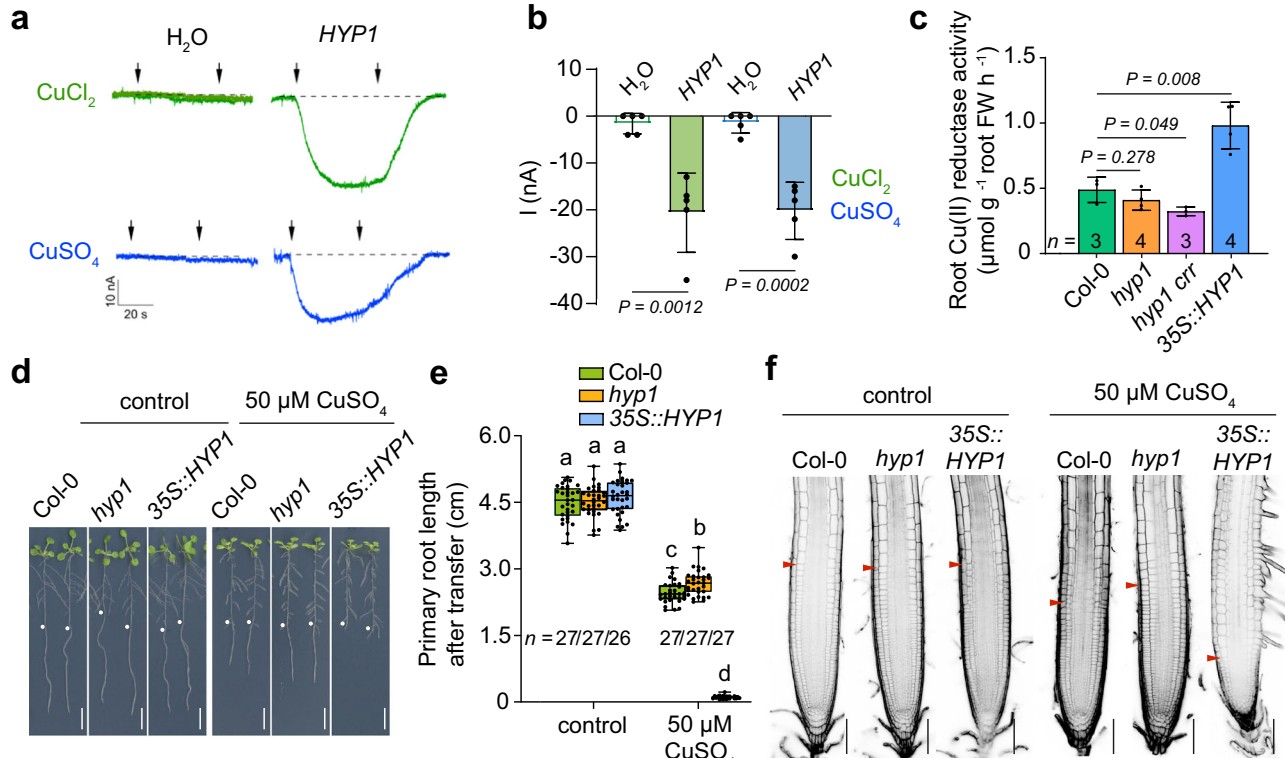

**Fig. 5 | HYP1 can mediate cupric reduction activity and alter Cu sensitivity of roots. a, b** Trans-plasma membrane current recordings (**a**) and calculated mean currents (**b**) in *X. laevis* oocytes injected with water ($H_2O$) or cRNA of *HYP1* (*HYP1*) in response to two sources of Cu(II) ($CuSO_4$ and $CuCl_2$) in standard bathing solution (pH 5.5), at a holding potential of −20 mV. The left and right arrows indicate the addition and removal of the Cu substrates, respectively. Bars represent means ± SD (*n* = 5 independent oocytes). *P* values according to two-sided Student's *t*-test. **c** Cu(II) reductase activity of wild-type (Col-0), *hyp1*, *hyp1 crr* and one transgenic line overexpressing *HYP1* (*35S::HYP1*). Ten-day-old seedlings grown on standard half-strength MS medium were used for the assay. Bars represent means ± SD (*n* = biological replicates constituted of 6 plants each as indicated in the plot). *P*-values according to two-sided Student's *t*-test. **d**–**f** *HYP1* overexpression increases

the sensitivity of primary roots to high Cu concentrations. Appearance of plants (**a**), primary root length (*n* = independent roots as indicated in the plot) (**b**) and root tip morphology (**c**). Ten-day-old seedlings were transferred to fresh medium containing 0.05 μM (control) or 50 μM $CuSO_4$ and analyzed after 6 days. For the box plots, horizontal line, median; edges of boxes, 25th (bottom) and 75th (top) percentiles; whiskers, minimum and maximum values; and dots, individual biological replicates. Different letters indicate significant differences (one-way ANOVA followed by post-hoc Tukey's test, *P* < 0.05). The exact *P*-values are provided as a Source Data file. In **d**, white dots indicate primary root position at the day of transfer, and the arrowheads in **f** the boundary between meristem and transition zone. Scale bars, 1 cm (**d**) and 100 μm (**f**). FW fresh weight. Source data are provided as a Source Data file.

primary root growth of *hyp1* plants (Fig. 6a, b). Next, we monitored Fe distribution in the apical zone of primary roots with Perls/DAB. Under sufficient P, Fe was more strongly distributed in the meristematic zone and especially near the stem cell niche, with Perls/DAB-dependent staining being comparable in wild-type and *hyp1* roots (Fig. 6c, d). This pool of stainable Fe likely reflects Fe precipitated as Fe-phosphate in the apoplast, as shown previously[5]. Under low P, Fe accumulation remained more concentrated around and within the stem cell niche but was strongly intensified in *hyp1* roots (Fig. 6c, d). Interestingly, Perls/DAB-stainable apoplastic Fe was decreased when *HYP1* was overexpressed, almost completely disappearing under low P. By staining callose with aniline blue, we found that the increased Fe accumulation in *hyp1* meristems was accompanied by increased callose deposition, especially in the stem cell niche and around cortical and endodermal cells in the meristem (Fig. 6e, f). Furthermore, we detected decreased mobility of the transcription factor SHORT ROOT (SHR) from the stele into quiescent center cells (Supplementary Fig. 11), suggesting that cell-to-cell mobility of proteins between these cells was impaired in *hyp1* roots.

To better determine where Fe was accumulating in the root meristems of low-P plants, we prepared longitudinal cross-sections of Perls/DAB-stained meristems. In root tips of wild-type plants, Fe was mainly deposited around stem cells, cortical cells and in some cells of the columella and root cap, while in meristems of *35S::HYP1* plants

stainable Fe was strongly diminished with only little deposits being detected around the QC (Fig. 6g). In *hyp1* meristems, Perls/DAB staining was more intense and was additionally detected around endodermal cells and in the stele, thus largely overlapping with the pattern of callose deposition (Fig. 6e, g). The stained Fe pools appeared mainly apoplastic, as the staining was strongly concentrated around cell boundaries and was highly reminiscent of apoplastic Fe localization reported previously[5]. Thus, these results suggested that HYP1 prevents aberrant Fe accumulation and callose deposition in the meristem of roots exposed to low-P medium.

## HYP1-dependent transcriptional responses in roots

To gain more insights into the putative role of HYP1 in Fe homeostasis and global transcriptional responses under low P, we performed a whole transcriptome sequencing (RNA-seq) analysis in root tips. Clear differences were detected when comparing the expression pattern of genes differentially expressed ($|\log_2 FC| \geq 1.0$, FDR < 0.05) in response to low P in wild type, *hyp1* and *35S::HYP1* (Fig. 6h; Supplementary Data 4). In wild-type root tips, 1113 genes were differentially expressed after 3 days of growth under low P compared to sufficient P (Fig. 6i; Supplementary Data 4). Disruption of *HYP1* resulted in the largest number of differentially expressed genes (DEGs) with 829 genes being differentially expressed exclusively in *hyp1* roots (322 up-regulated and 507 down-regulated).

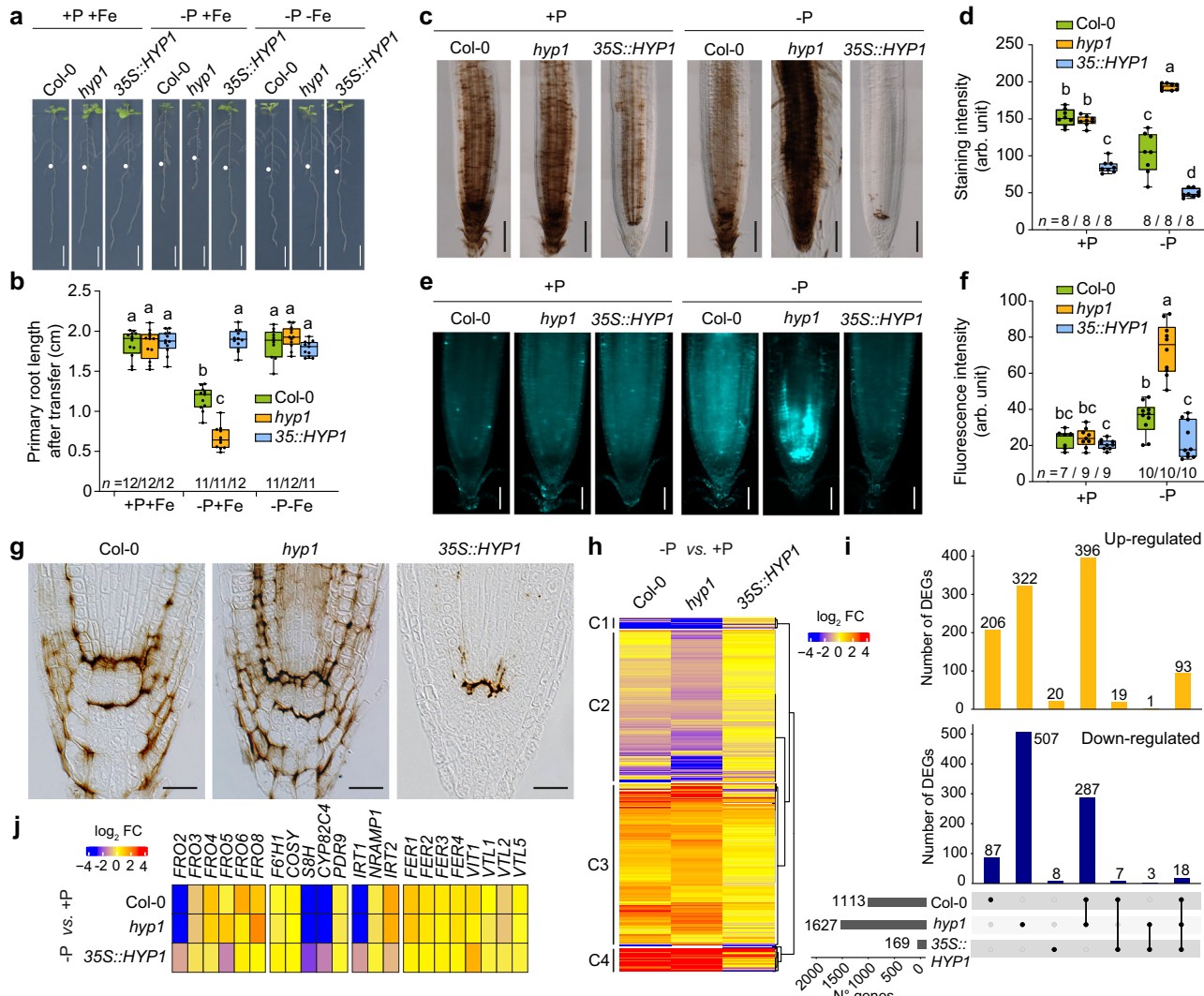

**Fig. 6 | HYP1 prevents Fe overaccumulation in the RAM under low P.**
**a**, **b** Appearance of plants (**a**) and primary root length of wild-type (Col-0), *hyp1*
mutant and one *HYP1*-overexpressing line (*n* = independent roots as indicated) (**b**).
Ten-day-old seedlings were transferred to fresh medium containing 625 µM P (+P)
or 5 µM P (−P) with 150 µM $FeCl_3$ (+Fe) or without added Fe (−Fe) and analyzed after
6 days. **c−f** Fe accumulation and distribution (**c**) and callose deposition (**e**) in the
apical region of primary roots of wild-type (Col-0), *hyp1* mutant and one *HYP1*-
overexpressing line. Quantification of Perls/DAB-stained Fe (*n* = 8 independent
roots) (**d**) and aniline blue-stained callose (*n* = independent roots as indicated) (**f**).
Ten-day-old seedlings were transferred to fresh medium containing 625 µM P (+P)
or 5 µM P (−P) with 150 µM $FeCl_3$ and roots stained after 3 days. **g** Longitudinal
sections of Perls/DAB-stained root tips 2 days after transfer from +P to -P medium.
The experiment was repeated two times with similar results and representative
images from one experiment are shown. **h** Heat map for hierarchically clustered

genes significantly regulated in response to low P ($|\log_2 FC| \geq 1$ −P *versus* +P,
FDR < 0.05) in root tips of wild type (Col-0), *hyp1* mutant and one *HYP1*-over-
expressing line (*35S::HYP1*) (*n* = 3 independent root pools). Ten-day-old seedlings
were transferred to a fresh medium containing 625 µM P (+P) or 5 µM P (−P) with 150
µM $FeCl_3$ for 3 days. **i** UpSet plots showing the number of genes exclusively or
commonly up- or down-regulated. **j** Heat map showing differential expression ($|\log_2$
FC$| \geq 1$ -P *versus* +P, FDR < 0.05) of genes related to Fe reduction, uptake or storage.
For the box plots in **b**, **d**, and **f**, horizontal line, median; edges of boxes, 25th
(bottom) and 75th (top) percentiles; whiskers, minimum, and maximum values; and
dots, individual biological replicates. Different letters indicate significant differ-
ences (one-way ANOVA followed by post-hoc Tukey's test). The exact *P*-values are
provided as a Source Data file. Scale bars, 1 cm (**a**), 100 µm (**c**), 50 µm (**e**), 20 µm (**g**).
In **a**, white dots indicate the primary root position on the day of transfer. Source
data are provided as a Source Data file.

In contrast, a comparatively small number of genes (169 in total)
was significantly altered in root tips of low P-grown *35S::HYP1* plants,
most of them being up-regulated. The majority of DEGs in *35S::HYP1*
overlapped with genes significantly responding to low P also in wild
type and *hyp1* (Fig. 6i; Supplementary Data 4). According to GO
analysis, this subset of DEGs was significantly enriched with func-
tional groups related to phosphate starvation responses, phosphate
transport and phosphate homeostasis (Fig. 6i and Supplementary
Fig. 12a), suggesting that primary P starvation responses were not
significantly disturbed by *HYP1* overexpression or disruption. This
result also indicated that *HYP1* overexpression restricted the tran-
scriptional responses mainly to P starvation and less to secondary

effects typically derived from the interaction with Fe. In fact, the
expression of many Fe-related genes, including *IRT1*, *FRO2*, *FRO8*,
*S8H*, *CYP82C4*, and *FER1*, which were strongly responsive to low P in
wild-type and *hyp1* roots, were only mildly or not differentially
regulated in *35S::HYP1* (Fig. 6j; Supplementary Data 5). Among genes
that failed to respond to low P in *hyp1* roots (i.e., up-regulated in
wild type but not in *hyp1*) we detected GO terms related to cell wall
composition or organization, while genes related to heme binding,
Fe-S cluster binding, and cell-cell junction organization were spe-
cifically down-regulated in roots of *hyp1* in response to low P
(Supplementary Fig. 12b, c). Together, these results reinforce a
putative involvement of HYP1 in maintaining Fe acquisition under

low-P conditions and highlight that processes related to P deficiency responses in the apoplast are significantly affected when HYP1 activity is lacking.

## HYP1 counteracts increased Fe availability resulting from low P-induced malate release

Considering the Fe-dependent root phenotype of the *hyp1* mutant, the HYP1-dependent ferric reductase activity in roots and the weak Perls/DAB staining in root tips of *35S::HYP1* plants, we hypothesized that HYP1 function is required to counteract the increased Fe solubility resulting from P deficiency-induced malate release. To test this possibility, we generated a *hyp1 almt1* double mutant. In the absence of ALMT1, disruption of *HYP1* did neither increase primary root sensitivity to low P nor Perls/DAB-stainable Fe pools in root tips (Fig. 7a–c and Supplementary Fig. 13a). Loss of *HYP1* in the *lpr1 lpr2* mutant background also had no effect on primary root growth or Fe staining. Thus, these results suggest that HYP1 activity becomes critical only when ALMT1-mediated malate export and LOW PHOSPHATE ROOT (LPR)-dependent apoplastic Fe(II) oxidation are active.

As malate becomes a predominant Fe(III)-chelator in P-starved roots[6,7], we next assessed HYP1-dependent ferric reduction of freshly prepared Fe(III)-malate complexes. Probably as a result of the strong down-regulation of *FRO2* in mature root zones, the capacity of roots of P-deficient wild-type plants to reduce Fe(III) decreased by approx. 50% (Fig. 7d). An even stronger decrease was detected for the *hyp1 crr* double mutant, while *HYP1* overexpression maintained the root ferric-chelate reductase activity at the same level as under sufficient P. Thus, these results demonstrate that overexpression of *HYP1* can overcome ALMT1-induced Fe solubilization by restoring, together with CRR, ferric reduction capacity in P-deficient roots.

## Locally defined HYP1 activity in the RAM sustains root growth under low P

Since *HYP1* is not only expressed in the RAM, we next investigated which expression domain was most critical for HYP1-mediated tolerance to low P by expressing *HYP1:GFP* under the control of different promoters. Recapitulating *HYP1* expression in root cap cells, in mature endodermal and vascular cells and in cells above the QC in the proximal part of the RAM with a 2151-bp *HYP1* promoter sequence fully restored primary root elongation, mature cell length and meristem size and integrity (Fig. 7e–i). Using the promoter of *BEARSKIN1* (*BRN1*) to confine *HYP1* expression in root cap cells did neither significantly improve root growth nor prevent loss of RAM integrity (Fig. 7e). Expression of *HYP1* in root cap cells did also not significantly change the RAM size and the length of mature cells (Fig. 7g–i). Expression of *HYP1* in mature endodermal cells with the promoter of *SCHENGEN1* (*SGN1*) was able to significantly increase the elongation of mature cells and partially restore primary root length but had limited effect in restoring RAM integrity and size (Fig. 7e–i). However, full recovery of primary root growth, mature cell length and RAM size and integrity was achieved when we used the promoter of *LPR1* to drive *HYP1* expression in the stem cell niche and in proximal endodermal and cortical cells in the RAM, and in endodermal cells in differentiated root zones (Fig. 7e–i). Thus, these results suggest that HYP1 activity in a confined zone in the proximal zone of the RAM is required to prevent RAM disintegration, while primary root elongation further relies on *HYP1* expression in mature endodermal cells.

Next, we monitored the impact of HYP1 on the fate of apoplastic Fe by assessing Perls/DAB-stainable Fe pools in the RAM after transferring plants from a Fe-containing to a Fe-depleted medium. When phosphate was present in the Fe-depleted medium, Fe pools were not significantly altered within 1 day after transfer and were comparable in wild type and *hyp1* (Fig. 8a, b). Under low P, Perls/DAB-stainable Fe pools were significantly depleted in the RAM of wild-type plants while remaining largely unchanged in *hyp1* plants.

We hypothesized that the HYP1-dependent depletion of stainable Fe pools could be associated with HYP1-driven reduction of soluble Fe(III)-malate complexes formed under P-limiting conditions. Indeed, Fe(III)-malate reduction capacity of excised root tips (approx. 3 mm-long) from P-deficient *hyp1* plants was significantly lower compared to wild-type and *HYP1*-overexpressing plants (Supplementary Fig. 13b). We also visualized Fe(II) in the primary root with the turn-on fluorescent probe RhoNox-1[41,42]. In order to compare RhoNox-1 signals in similar RAM areas, the staining was performed 1 day after transfer, when the size and integrity of the RAM of low P-exposed *hyp1* plants are not yet significantly altered (Fig. 1i and Supplementary Fig. 4a–d). In line with the ferric-chelate reductase activity detected in excised root tips, RhoNox-1-derived fluorescence decreased in response to low P and was significantly lower in the RAM of *hyp1* and much higher in the RAM of *35S::HYP1* compared to wild-type plants (Supplementary Fig. 13c–e). Together, these results suggested that, under low P, HYP1 promotes the depletion of apoplastic Fe pools by reducing Fe(III) solubilized by malate.

## Discussion

Ferric Fe reduction is key to several processes that maintain Fe homeostasis, including Fe uptake, allocation, and storage. Up to date, all characterized ferric-chelate reductases in plants belong to the flavo-cytochromes of the FRO family. Interestingly, plants also encode for proteins containing a cytochrome *b*561 domain, which is common in several ferric reductases active in animal cells, such as the human duodenal cytochrome *b* (Dcytb) involved with dietary nonheme Fe uptake[35,43]. In plants, the cytochrome *b*561 domain can occur as a single unit, as in diheme *b*-containing cytochrome *b*561 proteins (CYBASCs), or associated with one or more DOMON domains, as in CYBDOMs[31]. Our data uncovered a mechanism of ferric/cupric reduction in plants involving a CYBDOM protein. Although ferric reductase activity had been previously recorded for cytochrome *b*561-containing proteins in oocytes[32,44], the participation of such proteins in ferric reduction in plants had not yet been determined. By expressing *HYP1* in the *fro2* mutant, we show that HYP1 can mediate ferric-chelate reduction *in planta* and improve growth under low-Fe conditions (Fig. 4c–e). Our results also demonstrate that HYP1 can mediate trans-PM electron transfer to cupric substrates in oocytes and that root cupric reductase activity is significantly increased when *HYP1* is overexpressed (Fig. 5a–c). Other plant CYBDOMs likely also possess such activity, as a significant decrease in Cu(II) reduction was only detected in the *hyp1 crr* double mutant (Fig. 5c). Whereas the relevance of HYP1-mediated cupric reduction is still unclear, we demonstrate that the ability of HYP1 to reduce ferric chelates has a significant impact on root growth under low-P conditions.

Plants contain a larger number of *CYBDOM* genes than other organisms[27,45], suggesting that the association of DOMONs with the redox-active cytochrome *b*561 plays an important role in plants. Our AlphaFold-supported modelling indicated that the DOMON domain of HYP1 can coordinate one *b*-heme group (Fig. 3a, c), in line with previous electron paramagnetic resonance analysis of the DOMON-only protein AUXIN-INDUCED IN ROOT CULTURES 12 (AIR12; ref. 27). Thus, our results suggest that the three *b*-hemes of HYP1 form an electron conduit connecting cytosolic ascorbate with electron acceptors located in the apoplast, such as ferric and cupric substrates. Unfortunately, disruption of the predicted *b*-heme coordination site in the DOMON impaired HYP1 localization in the PM (Supplementary Fig. 8), preventing us to directly investigate its requirement for trans-PM electron transport in oocytes and for HYP1 function *in planta*. Regarding the electron donor, instead of NADPH, the common electron donor of FRO-type metalloreductases[15], cytochrome *b*561-containing proteins oxidize cytosolic ascorbate to reduce electron acceptors on the opposite side of the membrane. Our results indeed demonstrate that

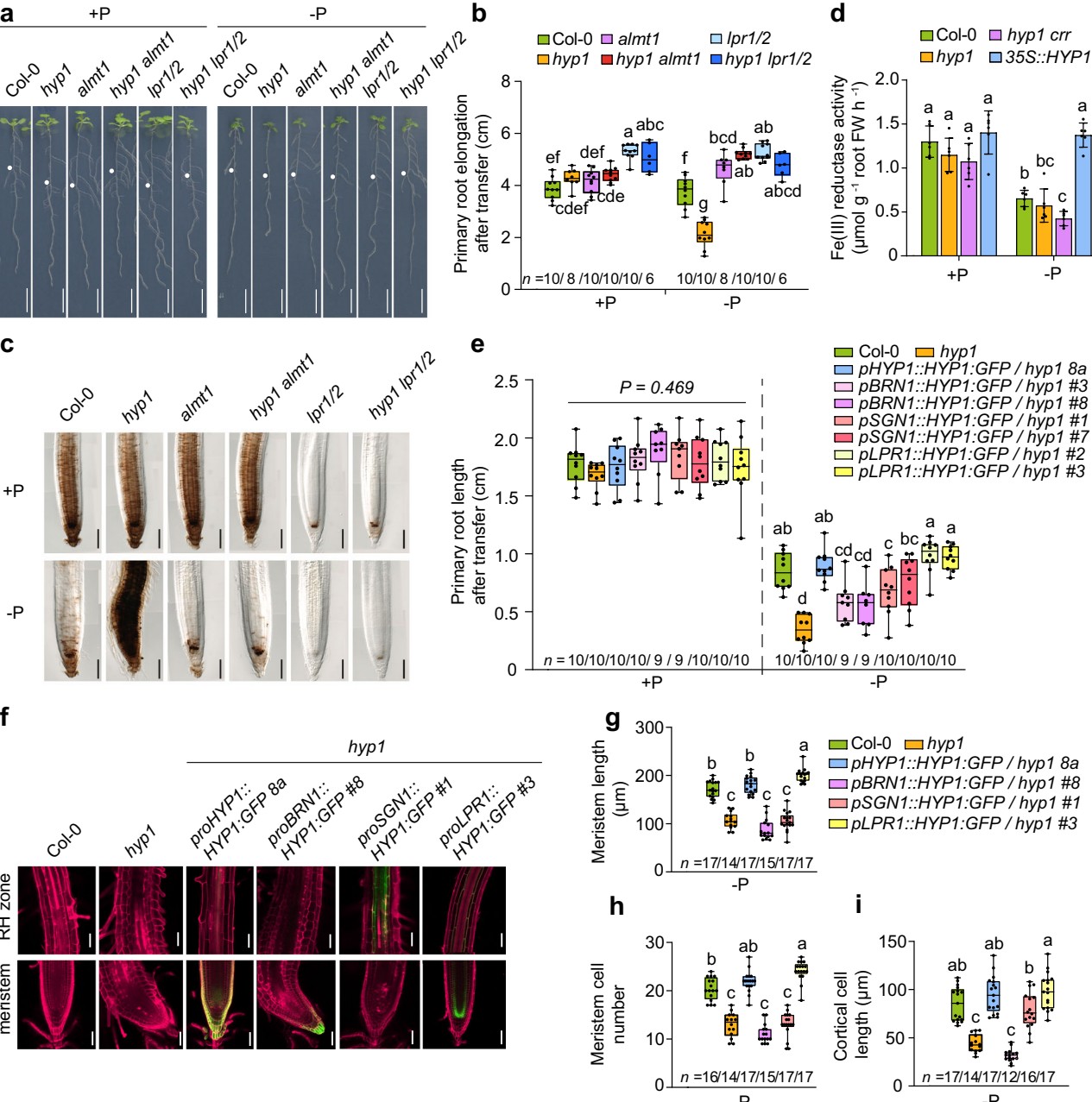

**Fig. 7 | HYP1 counteracts ALMT1-induced Fe accumulation in the apical region of P-deficient roots. a–c** HYP1 function under low P is critical only in the presence of ALMT1 and LPR1 and LPR2. Appearance of plants (**a**), primary root length after 6 days (*n* = independent roots as indicated) (**b**) and Perls/DAB staining of Fe after 3 days (**c**). **d** Root ferric-chelate reductase activity assayed with freshly prepared Fe(III)-malate complexes as substrate (*n* = 6 biological replicates consisting of 6 plants each). The assay was performed after transferring plants for 3 days to medium containing 625 μM P (+P) or 5 μM P (-P) with 150 μM FeCl₃. Bars represent means ± SD. **e–i** Cell type-specific complementation of *hyp1* with GFP translational fusions of HYP1 under the control of the indicated promoters. Primary root length (**e**), confocal images of propidium iodide-stained mature root zones and RAMs (**f**)

and quantification of meristem cell length (**g**), meristem cell number (**h**), and mature cortical cell length (**i**) (*n* = independent roots as indicated). Ten-day-old seedlings were transferred to fresh medium containing 625 μM P (+P) or 5 μM P (-P) with 150 μM FeCl₃ and analyzed after 3 days. For the box plots in **b**, **e**, and **g–i**, horizontal line, median; edges of boxes, 25th (bottom) and 75th (top) percentiles; whiskers, minimum and maximum values; and dots, individual biological replicates. In **b**, **d**, **e**, and **g–i**, letters indicate significant differences (one-way ANOVA followed by post-hoc Tukey's test, *P* < 0.05). The exact *P*-values are provided as a Source Data file. In **a**, dots indicate primary root position at the day of transfer. Scale bars, 1 cm (**a**), 100 μm (**c, j**) and 50 μm (**f**). Source data are provided as a Source Data file.

ascorbate serves as a cytosolic electron donor for HYP1 (Supplementary Fig. 6). The upregulation of genes encoding ascorbate biosynthesis genes in response to P deficiency (ref. 7; Fig. 1c) suggests that the electron donor for CYBDOM-mediated trans-PM electron transport is increased in P-deficient root cells. Thus, ascorbate may indirectly drive ferric-chelate reduction at the PM besides functioning as a chemical reductant of apoplastic ferric pools as proposed previously[7]. However,

determining whether the levels and redox state of apoplastic ascorbate in the root tips are altered depending on P availability or HYP1 activity still remains a challenge.

The accumulation of soluble Fe(III)-malate complexes in the apoplast of root tips has been proposed as a key determinant of the inhibition of primary root growth under low-P conditions[6,7]. Once mobilized, Fe is more prone to engage in apoplastic redox cycling

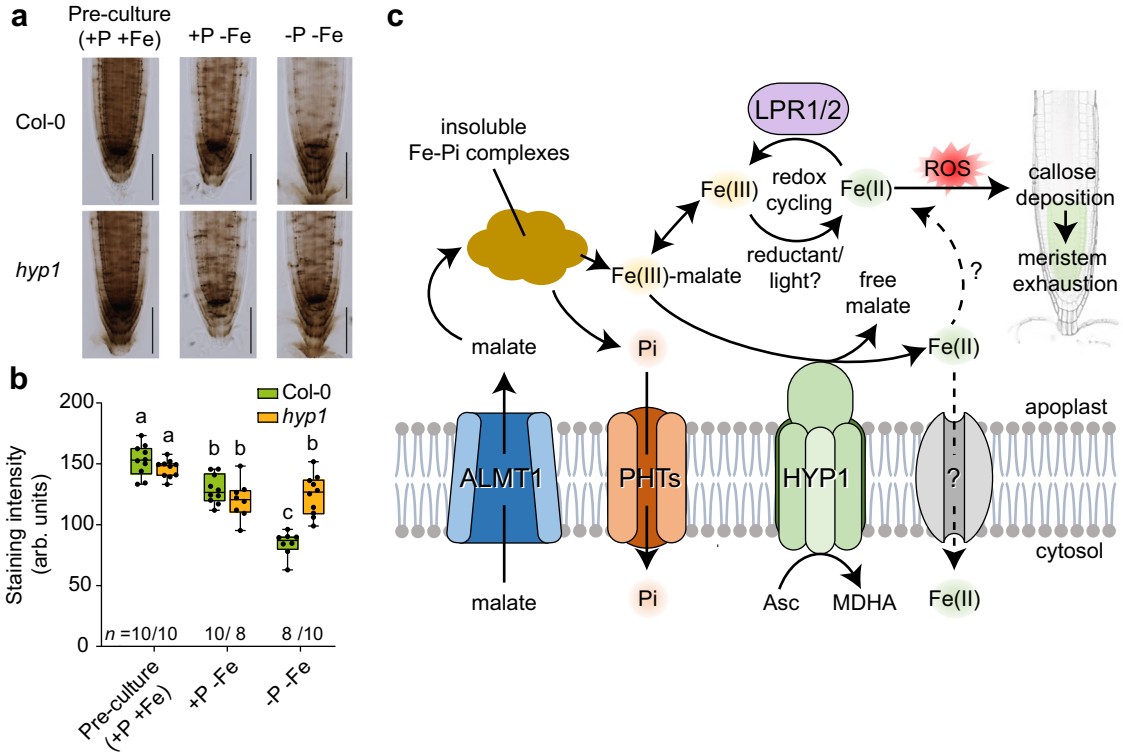

**Fig. 8 | Proposed role of HYP1 in Fe-dependent primary root adjustment to low P. a, b** *HYP1* expression in the meristem depletes apoplastic ferric Fe pools under low P. Perls/DAB staining of Fe (**a**) and quantification of signals (*n* = independent roots as indicated) (**b**) in the RAM of the indicated plants after preculture with 625μM P in a Fe-containing medium (+P+Fe) and after 1 day of growth on Fe-depleted medium containing 625 μM P (+P-Fe) or 5 μM P (-P-Fe). For the box plots in **b**, horizontal line, median; edges of boxes, 25th (bottom) and 75th (top) percentiles; whiskers, minimum and maximum values; and dots, individual biological replicates. Letters indicate significant differences (one-way ANOVA followed by post-hoc Tukey's test, *P* < 0.05). The exact *P*-values and source data are provided as a Source Data file. **c** Hypothetical model for the function of HYP1 in root adaptation to low-P conditions. Increased ALMT1-mediated export of malate in response to P

deficiency releases phosphate (Pi) from precipitated Fe-phosphate complexes while increasing the apoplastic concentration of soluble ferric-malate complexes. Ferric reduction by HYP1 facilitates the uptake of excess Fe by a yet unknown root tip-expressed Fe(II) transporter. Depletion of soluble Fe can prevent its participation in apoplastic redox cycling reactions potentially involving Fe(III) reduction by blue light or apoplastically-located chemical reductants and Fe(II) re-oxidation by the cell-wall-localized multicopper ferroxidases LPR1 and LPR2. Uncontrolled apoplastic redox cycling induces ROS-dependent aberrant callose deposition in the root apical meristem, causing meristem exhaustion. A putative interplay of HYP1 and LPRs remains elusive. ROS reactive oxygen species, PHTs phosphate transporters, Asc ascorbate, MDHA monodehydroascorbate.

reactions, producing ROS and triggering aberrant callose deposition[5]. Primary root length inhibition in relation with excessive accumulation of Fe in the apoplast has also been recorded in ammonium-stressed plants[46,47]. In the phloem of roots, ammonium-induced Fe over-accumulation induces massive callose deposition which results in decreased sucrose delivery to sustain root growth[46]. Under low P, Fe pools likely become more reactive as less phosphate is available to chemically interact with Fe while malate increases the pool of mobile Fe. Based on our findings, we hypothesize that HYP1-mediated ferric reduction prevents that excessive amounts of malate-mobilized Fe accumulate in the apoplast (Fig. 8c). Our cell type-specific complementation experiments indicated that HYP1 activity in the proximal RAM is critical to maintain meristem integrity and root growth (Fig. 7e–i). It is within this zone that *HYP1* expression and protein accumulation is most strongly induced in response to low P and where increased callose deposition is detected in the RAM of *hyp1* plants (Figs. 2b, c and 6e). In mature root zones, expression of *HYP1* in inner-root tissue (including the endodermis) can restore cell elongation but has limited ability to prevent RAM disintegration.

Our results further suggest that, in the context of P deficiency, the function of HYP1 appears to be less critical to support whole-plant Fe nutrition but rather to maintain Fe homeostasis in root tips. The identification of another CYBDOM (CRR) critical for maintaining root growth under low P in the accompanying study of Clúa et al.[30] reinforce the importance of these proteins in this process. Interestingly, the

main strategy-I mechanism of Fe uptake formed by FRO2 and the Fe(II) transporter IRON-REGULATED TRANSPORTER1 (IRT1) is not present in the root meristem and elongation zone[17,48]. Although future studies with Fe tracers will be required to quantify the Fe uptake capacity of cells within these two zones, we hypothesize that HYP1 may represent a component of a root tip-active Fe acquisition mechanism induced under low-P conditions (Fig. 8c). The up-regulation of *HYP1* in P-deficient roots indicates that the proposed root tip-active Fe acquisition mechanism is under a differential regulation compared to *FRO2* and *IRT1* in the root hair zone. One possibility to explain this difference is that cellular processes in the root apex are more sensitive to Fe-driven radical formation in the apoplast[5,47,49]. Thus, Fe-dependent modulation of root growth may attenuate root growth on low P-containing soil patches and increase exploration of P in the topsoil. In the root apex, *LPR1* is expressed in the stem cell niche and early cortical and endodermal cells[49], thus partially overlapping with HYP1 under low-P conditions. Since the activity of the multicopper oxidase LPR1 maintains higher Fe levels in the apoplast[5], this potentially increases the risk that apoplastic Fe engages in redox cycling and ROS formation. In this context, HYP1-mediated ferric reductase activity may either supply or, if the formed Fe(II) is immediately taken up, remove the substrate for LPR1. Our results favor the second scenario, as HYP1 activity is only necessary when LPRs are active (Fig. 7a–c). The ferric reduction by HYP1 could help plants to fine tune root growth in low-P soils by altering the pool of apoplastic Fe that is necessary to attenuate

root growth while simultaneously protecting RAM integrity against P deficiency-induced Fe toxicity.

Beyond P deficiency and outside the RAM, HYP1, and other CYB-DOMs might help to fine-tune other Fe-dependent redox processes in concert with LPR1 or LPR2[50,51]. Furthermore, some CYBDOMs have been found to show strong coregulation with auxin-regulated peroxidases, laccases, and GDSL-motif-containing lipases expressed in root endodermal cells[52]. Thus, it will be interesting to investigate whether the ferric and cupric reduction capacities of plant CYBDOMs can potentially modulate cell wall formation and composition and to determine whether these proteins can reduce other apoplastic electron acceptors. By establishing the phenotypic changes evoked upon altered *HYP1* expression at the organismal level, our study sheds light on the physiological roles of these poorly characterized proteins.

## Methods

### Plant materials
If not noted otherwise, the *Arabidopsis thaliana* accession Columbia-0 (Col-0) was used as wild type in this study. The following T-DNA tagged and transgenic lines were used in this study: SALK_115548 (*hyp1*), SALK_009629 (*almt1*), SALK_016297 and SALK_091930 (*lpr1 lpr2*), SALK_202530 (*crr*), SALK_013527 (At4g12980), SALK_006551 (At3g59070), SALK_092983 (At4g17280), SALK_204800 (At5g47530), SALK_110375 (At3g07570), SALK_111270 (At3g61750), SALK_089061 (At5g54830), SALK_205490 (*air12*), SALK_012348 (At2g04850), *proSHR::SHR:GFP*[53], *proCYCB1;1::GUS*[54], *proWOX5::GFP*[55]. All *A. thaliana* T-DNA insertion lines were obtained from the Nottingham Arabidopsis Stock Centre, except the *lpr1 lrp2* double mutant which was kindly provided by Prof. Dr. Steffen Abel. The lines *almt1 hyp1*, *hyp1 crr* and *lpr1 lpr2 hyp1* were generated by crossing the single *almt1* and *crr* or double *lpr1 lrp2* mutants with *hyp1*. Homozygous double or triple mutant plants were identified by PCR using the primers listed in Supplementary Table 1. The *A. thaliana* accession Columbia *glabra* (Col-*gl1*) was used as wild type for the *fro2* mutant.

### Plant growth conditions
Seeds were surface sterilized by 70% ethanol with 0.05% (v/v) Triton X-100, and pre-cultured on half-strength Murashige and Skoog salt mixture (½ MS) solid medium containing 0.5% sucrose, 1 mM MES (pH 5.5), and 1% Difco agar. Seeds were stratified for 2 days in the dark at 4 °C. Afterwards, Petri dishes were transferred to a growth chamber vertically positioned under a 22 °C/18 °C and 10-h/14-h (light/dark) regime at light intensity of 120 μmol photons m$^{-2}$ s$^{-1}$. For the microarray profiling shown in Fig. 1a–c, growth conditions inducing primary root inhibition at intermediate low-P conditions were used[11]. First, seedlings were precultured as described above for 7 days. Then, seedlings were transferred to fresh ½ MS solid medium containing either 625 μM KH$_2$PO$_4$ (+P) or 100 μM KH$_2$PO$_4$ + 525 μM KCl (-P) with Fe supplied as 75 μM Fe(III)-EDTA. Whole roots were harvested 1, 2, 4, and 6 days after transfer to treatments. Roots of 50 plants were pooled together to constitute one biological replicate. The same growth conditions were also used in the initial phenotyping experiment shown in Fig. 1d. In all other experiments, roots were shielded from direct light exposure by placing Petri dish plates inside cardboard boxes and intercalating each Petri dish with 3-cm thick black foam that extended from the bottom of the box, as shown in Supplementary Fig. 1a, b. In these experiments, 10-day-old seedlings instead of seven-day-old were transferred from the pre-culture to square Petri dishes (12 x 12 cm) containing modified half-strength MS medium with 0.5% sucrose, 2.5 mM MES (pH 5.5), 1% Difco agar, 625 μM KH$_2$PO$_4$ (+P) or 5 μM KH$_2$PO$_4$ + 620 μM KCl (-P), with or without 150 μM FeCl$_3$ (+Fe or -Fe, respectively), which was supplied as the sole Fe source after the autoclaved agar was cooled down. In experiments with complemented *fro2* plants, different Fe conditions were obtained by transferring 10-d-old seedlings to a modified half-strength MS medium with 0.5% sucrose, 1 mM MES (pH

5.5), 1% Difco agar, 75 μM Fe(III)-EDTA (+Fe, control), 20 μM Fe(III)-EDTA (low Fe), no added Fe (-Fe), as indicated in the legends of the corresponding figures. To phenotype plants under Cu excess, 10-day-old plants cultivated as indicated above were transferred to fresh solid half-strength MS medium containing 0.05 μM (control) or 50 μM CuSO$_4$. Phenotyping under low Fe and high Cu was also performed using the root-shaded setup described above.

### RNA extraction and quantitative real-time PCR
Total RNA was extracted from homogenized root samples using the NucleoSpin RNA Mini Kit (Machery-Nagel), followed by on-column DNase treatment (QIAGEN), according to the manufacturers' protocols. cDNA was synthesized from 0.5-1 μg RNA by reverse transcription using the RevertAid First Strand cDNA synthesis Kit (Thermo Fisher Scientific) and oligo(dT) primer. A ten- or twenty-times diluted cDNA sample was then used for quantitative real-time (RT) PCR analysis with the CFX384 Touch Real-Time PCR Detection System (Bio-Rad Laboratories) and the iQ SYBR Green Supermix (Bio-Rad Laboratories), using the primers listed in Supplementary Table 2. Amplification cycle protocols were as follows: 2 min at 95 °C; 40 cycles of 6 s at 95 °C and 30 s at 60 °C. Melting curves were verified at the end of 40 cycles for confirmation of primer specificities. All reactions were repeated in two technical and three biological replicates. Average Cq values were normalized by ΔΔCT formula against *UBIQUITIN 2* (AT2G36170) expression, as indicated in the legend of the corresponding figures.

### Transcript profiling and GO term enrichment analysis
For the identification of Fe-related genes responsive to P deficiency, we performed a microarray study. The expression of selected root hair-related genes from the microarray presented in this study was already reported previously[21]. Here, we present a more detailed analysis of the dataset, including one additional time-point from the same experiment. In brief, RNA was extracted from whole roots using a RNeasy plant mini kit (Qiagen) with on-column DNase treatment according to the manufacturer's instructions. cDNA was prepared using random hexamer primers and SuperScript II reverse transcriptase (Life Technologies) according to the manufacturer's protocol. RNA amplification, labeling and hybridization to Agilent microarrays (Arabidopsis V4, 021169, Gene Expression Microarray) were conducted following the manufacturer's protocol (Agilent Technologies). The full dataset was deposited in the NCBI Gene Expression Omnibus under accession GSE217790. Analysis of microarray data was performed with the R package limma[56]. Raw feature intensities were background corrected using "normexp," and "quantile" normalization method was used to normalize between arrays. Differential expression was calculated by fitting a linear model to log$_2$-transformed data by an empirical Bayes method[57]. To extract genes with significant expression differences, the cutoff |log$_2$ FC| ≥ 1 and the Benjamini–Hochberg false discovery rate (FDR) < 0.05 was applied. Significant differentially expressed genes (DEGs) over four time points between -P and +P samples were visualized in a heat map (|log$_2$ FC | ≥ 1, FDR < 0.05) generated with the hierarchical clustering (HCL) algorithm by using Pearson distance and average linkage clustering.

For RNA-seq analysis, total RNA was extracted as described above. Library construction and sequencing were performed at Novogene (UK) Company Limited. Briefly, RNA integrity and quantitation were determined with the Agilent 2100 Bioanalyzer. Total mRNA was enriched with oligo(dT) beads and randomly fragmented with fragmentation buffer. A stranded-specific library was then prepared for each sample and sequenced with a NovaSeq 6000 PE150 platform, yielding about 44 million high-quality reads per sample. After the removal of adapters, reads containing ploy-N and low-quality reads, the clean reads were aligned to the Arabidopsis genome (TAIR10) using HISAT2[58] with the default parameters. The reads mapped on exons were counted using featureCounts[59]. Significant DEGs were identified

with $|\log_2 FC| \geq 1$ and FDR < 0.05 using DESeq2[60]. The full dataset was deposited in the European Nucleotide Archive (ENA) under project ID PRJEB65916. To visualize intersections of DEGs in the three different genotypes in UpSet plots the R package Complex UpSet[61] was applied. Heat maps were drawn using the R package Complex Heatmap[62] by using Euclidean distance and cetroid linkage clustering. Gene ontology (GO) enrichment analyses were performed with clusterProfiler[63]. Statistical significance of GO terms was declared when the *P*-value of Benjamini−Hochberg FDR for multiple testing correction was < 0.05. To reduce redundant GO terms and keep one representative term, the method *simplify* was applied with a similarity cutoff > 0.7. Dot plot representations of significant GO terms were obtained using the R package enrichPlot[64].

### Plasmid construction and generation of transgenic lines

All constructs for plant expression were generated by GreenGate modular cloning[65]. The 2151-bp *HYP1* promoter sequence and the 2019-bp of *HYP1* open reading frame (without stop codon) were amplified from genomic DNA of Col-0 (CS60000) and the 1212-bp long *HYP1* coding sequence (without stop codon) amplified from complementary DNA (cDNA). To drive *HYP1* expression in specific cell types, the 2008-bp promoter fragment of *BRN1* (At1g33280), 2086-bp of *SGN1* (At1g61590), 2309-bp of *LPR1* (At1g23010) and 1941-bp of *FRO2* (At1g01580) were amplified from genomic DNA of Col-0 (CS60000). Amplification was performed using the Phusion High-Fidelity DNA Polymerase (New England Biolabs) using primers listed in Supplementary Table 3. Promoters, open reading frames and coding sequences were cloned into the GreenGate entry modules pGGA000 and pGGC000, respectively. Correct integration of the fragments into entry modules was verified by restriction digestion reactions and the cloned sequences were verified by sequencing. To generate transcriptional reporters and translational GFP fusions, individual entry modules were assembled either into the GreenGate binary vector pGGZ001, including a phosphinothricin or hygromycin resistance cassette for antibiotic selection in plants, or into pFASTR A-G for selection of transgenic plants based on the presence of red fluorescence in seeds. Mutated HYP1^M90L, HYP1^H174L and HYP1^H211L/H278L variants were generated by site-directed mutagenesis using the double-primer method[66] and the primers listed in Supplementary Table 2. The final binary vectors were transformed into the *Agrobacterium tumefaciens* strain GV3101 (containing the helper plasmid pSOUP if transformed with pGGZ001) and finally transferred to *A. thaliana* (Col-0; CS60000) plants via the flower dip method[67]. Representative transgenic lines for each construct were selected from 6-8 independent transformants and used for analysis. If not indicated otherwise in the figures, *35S::HYP1* line 5d was used as the representative *HYP1*-overexpressing line in Col-0 background. For oocyte expression, the cDNA sequence of native or mutated *HYP1* variants was amplified with primers containing *Xma*I (forward) or *Xba*I (reverse) restriction sites and cloned into pGEMHE[32].

### Root phenotypical analysis

To determine primary root lengths, roots were scanned using an Epson Expression 12000XL scanner (Seiko Epson) with a resolution of 300 dots per inch after they were clearly separated from one another on the agar plate. Primary root length was quantified with ImageJ software. Measurements of meristem cell length and mature cortical cell length were performed with ZEN 3.4 Blue Edition imaging software (Carl Zeiss Microscopy).

### Oocyte expression and two-electrode voltage clamp recordings

Oocytes were surgically removed from adult female *Xenopus laevis* frogs treated with collagenase (1 mg mL$^{-1}$) for 30 min at room temperature under constant shaking, and washed twice with standard bath solution composed of 96 mM NaCl, 2 mM KCl, 1 mM CaCl$_2$, 1 mM MgCl$_2$, and 5 mM HEPES (pH 7.5). Isolated oocytes were injected with

*HYP1* cRNA at 40 ng per oocyte (or with an equivalent volume of nuclease-free water) using a Drummond "Nanoject" microinjector. Oocytes were incubated at 18 °C in standard solution (supplemented with 0.1 mg mL$^{-1}$ gentamycin) for at least 24 h before voltage-clamp experiments.

Whole-cell membrane currents were measured at room temperature using a home-made two-electrode voltage-clamp amplifier and 0.2−0.4 MΩ glass electrodes with a tip diameter of a few micrometers, filled with 3 M KCl. By convention, negative currents are inward currents mediated by the efflux of anions/electrons (or influx of cations). Membrane currents elicited by exposure to extracellular electron acceptors were generally recorded at a holding potential of −20 mV. In some assays, as indicated in the text and figure legends, the standard bath solution was modified by setting the pH to 5.5 by substituting 10 mM HEPES with an equimolar concentration of MES, or by replacing NaCl, KCl, CaCl$_2$ and MgCl$_2$ with 200 mM MES and 20 mM BTP at pH 5.5 (BTP-MES solution). In experiments with a negative holding voltage of −120 mV, 1 mM lanthanum chloride was supplied to the bath solution, in order to inhibit endogenous background currents activated at these negative voltages. A gravity-driven perfusion system was used for continuous superfusion of oocytes during voltage-clamp recordings and for switching between different bath solutions. Ferric compounds were prepared as stock solutions and diluted appropriately in the bath solution.

For ascorbate injection experiments, ascorbate stock solutions (100 mM) were freshly prepared and adjusted to pH 7.0 with KOH. To raise the cytosolic ascorbate (Asc$_{cyt}$) concentration to 10 mM, oocytes were removed from the voltage-clamp setup after the initial recording series and injected with ascorbate stock solution using the Drummond "Nanoject" microinjector. A first injection of 50 nL stock solution and a second injection of 100 nL resulted in an increase of the initial cytosolic ascorbate (Asc$_{cyt}$) of 10 and 30 mM, respectively, considering a volume of about 500 nL for a mean oocyte diameter of 1 mm. After the injection, the oocyte volume transiently increased but recovered to its original size within minutes. Injected oocytes were allowed to recover for 30 min at room temperature before voltage-clamp measurements were resumed.

### GUS histochemical assay

GUS activity of plants expressing *proHYP1::GUS* and *proCYCB1;1::GUS* was assessed by incubating root samples in a solution containing 20 mg mL$^{-1}$ 5-bromo-4 chloro-3-indolyl-β-D-glucuronic acid (X-gluc) in 100 mM sodium phosphate, 0.5 mM K$_3$Fe(CN)$_6$, 0.5 mM K$_4$Fe(CN)$_6$ and 0.1% (v/v) Triton X-100 at 37 °C in darkness. Samples were then mounted on a clearing solution (chloral hydrate : water : glycerol = 8:3:1) and imaged using a Axio Imager 2 light microscope (Carl-Zeiss).

### Callose staining

Callose deposition was assessed by staining roots with aniline blue as described previously[5]. Briefly, roots were incubated for 1.5 hours in a solution containing 0.1% (w/v) aniline blue (AppliChem) in 100 mM Na-phosphate buffer (pH 7.2). Stained roots were mounted on water and directly visualized with a laser-scanning confocal microscope.

### Confocal microscopy analyses

For protein localization studies, roots of transgenic lines expressing GFP were excised, stained with propidium iodide (10 μg mL$^{-1}$) for 10 minutes, and mounted in water. GFP- and propidium iodide-dependent fluorescence were acquired with a confocal laser-scanning microscope (LSM 780; Zeiss, Germany) equipped with a 20X/0.8 M27 objective. GFP was excited with a 488 nm Argon laser and the emitted light was detected at 505−535 nm while propidium iodide was excited at 561 nm and detected at 600−700 nm. For co-localization of GFP with PM, root segments were stained with 4 μM FM6-64 (Invitrogen) in 50 mM Na-phosphate buffer (pH 7.2) for 15 min at 4 °C, and mounted in 50

mM Na-phosphate buffer (pH 7.2). FM4-64 was excited at 488 nm and detected at 640 nm. Aniline blue-stained callose was excited at 405 nm and emission detected at 415–480 nm. Fe(II)-RhoNox-1 complex was excited at 514 nm and emission detected at 531–703 nm.

Confocal microscopy analyses of *X. laevis* oocytes expressing *HYP1:EGFP*, *HYP1^{H211L/H278L}:EGFP*, and *HYP1^{M90L}:EGFP* were performed using a Leica SP2 microscope equipped with a 40x oil-immersion objective. FM4-64, diluted at a final concentration of 15 μM, was used to stain the PM. Both EGFP and FM4-64 were excited at 488 nm, and emissions were monitored at 505 to 512 nm and 645 to 655 nm, respectively. Confocal images were obtained by scanning the laser scan line axially (z scan) along a 10-μm section (0.5-μm steps) at the bottom of the oocyte.

### Fe staining

The staining of Fe by Perls/DAB was based on a procedure described previously[68], with modifications. In our assays, roots were rinsed one time with ultrapure water for 1 min, and then incubated for 5 min in a freshly prepared Perls staining solution [0.5% (v/v) HCl and 0.5% (w/v) K-ferrocyanide]. Roots were shortly rinsed in water to remove agar adhering to roots, while the more diluted Perls staining solution compared to the original protocol was necessary to prevent meristem disintegration by otherwise to high HCl concentration. Subsequently, roots were incubated in a methanol solution containing 10 mM $NaN_3$ and 0.3% (v/v) $H_2O_2$ for 1 h in the dark at room temperature. After washing three times with 100 mM phosphate buffer (pH 7.4) for 1 min each, roots were finally incubated for 5 min in an intensification solution containing 0.025% (w/v) DAB and 0.005% (v/v) $H_2O_2$ in 100 mM phosphate buffer (pH 7.4). Samples were then mounted on a clearing solution (chloral hydrate : water : glycerol = 8:3:1) and immediately imaged using a Axio Imager 2 light microscope (Carl Zeiss Microscopy, Germany). For Fe localization in root sections, roots stained with Perls/DAB as described above were subjected to a series of aldehyde fixation, dehydration, and resin embedding as described in Supplementary Table 4. Semi-thin sections of 2.5 μm thickness were cut using a Leica UCT microtome (Leica Microsystems, Germany), and mounted on slides in rapid mounting medium Entelan (Sigma-Aldrich, Germany). Sections were recorded with a 40X objective at fixed exposure time using a Zeiss Axio Imager M2 (Carl Zeiss Microscopy, Germany).

For detection of Fe(II) in the apical meristem, root tips were excised and incubated for 10 min in a freshly prepared buffer solution (2 mM $CaSO_4$ and 500 mM MES pH 5.5 adjusted with KOH) containing 2.5 μM RhoNox-1 fluorescent dye[41]. Roots were rinsed and mounted with the same buffer solution without the probe and immediately visualized with a confocal laser-scanning microscope.

### Ferric-chelate and cupric reductase assays

For ferric-chelate reductase activity, the reaction solution consisted of 0.2 mM $CaSO_4$, 5 mM MES pH 5.5 (KOH), 0.2 mM FerroZine [3-(2-pyridyl)-5,6-diphenyl-1,2,4-triazine-4′,4″-disulfonic acid]. The ferric substrates in these assays were either 0.1 mM Fe(III)-EDTA, when HYP1 complementation of *fro2* was tested, or freshly prepared 0.1 mM $FeCl_3$ : L-(−)-Malic acid complexes (1:2), when ferric-chelate reductase was assayed in response to P deficiency. Cupric reductase activity was assessed as described previously[38]. The reaction solution was composed by 0.2 mM $CuSO_4$, 0.6 mM $Na_3$citrate, and 0.4 mM BCDS (Sigma-Aldrich) in ultrapure water. The absorbance of formed Fe(II)-FerroZine or Cu(I)-BCDS complexes was determined at 562 nm or 483 nm, respectively, using a NanoDrop 2000 (NanoDrop). Extinction coefficients were 28.6 $mM^{-1}$ $cm^{-1}$ for the Fe(II)-FerroZine complex and 12.25 $mM^{-1}$ $cm^{-1}$ for the Cu(I)-BCDS complex.

### Shoot chlorophyll analysis

Whole shoot samples were weighted and incubated for 1–2 d at 4 °C in N,N′-dimethylformamide (Roth). The absorbance of the extracts was measured at 647 nm and 664 nm using a NanoDrop 2000 (NanoDrop) and the total chlorophyll concentration determined with the equations described in ref. 69.

### Elemental analysis

Whole root or shoot samples were dried at 65 °C and weighted into polytetrafluoroethylene tubes. Plant material was digested with concentrated $HNO_3$ (67-69%; Bernd Kraft) and pressurized in a high-performance microwave reactor (UltraCLAVE IV, MLS GmbH). Digested samples were diluted with de-ionized water (Milli-Q Reference A+, Merck Millipore). Element analysis was carried out by high-resolution inductively coupled plasma mass spectrometry (HR-ICP-MS) (ELEMENT 2, Thermo Fisher Scientific, Germany).

### Structure modeling

The structure of HYP1 was predicted using AlphaFold[36]. In addition, a homology model of HYP1 was developed before the release of AlphaFold, using the comparative modeling protocol available in the Rosetta software[70]. The following template structures were considered in comparative modeling: the cytochrome domain of cellobiose dehydrogenase enzymes from *Phanerodontia chrysosporium* (PDB: 1D7C), *Myriococcum thermophilum* (PDB: 4QI3) and *Neurospora crassa* (4QI7), and the cytochrome domain of Pyranose dehydrogenase from *Coprinopsis cinerea* (6JT6) as templates for the DOMON domain of HYP1, as well as the *A. thaliana* cytochrome *b*561 (PDB: 4O6Y, 4O79, 4O7G) and human duodenal cytochrome *b* structures (PDB: 5ZLE, 5ZLG) as templates for the cytochrome *b* domain of HYP1. The AlphaFold- and Rosetta-predicted HYP1 structure models were in close agreement with each other (Cα atom RMSDs of 1.5 Å and 2.3 Å over the cytochrome *b* and DOMON domains, respectively). For modeling the interactions with *b*-heme and ascorbate, the AlphaFold-generated model of HYP1 was used because of its estimated higher accuracy.

Three *b*-heme molecules were placed in the HYP1 structure model at positions inferred from the template structures listed above. The positions of the *b*-heme molecules were refined using the FastRelax protocol in the Rosetta software. Distance restraints between the Fe atom in the *b*-heme molecule and the coordinating histidine residues were applied in this step: restraints were set up between H245 and H314 and one *b*-heme in the cytochrome *b*561 domain, between H211 and H278 and the other *b*-heme in the cytochrome *b*561 domain, and between H174 and the *b*-heme in the DOMON domain. The final step was an unrestrained minimization of the whole structure in the Amber14:EHT force field using the MOE software (v. 2022.02 Chemical Computing Group ULC, Montreal, Canada).

### Ligand docking

The molecular binding mode of ascorbate (PubChemID: 54679076) was modeled using ligand docking calculations with the Rosetta software. Ascorbate was docked into HYP1's cytoplasmic facing pocket, which was identified by structural comparison with the ascorbate-bound structure of cytochrome *b*561 (PDB ID: 4O79). The structure analysis indicated that residues K77, K81, and R150 in cytochrome *b*561 coordinate ascorbate through hydrogen bonds. The equivalent residues in HYP1 are K234, R307 and R299. The centroid of these three residues was used as a starting point to place the ascorbate molecule in the HYP1 pocket prior to the docking calculations.

### Statistics and reproducibility

To analyze the significant differences among multiple groups, one-way ANOVA followed by post hoc Tukey's test at $P < 0.05$ was applied. Significant differences between two groups were assessed by a two-sided Student's *t*-test. Statistical tests were performed using SigmaPlot 11.0 software and plots prepared with GraphPad Prism software v.9.3.1 (https://www.graphpad.com/). All experiments were conducted at

least two times with similar results and data from a single representative experiment are presented.

## Reporting summary

Further information on research design is available in the Nature Portfolio Reporting Summary linked to this article.

## Data availability

The raw microarray data generated in this study have been deposited and available in the NCBI Gene Expression Omnibus (GEO) under accession GSE217790, while the raw RNA sequencing data have been deposited and available in the European Nucleotide Archive (ENA) under project ID PRJEB65916. All data generated during this study are included in this published article (and its Supplementary Information File). Source data are provided as a Source Data file. Source data are provided with this paper.

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

## Acknowledgements

We thank Dr. Benjamin D. Gruber for discussions, Jacqueline Fuge, Annett Bieber, and Elis Fraust for excellent technical assistance and Dr. Yudelsy A. Tandron Moya for conducting HR-ICP-MS analyses (Leibniz Institute of Plant Genetics and Crop Plant Research). We thank Prof. Dr. Paolo Trost (University of Bologna Alma Mater) for critically reading the manuscript. We are grateful to Prof. Dr. Steffen Abel (Leibniz Institute of Plant Biochemistry) for sharing the *proSHR::SHR:GFP* reporter line and the *lpr1 lpr2* double mutant. This work was supported by the Deutsche Forschungsgemeinschaft (DFG, German Research Foundation) grant WI1728/24-1 to N.v.W. and "Progetti di Ricerca di Interesse Nazionale" grant 20222CS2B3 to A. C. Costs for open access publication were partially funded by the DFG, grant 491250510.

## Author contributions

R.F.H.G. and N.v.W. conceived, designed, and coordinated the project. R.A.M., C.P., A.G., J.S.S., M.M., and R.F.H.G. performed all experimental work. A.H. analyzed transcriptome data. F.E. and G.K. performed the structural modeling. A.C. supervised the two-electrode voltage clamp experiments. G.K, C.P., A.C., N.v.W., and R.F.H.G. analyzed data; R.F.H.G. wrote the manuscript with inputs of all authors.

## Funding

## Competing interests

The authors declare no competing interests.
