## [Peer Review File · Nature Communications]

Ferric reduction by a CYBDOM protein counteracts increased iron availability in root meristems induced by phosphorus deficiencyREVIEWER COMMENTS

Reviewer #1 (Remarks to the Author):

An interesting story! The article is well organized, well presented and demonstrates the functions of an uncharacterized DOMON-containing protein HYP1 in Arabidopsis. The electron transport activities of this Cytochrome b561-containing protein were well characterized by a series of functional assays in *Xenopus laevis* oocytes, together with a clear analysis of its crystal structure. And, its ferric and cupric reduction activity was conducted using transgenic plants. The HYP1-driven ferric reduction is critical to prevent Fe overaccumulation in the apoplast, and this article provides evidence for the existence of a Fe uptake mechanism in root tips. It is clear and will greatly help researchers in the field to understand the issue. However, I have a few concerns and suggestion as shown below.

1. In FigS2, authors should provide a qRT-PCR analysis to confirm that the HYP1 mRNA levels in this mutant have reduced in both +Pi and -Pi conditions. Or, to detect the full-length HYP1 cDNA levels in this mutant.

2. I have some concerns about the conclusion that HYP1 and CRR are functional redundant. First, their expression patterns seem different. HYP1 is inducible by low-P but CRR is not, indicating that CRR may also function in +P conditions. Second, their subcellular localization is different. According to the manuscript submitted back-to-back, CRR is also localized in ER in addition to plasma membrane localization. Third, the more severe phenotype of *hyp1crr* in -P condition might also be explained as they function in different pathways that both required for root growth in that condition. A reciprocal complementary experiment by introducing pCRR-HYP1 into *crr* or pHYP1-CRR into *hyp1* should help to draw such a conclusion.

3. In line 342-346, "To visualize Fe internalization in the RAM and elongation zone, we then supplied Fe in the growth medium in its ferrous form, the most likely substrate for putative root tip expressed Fe uptake transporters. After 3 days, Perls/DAB-detected Fe was concentrated in small dots that appeared intracellular (Fig. 5c). The presence of these presumably intracellular Perls/DAB-stained pools was not altered in comparable root zones of *hyp1* plants."

I can not follow this result, how can these intracellular Perls/DAB-stained dots represent the Fe internalization?

4. In Fig4g, Perls/DAB staining of Fe in Col-0 in -P seems weaker than that in +P, which is not consistent with other results, like Fig4C.

5. For convenience, the authors should give an explanation of the model in Fig5d.

6. An important argument is that the -P induced primary root inhibition is an artificial result of exposure roots to blue light (Zheng et al., 2019) or dependent on blue light signaling (Gao et al, 2021). As there was no root-covering experiment performed in this study, it should at least discuss this point in this manuscript.

Reviewer #2 (Remarks to the Author):

From the outset, in a transcriptomic approach, the authors focused on the search for the ferric reductase gene which is induced at low Pi. They identified At5g35735, whose ko

mutant has a more pronounced root growth inhibition phenotype than the WT, under -Pi condition. They named this gene hypersensitive to low P 1 (HYP1).

HYP1 is a cytochrome b561 membrane protein, with an extracellular DOMON (dopamine β -monooxygenase N-terminal) domain; the Arabidopsis genome encodes ten DOMON-proteins. So far, no biological role has been assigned to these proteins in plants.

The authors assessed the ko mutant phenotype (in -Pi) of the nine members of this family ; two (air12 and crr) had reduced primary root length in -Pi. HYP1 is expressed in several tissues of the root tip and the HYP1-GFP protein localises in the plasma membrane. Using *Xenopus* oocytes bathed with ferricyanide (an electron acceptor), the authors show that HYP1 mediates an electron transport across the membrane that is exacerbated by injecting ascorbate in oocytes.

The AlphaFold 3D model of HYP1 shows high structural similarities with published experimental structures of cytochrome b561. Combined with Rosetta software tools, the authors positioned ascorbate and hemes in HYP1 structure. Genetic replacement of one heme-coordinating histidine by a leucine is enough to abolish HYP1 ferredoxin activity and function in root.

Genetics, Fe staining and biochemical ferric-reductase activity further link HYP1 activity to the ALMT1-LPR1 pathway. Interestingly, the root tip of the hyp1 ko mutant does not overaccumulate iron when it is supplied as a ferrous form (FeCl₂).

All together, this work add a new step in the Fe-dependent root growth inhibition under low P, attributes a biological role to a DOMON-protein, and open a wider area of research about the role of these DOMON-proteins in plant metal homeostasis and in other functions. This is more than enough to make this work interesting for a wide audience.

I really enjoyed reading this manuscript. The results are well presented and robust (as far as I can judge). I have only minor comments:

Lines 56-57: Add Svistoonoff et al (2007) in the references (they where the first to show the Fe-dependent inhibition of root growth under low-Pi).

Sup Fig. 2b: add « +P » on the left panel and « -P » on the right panel.

Fig. 2i: replace “A311” by “N311” in the scheme.

Fig. 5d: Where is the malate after the reduction of iron by HYP1 from the Fe⁺-malate complex?

Reviewer #3 (Remarks to the Author):

In the submission Maniero et al., the authors identify HYP1 as a gene of interest because of the upregulation of its transcript levels in roots under phosphate deficiency, based on the re-analysis of a previously published microarray-based time course dataset. Because of the similarity of the C-terminal membrane-spanning cytochrome b561-containing domain of the encoded protein to human ferric reductases. The authors continue by testing their hypothesis of a function of HYP1 as a ferric reductase through heterologous expression of a HYP1 cDNA in oocytes and HYP1-mediated phenotypic complementation of the Arabidopsis fro2 mutant lacking the primary root surface ferric chelate reductase. Both approaches

provide support that HYP1 can act to reduce extracellular ferric iron to the ferrous form. Phenotypic analyses of a *hyp1* mutant and an overexpressing line suggest that HYP1 serves to avoid an overly strong inhibition of primary root growth under phosphate deficiency. The authors show that mutants known to continue primary root growth under phosphate deficiency, *lpr1 lpr2* and *almt1*, are epistatic to *hyp1*. For a mechanistic explanation for the primary root elongation phenotypes of *hyp1*, the authors show enhanced Perls/DAB-stained root tips to suggest that under phosphate deficiency, HYP1 counteracts excessive Fe accumulation and by acting to increase intracellular Fe(II) levels accessible to the RhoNox-1 fluorescent dye. In conclusion, the authors present a model in which under phosphate deficiency, HYP1 acts in the root apical meristem by reducing extracellular Fe(III) to Fe²⁺, which is subsequently taken up into cells through an unknown transporter. The authors propose that this results in a decrease in apoplastic Fe(III) levels and thus less apoplastic redox cycling, attenuation of cell wall stiffening and callose deposition, thus promoting root growth. While the oocyte work and the *fro2* complementation provide elegant evidence for possible activities for HYP1, the data on root Fe accumulation and in planta function under phosphate deficiency are less convincing and require revision.

Major comments

1. In their present form, the Fe localization data provided in the manuscript (Fig. 4c, g; Fig. 5a, c; Fig. S11b), in conjunction with the presentation of the underlying methodology and associated information, do not sufficiently support the conclusions and model.

(a) The authors state that their enhanced Perls/DAB staining procedure visualizes exclusively apoplastic Fe(III) (line 301, compare detection of Fe in endodermal vacuoles in publication cited for method, Roschztardt et al., 2009). This statement would require explaining and discussing the development and details of the methodology used in this manuscript, as well as experimental evidence. The authors must provide evidence for the localization of the Fe pools detected as well as additional data that are more specific and more quantitative, or alternatively confirm their results with a different technique such as nanoSIMS or related (see more below).

(b) The authors also show decreased levels of RhoNox-1 staining for Fe(II) root tips of the *hyp1* mutant under phosphate deficiency (Fig. S11b). However, enhanced cell wall stiffening and callose deposition, which they also show or propose to occur in *hyp1* (Fig. 4d), are expected to decrease the permeability of the root tip for a dye like RhoNox-1. Moreover, the smaller meristem size of *hyp1* (Fig. S4e-f) is expected to contribute to decreased RhoNox-1 staining intensity in the root tip since it seems that staining is generally the most intense in meristematic cells, for example in the wild type. Alternatively, the iron probe visualizing Fe may also merely reflect a more oxidized redox state in this region of the root tip of the *hyp1* mutant.

2. While the reviewer accepts the evidence supporting the idea that HYP1 is capable of transferring electrons from cytosolic ascorbate to an extracellular electron acceptor such as Fe(III) (Fig. 2fgkl, Fig. 3a-e), these data do not yet demonstrate that HYP1 does act as a ferric reductase under phosphate deficiency in planta. The N-terminal extracellular DOMON (dopamine beta-monooxygenase) domain could also interact with other extracellular electron acceptors in planta under phosphate deficiency conditions, for example the “ROS species” or downstream target compounds oxidized in the pathway leading to cell wall stiffening or callose deposition (Fig. 5d). Compared to the wild type there seems to be no significant increase in ferric reductase activity in 35S-HYP1 under -P and it seems – if data were corrected for multiple comparisons – that the decrease in *hyp1* would not be statistically significant (Fig. S11a, and lines 328-331). In relative terms, the decrease seems small –

maybe by 20%. The reviewer feels that the support for this aspect of the authors' model is too weak. The reviewer also sees partial contradictions compared to the data provided in Fig. 2c (done in Fe deficiency, however) and Fig. 4h (35S::HYP1) and wonders about the composition of media in the respective experiments.

3. Previous publications have shown that under phosphate deficiency, primary root growth of wild-type *Arabidopsis* plants ceases through enhanced extracellular Fe accumulation specifically in the stem cell niche and at the quiescent center of the root apical meristem, ROS cycling and callose deposition in the root stem cell niche. According to the data presented in this manuscript, there is only little callose deposition under phosphate deficiency in the wild type (and much more broadly and differently localized than previously reported by Müller et al. (2015), but callose deposition in a somewhat smaller region that is more likely to include the stem cell niche in the *hyp1* mutant. This suggests to the reviewer that these phenotypes depend a lot on the specific growth conditions, for example phosphate and Fe(III) supply and speciation in the growth medium. In fact, in this manuscript, the authors largely used a highly unusual Fe supply (highly elevated total and in non-chelated form). While it could potentially be argued that this reflects the conditions in natural soils more appropriately than other compositions, the authors do not provide any evidence for this view and do not even argue their choice to rationalize their conditions in that manner. Instead, they rationalize the growth system "to avoid confounding effects" (line 124-128). In addition, based on the information given in materials and methods, different types of experiments were carried out with different Fe and P supplies. For transcriptomics and growth under Fe deficiency, for example, EDTA was used as the Fe(III) chelator. In general, there are potential issues with piecing together results from different growth system to generate datasets that are later compared directly or integrated into one working model. In addition, the reviewer had difficulties inferring from the information given in the methods how each of the experiments was conducted (e.g. growth medium for Fig. S11a and b). The authors should refer to the specific growth conditions more explicitly for all data shown. They should give more background information on differences they observed between previously conditions that were previously commonly employed and the conditions they decided to use in the end (no Fe chelator and shielded from the light). The authors should refer more specifically to the stem cell niche and quiescent center when showing or discussing Fe accumulation and callose deposition, or discuss explicitly their view on the localization of root processes under low Pi.

4. The reviewer believes that also in the ferric reduction assays carried out using roots, authors should always indicate which Fe(III) complex was used as a substrate, in the figure legend or in the panel. This seems to differ, e.g. in Fig 4h from Fig. 3c. The consequence is identical apparent ferric reductase activities in Fe-deficient roots (Fig. 3c) and in roots of plants grown in control Fe conditions (Fig. 4h), which seems counter-intuitive.

5. The authors refer to the issue of the inhibition of primary root elongation under low P known in wild-type *Arabidopsis*. In the context of the present work: Is it seen as unfortunate Fe toxicity or an adjustment of root architecture that overall serves to improve plant fitness under phosphate deficiency? This should be discussed and clarified, and the role of composition of agarose-based media should be addressed (see also 3.).

6. Continuing on 1(a). The enhanced Perls/DAB method used was taken from a publication in which embryos dissected from seeds or fixed sections of mature seeds were stained (Roschztardt et al., 2009). The authors summarize the method briefly, but this suggests that they did not use the same method. Incubation times are different, and the authors here

do not mention CoCl_2 , which was used by Roschztardt et al. (2009). Composition of solutions and all incubation times used for the Perls/DAB staining are absolutely critical and must be presented here clearly and in completion.

7. Continuing on 1(a). Unlike roots, staining for Fe of embryos dissected from seeds (Roschztardt et al., 2009) is not impacted by Fe diffusion from the medium into the apoplastic space and binding to or precipitating on the cell wall. In most experiments published, there are enormous amounts of Fe detectable by Perls stain in the root apoplast or evidently present based on the magnitude of Fe concentrations detected in bulk roots. Finding this alone is not sufficient to conclude on Fe-induced toxicity. In addition, for such staining procedures, whether and to which degree apoplastic Fe has been desorbed is critical information. The authors state that roots were rinsed in EDTA. This is insufficiently precise. Rinsing in EDTA will begin to remove some, or with time possibly most or even all, of the apoplastic Fe. The period for which such a rinse was conducted is absolutely critical, such rinses must be stringently controlled, and this information must be given in full and rationalized, in the context of the topic of this manuscript.

8. Continuing on 1(a). Line 336: "More Perls/DAB stainable ferric Fe". Perls/DAB is not specific for ferric iron. In fact, the methodological staining procedure includes conditions provoking redox cycling of iron for stain enhancement (Roschztardt et al., 2009). So in the eyes of the reviewer, it is not justified to conclude on the oxidation state of Fe *in vivo*.

9. Continuing on 1(a): Lines 345-347: The authors describe here that the dot-like structures they observe after Fe(II) feeding for 6 d are observed equally in root tips of the wild type and the *hyp1* mutant. The following conclusion in lines 348 to 350, saying that altogether there is evidence for a HYP1-dependent Fe(III) to Fe(II) reduction in root tips critical for root elongation in $-P$ does not hold up in my opinion because it is not conclusively shown that when Fe(III) is provided externally, less Fe (in total, there could be re-oxidation inside cells) ends up inside those cells in the root tips of *hyp1*, when compared to the wild type. Perls/DAB indicates that there is more Fe altogether after a longer exposure time of 6 d (Fig. 4c) or 3d (4g), but treatment of Fe(III) (similar to Fig 4g) and Fe(II) (similar to Fig. 5c) should be shown side-by-side and conducted identically in identically pre-cultivated plants.

10. Continuing on 1(a). Lines 379-380: "raised initial evidence for the existence of a Fe acquisition mechanism in this root zone Fig. 5a-c." 5a: The difference in Fe concentrations between genotypes detected here in the shoot may result from indirect effects or from HYP1 presence elsewhere in the root or even shoot. 5b: contains no evidence for HYP1 contribution to Fe acquisition. 5c also doesn't in another sense, but shows that Fe must get in somehow – this could occur also non-specifically via systems mediating influx of other divalent cations. The Fe-containing dots appear too large to constitute ferritin proteins – do they correspond to plastids (then please adjust the corresponding text, line 347). To the reviewer the composition of the medium enriched in Fe(II) used for Fig. 5c is unclear.

11. Continuing on 1(a) and 9. In relation to the working model, a strategy by plant cells to prevent apoplastic Fe(III) toxicity by first generating Fe^{2+} in the apoplast and then taking Fe^{2+} up into the cytoplasm seems risky and somewhat counter-intuitive. The authors should explain how they envisage this to function.

12. L. 76/77: "protein that is induced...": Please demonstrate at protein level, e.g. by immunoblot or analysis of sufficient CLSM frames for quantification and statistical analysis under identical settings and conditions, or alter that statement. Fluorescent images alone

(Fig. 2b, Fig. S5) are not sufficiently (semi-)quantitative. Increased levels of HYP1 (or HYP1-GFP) protein in root tips of Pi-deficient plants. Else the statement should be modified.

13. Fig. S5 suggests the presence of HYP1 protein and a substantial upregulation of its protein levels under phosphate deficiency, although this is preliminary, also in the root hair zone. This has the potential of significantly affecting the interpretation of the results presented here. Presently, the interpretation root growth phenotypes and alterations in root and whole-plant Fe levels and homeostasis is based exclusively on changes observed in the meristem, but the phenotype is the integrated consequence of all locations in the plant where HYP1 is active. The authors should experimentally address in a quantitative manner also what happens in the root hair zone and incorporate this information in their interpretation and working model. This may, for example, contribute to the observation of an apparent contribution of HYP1 – directly or indirectly – to shoot Fe levels under low P (Fig. 5b). The authors should also comment on the predominant signal in the outer cell layer – is this autofluorescence or HYP1-GFP (a non-transformed control, for example, would answer this question)? If it is HYP1-GFP (seems likely), how does this affect the interpretation of results, given that the proposed function of apoplastic Fe accumulation under phosphate deficiency – to the understanding of the reviewer – is highly localized to the stem cell niche of the root tip according to Muller et al. (2015)? Please make modifications in the text to adjust the description of HYP1 localization (lines 182-188), if needed, and to discuss this comment.

14. Further relating to the preceding point: In line 338 to 340 the authors explain how the wild type may end up containing higher Fe concentrations in shoots under low-P conditions than hyp1 mutants. The explanation is provided that HYP1 decreases apoplastic Fe sequestration in the wild type so that more ends up inside the root cells (in RAM, see Fig 11b). However, it is not clear whether the Fe in these specific cells would be mobile for movement from the root to the shoot. In fact, it is a little less likely because it is thought that only Fe taken up in the root hair zone is mobile for transport into the mature xylem vessels and up into the shoot. So do the authors envisage HYP1 (also) to act further up in this zone?

15. Line 301: Perls/DAB stainable apoplastic Fe... - apoplastic localization must be demonstrated. In fact, this contradicts with line 344, where the dots observed in Fig. 5c are interpreted as intracellularly localized. This means that the stain can localize both extracellular and intracellular Fe, and – without further modification or optimization of the method and associated validation - based on the microscopic images it cannot be concluded where the Fe is localized or which oxidation state it was in the living plant. This relates to some of the earlier comments.

16. In a number of instances, single root tips are shown for comparison to argue quantitative differences between them. A number of frames of root tips following identical treatment and processing and imaging with identical settings should be provided in the supplement, as well as image-based quantification with statistical analysis (e.g. Fig. 4c, d, g; Fig. 5a, c; Fig. S5; Fig. S10; Fig. S11b; possibly Fig. S4e, f).

17. Suppl. Fig. 2: It would be important to know whether there are any remaining full-length transcripts, and what the levels are of truncated transcript, in the hyp1 T-DNA insertion line. Also please indicate position of insertion shown in Fig. S2a relative to coding sequence, to specify which part of the HYP1 protein might be left in the hyp1 mutant.

18. In this same context, it would be better to have reproduction of the phenotype under –

phosphate in a second, independent T-DNA insertion line. Complementation lines in the *hyp1* mutant background, in conjunction with mutant HYP1 variants fused to GFP (Fig. 2m) in the same background, compensate for the lack of an independent mutant to some extent, but not entirely. Could the authors please demonstrate the presence of the protein under phosphate deficiency in the H211L/H278L lines (e.g. by confocal imaging of the GFP fluorescence)?

Comments on both manuscripts jointly

19. The reviewer wonders whether by supplying ferricyanide (Fig. 4A and B) as a substrate to CRR proteins, a previously reported general action of ferricyanide in protein-bound heme oxidation comes to bear, and whether the observed ferrereductase activities in this manuscript are thus an observation that does not correspond to the biological function of CRR. The reviewer wonders whether while CRR can donate an electron to Fe^{3+} under appropriately designed/alterd conditions, this is in fact a negligible process in planta under the usual range of physiological conditions including phosphate deficiency. The authors of both manuscripts should contemplate (and possibly provide reasons against) the following model: HYP1 or CRR might donate an electron to directly quench extracellular ROS and thus Fenton chemistry in the apoplast of the stem cell niche. There needs to be a fine balance though, because in principle and under "permissive" conditions, electron donation to apoplastic molecules via CRR or HYP1 (e.g. in overexpressing shoots) can alternatively enhance ROS production and thus promote Fenton chemistry. Next, evidently, in $-\text{Pi}$ conditions, LPR1/2 alone has sufficient Fe^{2+} substrate available to do its job, even without CRR (or without HYP1), and possibly without both of them (?), and thus LPR1/2 alone is able to account for the Fe-dependence of primary root growth inhibition phenotypes; the need for something additional (CRR/HYP1) to generate Fe^{2+} (in this case to moderate root growth inhibition) is not evident. Quantitative depletion of Fe(III) in the root apoplast through reduction to Fe^{2+} seems like a goal that is hard to achieve in a location that is permeable to the soil solution containing plenty of Fe(III) .

20. How do the authors explain the root growth difference in *crr1 almt1* (barely longer/meristem size not larger than *crr*, only about 50% root length of *almt1*) compared to *hyp1 almt1* (like *almt1*) in the accompanying manuscript under $-\text{Pi}$?

21. The authors should provide results from the root growth assay conducted under phosphate-sufficient and -deficient conditions also for the *crr hyp1* double mutant, in addition to the single mutants and the wild type. Including this in both manuscripts will contribute to the comparability of both manuscripts because the media and cultivation systems were different.

22. The reviewer is struggling with the impression that plenty of Fe (see Fig. 2G, likely mostly Fe(III)) seems to be present near QC/SCN in $+\text{P}$ conditions, but no stalling of primary root elongation occurs. While the presence of Fe is clearly necessary for this under $-\text{Pi}$ conditions, the reviewer wonders whether much of what is observed several days later is merely a secondary and broader consequence of highly localized events that occurred within a much shorter period of time after the initiation of Pi deficiency.

23. The authors should consider in their models that the CYBDOM proteins may act somewhat independently in different parts of the plant – so in the opinion of the reviewer, if they generate an extracellular Fe^{2+} pool that is then taken up to contribute to shoot Fe levels, this Fe may not originate from the Fe^{2+} pool localized where stalling of root growth

occurs under $-P_i$ conditions. HYP1 appears to be present in a variety of cell types and even under $+P_i$ conditions in a subset of these. This complicates pinning down the location of where the protein acts to contribute to foster primary root elongation under $-P_i$.

Minor comments

24. The reviewer would like to see more discussion on the localization of HYP1 protein in relation to the localization of Fe (e.g. line 301). There appears to be a lot of Fe in the root hair zone in Fig. 4g (see also comments 13 and 14).

25. The authors are asked to provide the information of how much phosphate and iron (μM in the final medium) the agar and the sugar components added to their media.

26. Line 307: the ferric reduction activity of HYP1: the activity was not measured using the purified protein, but in planta and in oocytes. The conclusion here should be adjusted to ensure its accuracy.

27. L.298-299: “apoplast” ... “as shown previously”. The publication referred to here shows a more highly localized accumulation of Fe in phosphate deficiency than the broad accumulation shown in Fig. 4c, g; 5a). While Muller et al. (2015) examined in very much detail the localization of Fe also in the apoplast in very specific localizations, this demonstration is not conducted here.

28. The *hyp1* phenotype under $-P$ is clear, but much less dramatic than the end of the introduction suggests. Please adjust the wording of the text.

29. Line 83/84: “reduces ferric and cupric ions”: Reviewer feels that the protein “can reduce...”. This is more consistent with the conditions under which this was observed in the present manuscript. Line 263: HYP1 mediates ferric and cupric reduction (“can mediate”). Lines 274, 290: “can function as”.

30. Lines 84/85: critical to prevent Fe overaccumulation in the apoplast – not addressed directly in this manuscript

31. In Fig. 1d, the phenotype does not appear to be very strong here (and it is hard to relate the photo to the measurements in 1e), and it would be critical for the reader to know where the root tip was positioned in the moment the low-P treatment was initiated. Could the authors please mark this in the plates and also add a scale bar in 1d.

32. In Fig. 1f, overexpressor roots seem longer than wild type, but not in 1g – why? Can the authors clarify this in the figure or manuscript text.

33. In Fig. 1h, *hyp1* hardly seems to differ from Col- under $-P$ until 3 d. This contradicts Fig. 1d-g.

34. Line 304: “and decreased cell-cell communication” – replace by “and decreased cell-to-cell mobility of proteins (add where this was observed because it was a highly localized observation, and it remains unknown what is happening elsewhere, e.g. across the meristem).

35. Fig. 1i: *hyp1*, but none of the other lines, have a large number of root hairs initiating already very close to the root tip from 2 d after transfer into low-phosphate on. This

observation should be referred to in the text. What does it mean (also in relation to Fig. 3k)?

36. Fig. 3k: The image for 35S::HYP1 shows a lot of root hairs already very close to the root tip in seedlings grown in high Cu. This observation should be referred to in the text. What does it mean (also in relation to Fig. 1i)?

37. Line 386-7: The reviewer would generally expect that intracellular Fe-mediated ROS generation is a much bigger risk than in the apoplast

38. 396-7: CYBDOM-dependent ROS formation would be predicted to have the opposite effect, i.e. less root growth inhibition in the mutant. Add one more sentence to explain how this is meant, or else maybe delete this?

39. In some figure legends, n is not formatted in italics.

40. Line 44: “very low or only hardly available” – both mean almost the same – please rephrase.

41. Line 80/81: “ comprised of” rephrase

42. Line 100: “downregulated by P deficiency” – please rephrase, e.g. “downregulation in response to...”. There are a number of similar instances in the manuscript.

43. Line 102: The downregulation of IRT1 transcript levels under P deficiency was reported before and extensively discussed in the literature; authors should mention this and cite the publication that was the first to report this.

44. Line 148: ... “showed more severe” is unclear – rephrase sentence?

45. Fig. 2a: add higher-resolution images of root tips.

46. Fig. 1: In (c), the label should say “NADPH ferric reductases”, not “NADH...”

47. Line 151: replace “play” by “have”

48. Fig. 2 and Legend Fig. 2, FeCN: please replace by correct chemical formula;

49. Amino acids mutated in Fig. 2k, l should be marked in h or j, clarify in h or j the position in the protein of what is shown in i. Legend must define arrows in c, f, k, as well as in each case both the round shapes symbolizing a metal ion (which one?) and all cofactors in h, j and n.

50. Fig. 3 and Legend Fig. 3. h: Adjust statistics to correct for multiple comparisons of means.

51. Fig. 4g add size bars.

52. Fig. S3a: specify technique in legend – assume microarray data? An independent confirmation of HYP1 transcript levels via RT-qPCR would be nice.

53. Suppl. Fig. S4a: define arrow positions in legend text below.
54. Suppl. Fig. S6: Replace FeCN (2x) by correct chemical formula, insert spaces as appropriate in legend text and figure. Please specify concentration of ascorbate and GSH used in panel d (e.g. in legend text below or refer to panel c) – 10 mM?
55. Suppl. Fig. S7: FeCN as above, show SD instead of s.e.m., “cyt” is cut off at the bottom in panel b. Legend d: “Acscyt + 10 mM)” very hard to understand, best to add “Asc” again. Can the authors please explain more precisely the meaning of the two vertical arrows in a, b and e, respectively.
56. Suppl. Fig. 8: This is a nice figure showing that the H211L/H278L mutant variant is produced and plasma membrane-localized in oocytes, as well as why this mutant was made and not others.
57. Suppl. Fig. S9: Please explain precisely the significance of the two vertical arrows in a. Please show s.d. instead of s.e.m. in b.
58. Suppl. Fig. 10: This is a nice figure because it shows the details around the quiescent center.
59. Suppl. Fig 11: Statistics of a need modification. Does RhoNox-1 move into cells quantitatively – could a useful control be introduced? Please specify in which form, at which concentration and how long Fe(II) was fed to roots and how its oxidation was prevented.
60. Please refer to accompanying manuscript in the manuscript text (just like the accompanying manuscript does).

Reviewer #4 (Remarks to the Author)

Please see the attachment.

The manuscript by Maniero et al. identified a CYBDOM protein named as HYP1 in Arabidopsis, and they found that HYP1 is required for the low P responses of root growth. They found that HYP1 functions as a ferric reductase and mediates electron transfer through plasma membrane in root cells. The story is quite interesting, and this research provides insights into the mechanisms of low P responses. I feel that the identification of this novel gene will facilitate the research in this field. Please see the following for my comments, and I wish the comments are helpful for the authors to improve the manuscript.

1. The electrophysiological data show the larger currents recorded at -20 mV in the oocytes injected with HYP1 cRNA relative to the oocytes injected with water, and the currents become bigger when more ascorbate was injected into the oocytes. My first concern is you added FeCN in the bath solution, and injected ascorbate into the oocytes. That means ascorbate and FeCN have no chance to mix together and bind to each other. Then how they form a complete electron transfer chain, and how electron transferring happens? If a small holding potential is enough to drive electron efflux, then how to explain the ascorbate-specificity of the currents. Supp Figure 6 showed another donor, GSH, is not capable of generating currents. What if you add FeCN and ascorbate together in the bath solution, could you record any currents, for example outward cation currents/inward anion currents? Another concern is the possibility of Ca²⁺-activated anion currents. The bath solution contains 1 mM CaCl₂, and the currents were recorded at -20 mV. It is known that the influx of Ca²⁺ activates large anion efflux in oocytes when the holding potential applied to the plasma membrane is negative. In my experience, about 80% of the whole-oocyte currents are resulted from the efflux of anion in case Ca²⁺ influx really happens, and only a little portion of the currents is derived from Ca²⁺ and others. Some people add some DIDS, an anion channel blocker in the bath solution, to inhibit the anion efflux, then they claim the currents they recorded are Ca²⁺ currents. That is not correct because DIDS is a quite weak blocker and cannot abolish Ca²⁺-activated anion channel currents. In this research it is possible that the currents could be a mixture of electron efflux from ascorbate to FeCN and some anion efflux activated by Ca²⁺. So I think the authors need to exclude the possibility of anion efflux activated by Ca²⁺ influx. To address this, I guess the authors could try the followings. One is to remove Ca²⁺ at all from the bath solution, and to replace all chloride in both bath and pipette solution with a membrane-impermeable anion. You could use salt bridge containing Cl⁻ to form a closed circuit. I think gluconate could be a good choice. If you record similar currents in the oocytes expressing HYP1, and the currents are dependent on ascorbate and FeCN, then the data will be more convincing. Alternatively, the authors also could use a different cell system, such as HEK293. This is because Ca²⁺ activates anion efflux only in oocytes, and this does not happen in many other mammalian cell lines. I guess you need to do patch clamping in these other cell lines rather than voltage clamping.

2. I am also curious what the concentration of ascorbate in root cells, whether the level of ascorbate in root cells is high enough to generate large current, and whether Pi deficiency induces the accumulation of ascorbate in root cells.

REVIEWER COMMENTS

We thank the reviewers for appreciating our work and for providing constructive criticisms and suggestions. The manuscript has been substantially revised to address all points raised by the reviewers. We did this by conducting new experiments, adding more detailed description of the employed methods, and by clarifying several issues in the text. With these revisions, we think our manuscript has greatly improved.

Reviewer #1 (Remarks to the Author)

An interesting story! The article is well organized, well presented and demonstrates the functions of an uncharacterized DOMON-containing protein HYP1 in Arabidopsis. The electron transport activities of this Cytochrome b561-containing protein were well characterized by a series of functional assays in *Xenopus laevis* oocytes, together with a clear analysis of its crystal structure. And, its ferric and cupric reduction activity was conducted using transgenic plants. The HYP1-driven ferric reduction is critical to prevent Fe overaccumulation in the apoplast, and this article provides evidence for the existence of a Fe uptake mechanism in root tips. It is clear and will greatly help researchers in the field to understand the issue. However, I have a few concerns and suggestion as shown below.

Response: We thank the reviewer for the encouraging comments.

1. In FigS2, authors should provide a qRT-PCR analysis to confirm that the HYP1 mRNA levels in this mutant have reduced in both +Pi and -Pi conditions. Or, to detect the full-length HYP1 cDNA levels in this mutant.

Response: We now show that *HYP1* expression is strongly decreased in the T-DNA insertion mutant, irrespective of the P condition. The result is presented in new Suppl. Fig. 2a and mentioned in lines 142-143.

2. I have some concerns about the conclusion that HYP1 and CRR are functional redundant. First, their expression patterns seem different. HYP1 is inducible by low-P but CRR is not, indicating that CRR may also function in +P conditions. Second, their subcellular localization is different. According to the manuscript submitted back-to-back, CRR is also localized in ER in addition to plasma membrane localization. Third, the more severe phenotype of *hyp1crr* in -P condition might also be explained as they function in different pathways that both required for root growth in that condition. A reciprocal complementary experiment by introducing pCRR-HYP1 into *crr* or pHYP1-CRR into *hyp1* should help to draw such a conclusion.

Response: The reviewer is right that the phenotype of *hyp1 crr* is not sufficient to firmly conclude that HYP1 and CRR are functional redundant, especially considering the additional localization of CRR in the ER. To perform the experiment suggested by the reviewer we would need to generate new transgenic lines, which would take several months. As concluding whether or not HYP1 and CRR act redundantly is not a main message for our story, because the *hyp1* single mutant has already a strong conditional phenotype, we toned down our conclusion, restricting it to what is shown by our data. Thus, we now write (lines 166-167) "...suggesting that the activities of these two CYBDOM proteins act in an additive way to sustain root growth under low-P conditions."

3. In line 342-346, "To visualize Fe internalization in the RAM and elongation zone, we then supplied Fe in the growth medium in its ferrous form, the most likely substrate for putative root tip expressed Fe uptake transporters. After 3 days, Perls/DAB-detected Fe was concentrated in small dots that appeared intracellular (Fig. 5c). The presence of these presumably intracellular Perls/DAB-stained pools was not altered in comparable root zones of *hyp1* plants." I can not follow this result, how can these intracellular Perls/DAB-stained dots represent the Fe internalization?

Response: Previously, it was demonstrated that Perls/DAB-stained dots in cell of the root apical meristem represent Fe associated to ferritins accumulating in plastids, and that storage of Fe in ferritins protects meristems against Fe-induced oxidative stress when roots are exposed to high external Fe (Reyt et al., 2015 Mol Plant DOI: 10.1016/j.molp.2014.11.014). However, the reviewer is right that this is only an indirect evidence of Fe internalization, as it does not rule out the possibility that cytoplasmic or vacuolar Fe is simply remobilized to plastids when root tips were exposed to Fe(II) in our experiments. As described in more details in our response to Reviewer #3, it is still necessary to establish and validate a methodology to directly assess Fe uptake specifically in the root meristem, which is beyond the scope of the present study. Therefore, we removed this particular result from the manuscript and now only hypothesize the existence of an Fe acquisition mechanism in the root meristem (hypothetical model Fig. 6l and discussion in lines 497-522), which could rely on the ferric reductase activity of HYP1.

4. In Fig 4g, Perls/DAB staining of Fe in Col-0 in -P seems weaker than that in +P, which is not consistent with other results, like Fig 4c.

Response: Thank you for pointing this out. Following the suggestion of Reviewer #3, we repeated the Perls/DAB stainings and did not use EDTA to rinse roots prior to the staining. As can be seen in Figs. 5c (former Fig. 4c), and Fig. 6c (former 5g), stainable Fe pools are indeed stronger in +P roots of Col-0 plants. This is in agreement with previous studies (e.g., Müller et al., 2015; Dong et al., 2017 DOI: 10.1016/j.molp.2016.11.001). These pools represent precipitated Fe-phosphates (Müller et al., 2015) which did not trigger callose deposition, and did not differ significantly between Col-0 and *hyp1* (Fig. 5c-f). Under low P, these pools can be mobilized by malate and putatively taken up by meristematic cells, in a HYP1-dependent manner.

5. For convenience, the authors should give an explanation of the model in Fig 5d.

Response: The hypothetical model is briefly explained in the legend of Fig. 6l. In the revised manuscript, we also discuss it further in lines 479-522.

6. An important argument is that the -P induced primary root inhibition is an artificial result of exposure roots to blue light (Zheng et al., 2019) or dependent on blue light signaling (Gao et al., 2021). As there was no root-covering experiment performed in this study, it should at least discuss this point in this manuscript.

Response: This is an important point. Except for the original microarray experiment and the phenotyping shown in Fig. 1d, all experiments shown in the manuscript were performed using a setup in which roots were shielded from direct light exposure. As we already showed in Suppl. Fig. 1,

covering the agar plates strongly prevented photoreduction of Fe(III). We also performed an RNA-seq using our modified root-shaded cultivation setup (new Fig. 5h-j, Suppl, Fig. 12). Following the recommendations of Reviewer #3, we now also explain our growth conditions in much more details (lines 134-141 and in Methods lines 563-571).

Reviewer #2 (Remarks to the Author)

From the outset, in a transcriptomic approach, the authors focused on the search for the ferric reductase gene which is induced at low Pi. They identified At5g35735, whose ko mutant has a more pronounced root growth inhibition phenotype than the WT, under -Pi condition. They named this gene hypersensitive to low P 1 (HYP1).

HYP1 is a cytochrome b561 membrane protein, with an extracellular DOMON (dopamine β -monooxygenase N-terminal) domain; the Arabidopsis genome encodes ten DOMON-proteins. So far, no biological role has been assigned to these proteins in plants.

The authors assessed the ko mutant phenotype (in -Pi) of the nine members of this family ; two (air12 and crr) had reduced primary root length in -Pi. HYP1 is expressed in several tissues of the root tip and the HYP1-GFP protein localises in the plasma membrane. Using *Xenopus* oocytes bathed with ferricyanide (an electron acceptor), the authors show that HYP1 mediates an electron transport across the membrane that is exacerbated by injecting ascorbate in oocytes.

The AlphaFold 3D model of HYP1 shows high structural similarities with published experimental structures of cytochrome b561. Combined with Rosetta software tools, the authors positioned ascorbate and hemes in HYP1 structure. Genetic replacement of one heme-coordinating histidine by a leucine is enough to abolish HYP1 ferredoxin activity and function in root.

Genetics, Fe staining and biochemical ferric-reductase activity further link HYP1 activity to the ALMT1-LPR1 pathway. Interestingly, the root tip of the hyp1 ko mutant does not overaccumulate iron when it is supplied as a ferrous form (FeCl₂).

All together, this work add a new step in the Fe-dependent root growth inhibition under low P, attributes a biological role to a DOMON-protein, and open a wider area of research about the role of these DOMON-proteins in plant metal homeostasis and in other functions. This is more than enough to make this work interesting for a wide audience.

I really enjoyed reading this manuscript. The results are well presented and robust (as far as I can judge).

Response: We thank the reviewer for the encouraging comments.

I have only minor comments:

Lines 56-57: Add Svistoonoff et al (2007) in the references (they were the first to show the Fe-dependent inhibition of root growth under low-Pi).

Response: Thank you for this suggestion. The reference was added in line 62.

Sup Fig. 2b: add « +P » on the left panel and « -P » on the right panel.

Response: We have now added the labels in Suppl. Fig 2c (former Suppl. Fig. 2b).

Fig. 2i: replace “A311” by “N311” in the scheme.

Response: Thank you for spotting this mistake. Corrected!

Fig. 5d: Where is the malate after the reduction of iron by HYP1 from the Fe⁺-malate complex?

Response: Thank you for this interesting question. If not degraded (e.g., by soil rhizosphere microbes) or (taken up,) it could form new complexes with Fe(III) or other ligands [e.g., Al(III)] present in the root apoplast. Although at this stage we cannot be sure about the fate of malate after Fe(III) reduction, we have now represented a free malate molecule after Fe(III) reduction in our revised model (Fig. 6l).

Reviewer #3 (Remarks to the Author)

In the submission Maniero et al., the authors identify HYP1 as a gene of interest because of the upregulation of its transcript levels in roots under phosphate deficiency, based on the re-analysis of a previously published microarray-based time course dataset. Because of the similarity of the C-terminal membrane-spanning cytochrome b561-containing domain of the encoded protein to human ferric reductases. The authors continue by testing their hypothesis of a function of HYP1 as a ferric reductase through heterologous expression of a HYP1 cDNA in oocytes and HYP1-mediated phenotypic complementation of the Arabidopsis *fro2* mutant lacking the primary root surface ferric chelate reductase. Both approaches provide support that HYP1 can act to reduce extracellular ferric iron to the ferrous form. Phenotypic analyses of a *hyp1* mutant and an overexpressing line suggest that HYP1 serves to avoid an overly strong inhibition of primary root growth under phosphate deficiency. The authors show that mutants known to continue primary root growth under phosphate deficiency, *lpr1 lpr2* and *almt1*, are epistatic to *hyp1*. For a mechanistic explanation for the primary root elongation phenotypes of *hyp1*, the authors show enhanced Perls/DAB-stained root tips to suggest that under phosphate deficiency, HYP1 counteracts excessive Fe accumulation and by acting to increase intracellular Fe(II) levels accessible to the RhoNox-1 fluorescent dye. In conclusion, the authors present a model in which under phosphate deficiency, HYP1 acts in the root apical meristem by reducing extracellular Fe(III) to Fe²⁺, which is subsequently taken up into cells through an unknown transporter. The authors propose that this results in a decrease in apoplastic Fe(III) levels and thus less apoplastic redox cycling, attenuation of cell wall stiffening and callose deposition, thus promoting root growth. While the oocyte work and the *fro2* complementation provide elegant evidence for possible activities for HYP1, the data on root Fe accumulation and in planta function under phosphate deficiency are less convincing and require revision.

Response: We thank the reviewer for the encouraging comments.

Major comments

1. In their present form, the Fe localization data provided in the manuscript (Fig. 4c, g; Fig. 5a, c; Fig. S11b), in conjunction with the presentation of the underlying methodology and associated information, do not sufficiently support the conclusions and model.

(a) The authors state that their enhanced Perls/DAB staining procedure visualizes exclusively apoplastic Fe(III) (line 301, compare detection of Fe in endodermal vacuoles in publication cited for method, Roschztardt et al., 2009). This statement would require explaining and discussing the development and details of the methodology used in this manuscript, as well as experimental evidence. The authors must provide evidence for the localization of the Fe pools detected as well as additional data that are more specific and more quantitative, or alternatively confirm their results with a different technique such as nanoSIMS or related (see more below).

Response: We agree that Perls-DAB does not stain exclusively Fe(III). We therefore rephrased all sentences and only indicate “Fe” without indicating the speciation when referring to results obtained with Perls/DAB staining. Determining the speciation of Fe pools is very difficult and may not be possible with nanoSIMS. To gain a glimpse at Fe(II) pools we used the Fe(II)-sensitive fluorophore RhoNox-I, which indicates the presence of less Fe(II) in the RAM of *hyp1* and more in *35S::HYP1* under low P (Suppl. Fig. 13d-e). To provide further support for Fe localization in the apoplast, we now present cross sections of Perls-DAB-stained root tips (new Fig. 5g), which indicate apoplastic localization, similar as reported previously (Müller et al., 2015).

(b) The authors also show decreased levels of RhoNox-1 staining for Fe(II) root tips of the *hyp1* mutant under phosphate deficiency (Fig. S11b). However, enhanced cell wall stiffening and callose deposition, which they also show or propose to occur in *hyp1* (Fig. 4d), are expected to decrease the permeability of the root tip for a dye like RhoNox-1. Moreover, the smaller meristem size of *hyp1* (Fig. S4e-f) is expected to contribute to decreased RhoNox-1 staining intensity in the root tip since it seems that staining is generally the most intense in meristematic cells, for example in the wild type. Alternatively, the iron probe visualizing Fe may also merely reflect a more oxidized redox state in this region of the root tip of the *hyp1* mutant.

Response: We note that RhoNox-1 staining was performed 1 day after transfer, when differences in meristem size between *hyp1* and WT were only minor. This point is now mentioned in lines 428-430. We also show one control to demonstrate that RhoNox-1 fluorescence is induced by Fe(II) but not Fe(III) and quantified RhoNox-1 signal intensities 1 day after transfer to +P or -P conditions (Suppl. Fig. 13c-e). The quantification showed that RhoNox-1 signals were more significantly decreased by low P in *hyp1* than WT while increased in *35S::HYP1*.

2. While the reviewer accepts the evidence supporting the idea that HYP1 is capable of transferring electrons from cytosolic ascorbate to an extracellular electron acceptor such as Fe(III) (Fig. 2fgkl, Fig. 3a-e), these data do not yet demonstrate that HYP1 does act as a ferric reductase under phosphate deficiency in planta. The N-terminal extracellular DOMON (dopamine beta-monooxygenase) domain could also interact with other extracellular electron acceptors in planta under phosphate deficiency conditions, for example the “ROS species” or downstream target compounds oxidized in the pathway leading to cell wall stiffening or callose deposition (Fig. 5d). Compared to the wild type there seems to be no significant increase in ferric reductase activity in *35S-HYP1* under -P and it seems – if data were corrected for multiple comparisons – that the decrease in *hyp1* would not be statistically significant (Fig. S11a, and lines 328-331). In relative terms, the decrease seems small – maybe by 20%. The reviewer feels that the support for this aspect of the authors’ model is too weak. The reviewer also sees partial contradictions compared to the data provided in Fig. 2c (done in Fe deficiency, however) and Fig. 4h (*35S::HYP1*) and wonders about the composition of media in the respective experiments.

Response: As shown by our results, HYP1 can also transfer electrons to Cu(II) and thus Fe(III) is not the sole electron acceptor. Due to the short half-life of ROS, a trans-membrane electron transfer assay to ROS has to our knowledge not been established. Our conclusion on the role of HYP1 in ferric reduction under low P is supported by the ferric reductase activity of isolated root tips and the staining with RhoNox-1 (Suppl. Fig. 13), which are significantly decreased in *hyp1* and increased in *35S::HYP1*, while the diminished disappearance of apoplastic Fe in *hyp1* meristems after transfer to low-P medium likely result from this activity (Fig. 6j,k).

The composition of the media used for the experiments shown in Figs. 4c and 6d (former 3c and 4h) were different because they tested different hypotheses. In the experiment shown in Fig. 4c, we used Fe-deficient conditions to assess HYP1-driven ferric reductase in the *fro2* background, while in the experiment of Fig. 6d the focus was on reductase activity in response to P conditions. We have now indicated these differences more clearly in the corresponding figure legends and in the Methods (lines 554-578).

3. Previous publications have shown that under phosphate deficiency, primary root growth of wild-type *Arabidopsis* plants ceases through enhanced extracellular Fe accumulation specifically in the stem cell niche and at the quiescent center of the root apical meristem, ROS cycling and callose deposition in the root stem cell niche. According to the data presented in this manuscript, there is only little callose deposition under phosphate deficiency in the wild type (and much more broadly and differently localized than previously reported by Müller et al. (2015), but callose deposition in a somewhat smaller region that is more likely to include the stem cell niche in the *hyp1* mutant. This suggests to the reviewer that these phenotypes depend a lot on the specific growth conditions, for example phosphate and Fe(III) supply and speciation in the growth medium. In fact, in this manuscript, the authors largely used a highly unusual Fe supply (highly elevated total and in non-chelated form). While it could potentially be argued that this reflects the conditions in natural soils more appropriately than other compositions, the authors do not provide any evidence for this view and do not even argue their choice to rationalize their conditions in that manner. Instead, they rationalize the growth system “to avoid confounding effects” (line 124-128). In addition, based on the information given in materials and methods, different types of experiments were carried out with different Fe and P supplies. For transcriptomics and growth under Fe deficiency, for example, EDTA was used as the Fe(III) chelator. In general, there are potential issues with piecing together results from different growth system to generate datasets that are later compared directly or integrated into one working model. In addition, the reviewer had difficulties inferring from the information given in the methods how each of the experiments was conducted (e.g. growth medium for Fig. S11a and b). The authors should refer to the specific growth conditions more explicitly for all data shown. They should give more background information on differences they observed between previously conditions that were previously commonly employed and the conditions they decided to use in the end (no Fe chelator and shielded from the light). The authors should refer more specifically to the stem cell niche and quiescent center when showing or discussing Fe accumulation and callose deposition, or discuss explicitly their view on the localization of root processes under low Pi.

Response: We apologize for not defining the growth conditions more explicitly in the original manuscript. Our initial transcriptome was performed in growth medium with Fe supplied as the soluble form as Fe(III)-EDTA, which was comparable to other studies (e.g., Müller et al., 2015; Dong et al., 2016 DOI 10.1016/j.molp.2016.11.001). However, to more specifically address the possibility that a mechanism of Fe(III) reduction is activated in response to malate release, we decided to supply Fe as the non-chelated form FeCl₃ instead of e.g. Fe(III)-EDTA to better allow the formation of Fe(III)-

malate complexes in the root apoplast and avoid confounding effects by chelate exchange reactions when using Fe(III)-EDTA. Compared to 100 μ M Fe(III)-EDTA, FeCl₃ is poorly soluble under the nutrient and pH conditions used in our experiments, as we now show in revised Suppl. Fig. 1d, justifying the use of elevated concentrations. Apart from the original transcriptome and the experiment shown in Fig. 1d, all other experiments dealing with P were performed with this Fe source. As suggested by the reviewer, we now 1) describe more explicitly the growth conditions used in the different experiments in the text and in legends of the corresponding figures; 2) explain in more detail in lines 134-141 and 563-578 what modifications were made in the growth media and why; and 3) refer more precisely the root zones in which the reported changes in callose deposition, Fe accumulation and cellular defects occur (e.g., lines 321-323, 328-321, 336-341). The differences in the growth conditions (Fe source, shaded vs non-shaded roots) used in our study and the study of Müller et al. (2015) likely explain the differences in the pattern of aniline blue staining in Col-0 roots. However, as now reinforced by the quantification of aniline blue-derived fluorescence, callose deposition was significantly increased in the RAM of Col-0 plants grown under low P (Fig. 5e,f).

4. The reviewer believes that also in the ferric reduction assays carried out using roots, authors should always indicate which Fe(III) complex was used as a substrate, in the figure legend or in the panel. This seems to differ, e.g. in Fig 4h from Fig. 3c. The consequence is identical apparent ferric reductase activities in Fe-deficient roots (Fig. 3c) and in roots of plants grown in control Fe conditions (Fig. 4h), which seems counter-intuitive.

Response: Thank you for this suggestion. We now indicate in the legends of Fig. 4c, 6d and Suppl. Fig. 13b which Fe(III) complex was used in the ferric reduction assays. When assessing HYP1 ability to complement FRO2 in response to Fe deficiency, we used Fe(III)-EDTA following the standard protocol. However, in experiments with low P, we supplied freshly prepared Fe(III)-malate complexes, as this is the more physiological relevant substrate under low-P conditions. We now explain the reasons for these choices when describing ferric-chelate reductase assays in the Methods.

5. The authors refer to the issue of the inhibition of primary root elongation under low P known in wild-type Arabidopsis. In the context of the present work: Is it seen as unfortunate Fe toxicity or an adjustment of root architecture that overall serves to improve plant fitness under phosphate deficiency? This should be discussed and clarified, and the role of composition of agarose-based media should be addressed (see also 3.).

Response: In our study, we did not directly address whether the primary root inhibition under low P provides an adaptive advantage to plants. However, several previous studies (e.g., Svistoonoff et al., 2007; Ward et al., 2008 DOI: 10.1104/pp.108.118562; Müller et al., 2015) have shown that Fe is critical for this response. The increased Fe reactivity (and eventually toxicity) may originate from the release of malate. In fact, disruption of malate export in the *almt1* mutant prevents root growth inhibition (Balzergue et al., 2017; Mora-Macias et al., 2017). Perhaps plants take advantage of the interaction of Fe and P to modulate root growth and to alter their root system architecture to increase foraging in the topsoil rather than in depth. Our study suggests that irrespective of whether the attenuation of primary root elongation improves plant fitness under P deficiency or not, if the increased Fe availability resulting from the stimulated malate exudation is not counteracted by HYP1 activity, it results in irreversible loss of the root apical meristem. We thank the reviewer for instigating the discussion in this direction. We discuss now these points in lines 511-522.

6. Continuing on 1(a). The enhanced Perls/DAB method used was taken from a publication in which embryos dissected from seeds or fixed sections of mature seeds were stained (Roschztardtz et al., 2009). The authors summarize the method briefly, but this suggests that they did not use the same method. Incubation times are different, and the authors here do not mention CoCl_2 , which was used by Roschztardtz et al. (2009). Composition of solutions and all incubation times used for the Perls/DAB staining are absolutely critical and must be presented here clearly and in completion.

Response: Indeed, we had to make some adjustments in the original method as e.g., the original concentration of HCl reported in Roschztardtz et al. (2009) often resulted in the desintegration of the meristems. We omitted CoCl_2 , which functions as a DAB enhancer, because we already detected strong signals without it and the brown precipitates provide a better contrast to imaging than the intense, black-colored precipitates formed when CoCl_2 is included. We now mention that the Perls/DAB staining was based on Roschztardtz et al. (2009) but fully describe the methodology that we used in “Fe staining” in the Methods section (lines 717-729).

7. Continuing on 1(a). Unlike roots, staining for Fe of embryos dissected from seeds (Roschztardtz et al., 2009) is not impacted by Fe diffusion from the medium into the apoplastic space and binding to or precipitating on the cell wall. In most experiments published, there are enormous amounts of Fe detectable by Perls stain in the root apoplast or evidently present based on the magnitude of Fe concentrations detected in bulk roots. Finding this alone is not sufficient to conclude on Fe-induced toxicity. In addition, for such staining procedures, whether and to which degree apoplastic Fe has been desorbed is critical information. The authors state that roots were rinsed in EDTA. This is insufficiently precise. Rinsing in EDTA will begin to remove some, or with time possibly most or even all, of the apoplastic Fe. The period for which such a rinse was conducted is absolutely critical, such rinses must be stringently controlled, and this information must be given in full and rationalized, in the context of the topic of this manuscript.

Response: Thank you for pointing this out. Indeed, in our original stainings we also briefly rinsed roots with EDTA prior to staining to remove Fe pools loosely adhering to the root surface. However, following the reviewer’s concerns, we repeated all Perls/DAB experiments this time only including a brief 1-min rinse with double-distilled water. The overall pattern of Fe distribution did not change substantially and our conclusions remained unchanged. As mentioned above, the methodology used is fully described in “Fe staining” in the Methods section.

8. Continuing on 1(a). Line 336: “More Perls/DAB stainable ferric Fe”. Perls/DAB is not specific for ferric iron. In fact, the methodological staining procedure includes conditions provoking redox cycling of iron for stain enhancement (Roschztardtz et al., 2009). So in the eyes of the reviewer, it is not justified to conclude on the oxidation state of Fe in vivo.

Response: We agree with the reviewer. We wanted to refer to ferric Fe as part of the pool stained by Perls/DAB. We corrected this mistake throughout the manuscript.

9. Continuing on 1(a): Lines 345-347: The authors describe here that the dot-like structures they observe after Fe(II) feeding for 6 d are observed equally in root tips of the wild type and the hyp1 mutant. The following conclusion in lines 348 to 350, saying that altogether there is evidence for a

HYP1-dependent Fe(III) to Fe(II) reduction in root tips critical for root elongation in -P does not hold up in my opinion because it is not conclusively shown that when Fe(III) is provided externally, less Fe (in total, there could be re-oxidation inside cells) ends up inside those cells in the root tips of *hyp1*, when compared to the wild type. Perls/DAB indicates that there is more Fe altogether after a longer exposure time of 6 d (Fig. 4c) or 3d (4g), but treatment of FeIII (similar to Fig 4g) and FeII (similar to Fig. 5c) should be shown side-by-side and conducted identically in identically pre-cultivated plants.

Response: We agree with the reviewer that this was not sufficient direct evidence for Fe uptake in root tips. During the revision, we tried to establish a methodology to investigate more directly Fe uptake using the stable ^{58}Fe isotope and our sector-field high-resolution ICP-MS device. However, it remains challenging to expose only the RAM and elongation zone to the isotope, and only very short incubation times can be used as tips continuously grow and cells differentiate. Furthermore, to determine how much Fe ends up in cells of these root zones, it is necessary to quantify ^{58}Fe concentrations in protoplasts. So far, we could not achieve sufficient resolution for reliable quantification of the isotope in such small samples. Therefore, we omitted the Perls/DAB staining of Fe(II)-exposed plants from the revised version. Instead, we focus on other aspects raised during the revision and provide new evidence for the role of Fe in the phenotypes (incl. RNA-seq) and show in which cells HYP1 is more critical for sustaining root growth under low P.

10. Continuing on 1(a). Lines 379-380: “raised initial evidence for the existence of a Fe acquisition mechanism in this root zone Fig. 5a-c.” 5a: The difference in Fe concentrations between genotypes detected here in the shoot may result from indirect effects or from HYP1 presence elsewhere in the root or even shoot. 5b: contains no evidence for HYP1 contribution to Fe acquisition. 5c also doesn't in another sense, but shows that Fe must get in somehow – this could occur also non-specifically via systems mediating influx of other divalent cations. The Fe-containing dots appear too large to constitute ferritin proteins – do they correspond to plastids (then please adjust the corresponding text, line 347). To the reviewer the composition of the medium enriched in Fe(II) used for Fig. 5c is unclear.

Response: We agree that the shoot Fe concentrations may not be related to HYP1 activity in root tips but demonstrate now that *HYP1* expression only in root tips is sufficient to maintain root growth under low P (new Fig. 6e-i). Regarding the Perls/DAB-stained dot-like structures, we determined that they are intracellular and, from their size, likely represent ferritin contained in plastids (Fig. 1 for Reviewers). However, since we cannot directly determine whether the Fe present in these structures originates from uptake, we omit these results in the revised version.

Figure 1 for Reviewers.

11. Continuing on 1(a) and 9. In relation to the working model, a strategy by plant cells to prevent apoplastic Fe(III) toxicity by first generating Fe²⁺ in the apoplast and then taking Fe²⁺ up into the cytoplasm seems risky and somewhat counter-intuitive. The authors should explain how they envisage this to function.

Response: We are sorry that we did not discuss this point in more detail in the earlier version. We hypothesize Fe²⁺ internalization allows plants to defend better against an increase of Fe solubility, as the excess Fe can, within limits, be stored and detoxified Fe in vacuoles and ferritins, respectively. It is also possible that the relatively young cell wall of meristematic cells is less more sensitive to Fe overload or that Fe binding to and its effect on cell wall components differs along the axial zones of roots. However, these possibilities are very speculative at this stage. Instead, we extended the Discussion on other points more directly related to the presented data (see revised Discussion section).

12. L. 76/77: “protein that is induced...”: Please demonstrate at protein level, e.g. by immunoblot or analysis of sufficient CLSM frames for quantification and statistical analysis under identical settings and conditions, or alter that statement. Fluorescent images alone (Fig. 2b, Fig. S5) are not sufficiently (semi-)quantitative. Increased levels of HYP1 (or HYP1-GFP) protein in root tips of Pi-deficient plants. Else the statement should be modified.

Response: We have now repeated the experiment and quantified GFP signals in root tips of plants grown under +P or -P with the same microscope settings. As shown in new Fig. 2d, HYP1:GFP-derived fluorescence intensities increased significantly in the area above the QC but not in the root cap in response to low P. We therefore, rephrased the sentence to (lines 207-209): “Low P availability increased the abundance of HYP1:GFP and, in the RAM, induced its appearance in several cell files above the stem cell niche while having no effect on HYP1:GFP-derived fluorescence in root cap cells (Fig. 2d and Suppl. Fig. 5)”

13. Fig. S5 suggests the presence of HYP1 protein and a substantial upregulation of its protein levels under phosphate deficiency, although this is preliminary, also in the root hair zone. This has the potential of significantly affecting the interpretation of the results presented here. Presently, the interpretation root growth phenotypes and alterations in root and whole-plant Fe levels and homeostasis is based exclusively on changes observed in the meristem, but the phenotype is the integrated consequence of all locations in the plant where HYP1 is active. The authors should experimentally address in a quantitative manner also what happens in the root hair zone and incorporate this information in their interpretation and working model. This may, for example, contribute to the observation of an apparent contribution of HYP1 – directly or indirectly – to shoot Fe levels under low P (Fig. 5b). The authors should also comment on the predominant signal in the outer cell layer – is this autofluorescence or HYP1-GFP (a non-transformed control, for example, would answer this question)? If it is HYP1-GFP (seems likely), how does this affect the interpretation of results, given that the proposed function of apoplastic Fe accumulation under phosphate deficiency – to the understanding of the reviewer – is highly localized to the stem cell niche of the root tip according to Muller et al. (2015)? Please make modifications in the text to adjust the description of HYP1 localization (lines 182-188), if needed, and to discuss this comment.

Response: To address the reviewer’s concerns, we investigated in which cells HYP1 activity is more critical to maintain RAM integrity under low P. We therefore expressed HYP1:GFP under the control of different promoters (*proHYP1*, *proBRN1*, *proLPR1*, or *proSGN1*). The new results show that HYP1

activity around and in the stem cell niche but not in the root cap or mature endodermis is required to fully restore primary root growth and prevent RAM desintegration. This largely coincides with the zones in which Fe and callose overaccumulates in the mutant, and is in line with the results of Müller et al. (2015). The new results are presented in Fig. 6e-i and described in lines 399-416.

14. Further relating to the preceding point: In line 338 to 340 the authors explain how the wild type may end up containing higher Fe concentrations in shoots under low-P conditions than hyp1 mutants. The explanation is provided that HYP1 decreases apoplastic Fe sequestration in the wild type so that more ends up inside the root cells (in RAM, see Fig 11b). However, it is not clear whether the Fe in these specific cells would be mobile for movement from the root to the shoot. In fact, it is a little less likely because it is thought that only Fe taken up in the root hair zone is mobile for transport into the mature xylem vessels and up into the shoot. So do the authors envisage HYP1 (also) to act further up in this zone?

Response: Good point. While HYP1 activity in a relatively small number of cells in the RAM is related to the root phenotype, the differences in shoot Fe concentrations may additionally be influenced by HYP1 present in the inner-most tissue of mature root zones. Our hypothesis is that Fe(III) reduced by HYP1 in the RAM is taken up and either directly used or stored. The lack of a vascular connection with these cells makes it unlikely that this Fe would be transferred to shoots. Since the presented results were obtained from plants cultivated for 20 days on +P and -P and we now focus more specifically on the role of Fe in RAM integrity, we removed this dataset from the revised manuscript.

15. Line 301: Perls/DAB stainable apoplastic Fe... - apoplastic localization must be demonstrated. In fact, this contradicts with line 344, where the dots observed in Fig. 5c are interpreted as intracellularly localized. This means that the stain can localize both extracellular and intracellular Fe, and – without further modification or optimization of the method and associated validation - based on the microscopic images it cannot be concluded where the Fe is localized or which oxidation state it was in the living plant. This relates to some of the earlier comments.

Response: We now clarify this point by showing cross sections of Perls/DAB-stained roots in which staining can be more clearly seen in the apoplast (new Fig. 5g). If Fe is supplied in the form of Fe(II), Perls/DAB stain in the apoplast become less intense while more intense staining is indeed detected in intracellular dot-like structures, as we show in Fig. 1 to Reviewers (see point 10).

16. In a number of instances, single root tips are shown for comparison to argue quantitative differences between them. A number of frames of root tips following identical treatment and processing and imaging with identical settings should be provided in the supplement, as well as image-based quantification with statistical analysis (e.g. Fig. 4c, d, g; Fig. 5a, c; Fig. S5; Fig. S10; Fig. S11b; possibly Fig. S4e, f).

Response: In all cases, we show representative results from independent experiments. To provide statistical support to the representative images, in the revised manuscript we present the quantification of HYP1::GFP-derived signals in Fig. 2d, and image-based quantification of Perls/DAB, callose and RhoNox-1 in Fig. 5d and f, and Suppl. Figs. 13a,e.

17. Suppl. Fig. 2: It would be important to know whether there are any remaining full-length transcripts, and what the levels are of truncated transcript, in the *hyp1* T-DNA insertion line. Also please indicate position of insertion shown in Fig. S2a relative to coding sequence, to specify which part of the HYP1 protein might be left in the *hyp1* mutant.

Response: As replied to Reviewer #1, we checked *HYP1* expression in the T-DNA insertion mutant by qPCR. As shown in new Suppl. Fig. 2a, *HYP1* expression is strongly decreased in the mutant compared to wild type irrespective of the P condition. The insertion is in the sole intron, with exon 1 coding for the DOMON and a short part of the apical side of the cytochrome *b561* and exon 2 for the majority of the cytochrome *b561* domain. This is now also indicated in the scheme shown in Suppl. Fig. 2b.

18. In this same context, it would be better to have reproduction of the phenotype under – phosphate in a second, independent T-DNA insertion line. Complementation lines in the *hyp1* mutant background, in conjunction with mutant HYP1 variants fused to GFP (Fig. 2m) in the same background, compensate for the lack of an independent mutant to some extent, but not entirely. Could the authors please demonstrate the presence of the protein under phosphate deficiency in the H211L/H278L lines (e.g. by confocal imaging of the GFP fluorescence)?

Response: There is no second independent T-DNA insertion line available in which *HYP1* was knocked out. However, we believe that the fact that *HYP1* transcripts are strongly decreased in *hyp1* (new Suppl. Fig. 2b), that *hyp1* can be complemented with the *HYP1*'s full-length ORF or cDNA, that *HYP1* overexpression results in an opposite phenotype compared to the mutant, and that a double mutant with another CYBDOM gene (i.e., CRR) impairs the phenotype, provides convincing and sufficient evidence that the mutation in *HYP1* is indeed causal for the low P hypersensitivity.

We now show a image of *pHYP1::HYP1^{H211L/H278L}:GFP* in Suppl. Fig. 8g. In line with the localization obtained in oocytes, *HYP1^{H211L/H278L}* still integrates in the plasma-membrane also *in planta*.

Comments on both manuscripts jointly

Response: The responses to these comments were formulated jointly and slightly modified to highlight points that are specific to each manuscript.

19. The reviewer wonders whether by supplying ferricyanide (Fig. 4A and B) as a substrate to CRR proteins, a previously reported general action of ferricyanide in protein-bound heme oxidation comes to bear, and whether the observed ferrireductase activities in this manuscript are thus an observation that does not correspond to the biological function of CRR. The reviewer wonders whether while CRR can donate an electron to Fe³⁺ under appropriately designed/altered conditions, this is in fact a negligible process in *planta* under the usual range of physiological conditions including phosphate deficiency. The authors of both manuscripts should contemplate (and possibly provide reasons against) the following model: HYP1 or CRR might donate an electron to directly quench extracellular ROS and thus Fenton chemistry in the apoplast of the stem cell niche. There needs to be a fine balance though, because in principle and under “permissive” conditions, electron donation to apoplastic molecules via CRR or HYP1 (e.g. in overexpressing shoots) can alternatively enhance ROS production and thus promote Fenton chemistry. Next, evidently, in –Pi conditions, LPR1/2 alone has sufficient Fe²⁺ substrate available to do its job, even without CRR (or without HYP1), and possibly without both of them (?), and thus LPR1/2 alone is able to account for the Fe-dependence of primary

root growth inhibition phenotypes; the need for something additional (CRR/HYP1) to generate Fe²⁺ (in this case to moderate root growth inhibition) is not evident. Quantitative depletion of Fe(III) in the root apoplast through reduction to Fe²⁺ seems like a goal that is hard to achieve in a location that is permeable to the soil solution containing plenty of Fe(III).

Response: The link between Fe and primary root growth under low P has been demonstrated in previous publications describing numerous mutants in this pathway (*lpr1*, *pdr1*, *almt1/stop1*, *als3*) as well as with *hyp1* and *crr* in the co-submitted manuscripts. Thus, the case for implicating Fe as the electron donor/acceptor in this pathway is quite strong. Nonetheless, as the reviewer indicates, the topic of ROS generation is very complex, with related pathways able to either generate or sequester ROS depending on the conditions. Although the case for the implication of HYP1/CRR and LPR1 in Fe oxido-reduction is strong, we cannot exclude that other molecules may be involved, either as direct substrates for these CYBDOMs, or in secondary reactions following Fe reduction. We discuss these points in more detail in the text (lines 471-478, 516-519 and 529-532).

The authors of both manuscripts do not think that the function of HYP1 and CRR is to provide Fe²⁺ to LPR1. Instead, we raise evidence that HYP1 can reduce Fe(III) chelated by malate. The reviewer reasoned that “Quantitative depletion of Fe(III) in the root apoplast through reduction to Fe²⁺ seems like a goal that is hard to achieve in a location that is permeable to the soil solution containing plenty of Fe(III)”. This sounds logical but so far, the experimental evidence contradicts this hypothesis since LPR1 is needed to regulate the primary root growth inhibition by promoting Fe³⁺ production. The root growth of *lpr1* mutant is insensitive to low P even when plenty of Fe³⁺ is present in the media, indicating that Fe³⁺ must be generated *de novo* by LPR1 to have its effect. We now also discuss some possibilities to explain these observations (lines 512-522). We have tried in the Discussion to hit the right balance between providing hypothesis on how HYP1 may work while avoiding being too speculative.

20. How do the authors explain the root growth difference in *crr1 almt1* (barely longer/meristem size not larger than *crr*, only about 50% root length of *almt1*) compared to *hyp1 almt1* (like *almt1*) in the accompanying manuscript under -Pi?

Response: In an attempt to clarify this point, both labs have exchanged seeds and grown them according to the protocol used in each respective lab. In our lab and with our growth conditions, disruption of ALMT1 does not significantly alter the phenotype of the *crr* (Fig. 2 for Reviewers). On the other hand, the less sensitive phenotype of our *hyp1 almt1* double mutant was confirmed in the Poirier lab, even though under their conditions the double mutant had significantly shorter primary root under low P than *almt1* (Fig. 2 for Reviewers).

Clearly, these results indicate that differences in experimental conditions used between the two labs are having an effect on the epistatic relationships with *almt1*. The protocols between the two labs differ in many aspects, including media composition (e.g. 1/6 MS vs 1/2 MS; FeEDTA vs FeCl₃) and growth condition (e.g. plants grown directly on +/- Pi plates vs plants grown first on +Pi and then transferred to +/- Pi plates; light-exposed vs light-shielded agar plates). Obviously, the interactions between ALMT1 and the two CYBDOMs are complex, and likely involve several factors, including the influence of growth conditions as well as the expression pattern of the genes. The growth conditions may also impact on the role of ALMT1-dependent malate exudation and cause discrepancies, as reported for Fe distribution in previous studies (Mora-Macias et al., 2017 and Balzergue et al., 2014).

Besides the growth conditions, the less pronounced effect of ALMT1 disruption in the *crr* background could be related to putative differences in the location of CRR and HYP1 and the from the fact that

CRR is not involved in cell elongation. We feel that fully dissecting the complexity associated with ALMT1 is beyond the scope of our papers and that explaining all of these results would unnecessarily divert the attention of the reader away from the main outcomes of our studies, being the impact of HYP1 and CRR on root growth and Fe homeostasis. Thus we provide this information to the reviewer but have not included it in our manuscripts.

Fig. 2 for Reviewers.

21. The authors should provide results from the root growth assay conducted under phosphate-sufficient and -deficient conditions also for the *crr hyp1* double mutant, in addition to the single mutants and the wild type. Including this in both manuscripts will contribute to the comparability of both manuscripts because the media and cultivation systems were different.

Response: In the previous version, we already showed the phenotypic analysis of *hyp1 crr* in our growth conditions (Suppl. Fig. 3). However, the single *crr* mutant was not included in that experiment. Following the reviewer's request, we performed a new experiment with wild-type, *hyp1 crr-1* and the two single mutants. According to our results (updated Suppl. Fig. 3e,f), there are only small differences between these mutants under sufficient P, while under our low-P conditions there is a gradient in the length of primary root growth Col-0 > *crr* > *hyp1* > *crr hyp1*. In the companion manuscript of Clúa et al., similar results were obtained although the media and growth conditions used in their study were distinct (as described in point 21), showing the robustness of the phenotypes.

22. The reviewer is struggling with the impression that plenty of Fe (see Fig. 2G, likely mostly Fe(III)) seems to be present near QC/SCN in +P conditions, but no stalling of primary root elongation occurs. While the presence of Fe is clearly necessary for this under -Pi conditions, the reviewer wonders whether much of what is observed several days later is merely a secondary and broader consequence of highly localized events that occurred within a much shorter period of time after the initiation of Pi deficiency.

Response: We agree with the reviewer that it is somewhat puzzling that under +P condition, although Pearls-DAB staining does reveal adequate level of apoplastic Fe, primary root growth is not stalling. Previously, Müller et al. (2015) showed that under sufficient P, apoplastic Fe is colocalized with P, suggesting that Fe and phosphate form a complex. Such co-localization was not detected under low P. The increased secretion of malate in response to P deficiency increases the solubility of apoplastic Fe, making it more prone to engage in redox reactions. Unfortunately, the pools of soluble and insoluble Fe cannot be distinguished by Pearls/DAB. We cannot formerly exclude the hypothesis suggested above by the reviewer. It is also plausible that the chemistry of the apoplast under low-P conditions may impact in what form/complex Fe is present, how it interacts with cell wall components, and how P-deficient meristematic cells react to malate-mediated Fe solubilization. All of this is quite speculative at this point, but again useful to keep in mind while research on the topic is progressing.

23. The authors should consider in their models that the CYBDOM proteins may act somewhat independently in different parts of the plant – so in the opinion of the reviewer, if they generate an extracellular Fe²⁺ pool that is then taken up to contribute to shoot Fe levels, this Fe may not originate from the Fe²⁺ pool localized where stalling of root growth occurs under -Pi conditions. HYP1 appears to be present in a variety of cell types and even under +Pi conditions in a subset of these. This complicates pinning down the location of where the protein acts to contribute to foster primary root elongation under -Pi.

Response: We agree with the reviewer that different CYBDOM members may act independently in different cells/tissues, and that the contribution of CYBDOMs to the transport of Fe (and perhaps other metals like Cu) to shoots (or other tissues) might be, to some extent, independent of the localized effect that promotes root arrest under low P.

To address the reviewer's specific comment on HYP1, we performed a cell type-specific complementation experiment. By expressing *HYP1* in specific cells of the root tip, we found that *HYP1* location in a small population of cells around the stem cell niche in the proximal part of the RAM is necessary to sustain wild type-like root growth and meristem integrity under low P (new Fig. 6e-i). The new results provide support that *HYP1* location in these cells and not in the root cap is

more critical for preventing RAM desintegration under low P, matching with the low P-induced HYP1 protein localization in wild type roots (Fig. 2c,d). Outside this domain and beyond P deficiency, HYP1 and other CYBDOMs might help to fine tune Fe-dependent redox homeostasis perhaps in concert with LPR1/LPR2, which also locate in plant parts other than root tips (e.g., Xu et al., 2022 DOI:10.1016/j.molp.2022.11.003 and Zhu et al., 2022 DOI: 10.3389/fpls.2022.958984). We discuss these possibilities in lines 522-524.

Minor comments

24. The reviewer would like to see more discussion on the localization of HYP1 protein in relation to the localization of Fe (e.g. line 301). There appears to be a lot of Fe in the root hair zone in Fig. 4g (see also comments 13 and 14).

Response: Thank you for this suggestion. The new longitudinal sections Perls/DAB-stained roots (Fig. 5g) allowed us to more clearly determine where Fe is deposited. In lines 483-489, we now discuss how this localization refers to HYP1 localization and the cell type-specific complementation of *hyp1*. The deposit of Fe at the the outgrowing tip of root hairs was described previously (Müller et al., 2015). However, since we did not investigate root hair-related parameters in this study, we did not look at this localization further.

25. The authors are asked to provide the information of how much phosphate and iron (μM in the final medium) the agar and the sugar components added to their media.

Response: A full description of the agar composition used in the different experiments is provided in the Methods (lines 554-578). The concentration of P and the concentration and form of Fe supplied in the different experiments are now indicated in the corresponding figure legends.

26. Line 307: the ferric reduction activity of HYP1: the activity was not measured using the purified protein, but in planta and in oocytes. The conclusion here should be adjusted to ensure its accuracy.

Response: Thank you for pointing this out. We now rephrased to (lines 378-379): "...the HYP1-dependent ferric reductase activity in roots..."

27. L.298-299: "apoplast" ... "as shown previously". The publication referred to here shows a more highly localized accumulation of Fe in phosphate deficiency than the broad accumulation shown in Fig. 4c, g; 5a). While Muller et al. (2015) examined in very much detail the localization of Fe also in the apoplast in very specific localizations, this demonstration is not conducted here.

Response: We now present longitudinal sections Perls/DAB-stained roots (Fig. 5g), which indicate more clearly where Fe is deposited in the RAM. The new results are described in lines 335-345.

28. The *hyp1* phenotype under $-P$ is clear, but much less dramatic than the end of the introduction suggests. Please adjust the wording of the text.

Response: We now changed the sentence to (lines 82-85): “Due to the more severe loss of meristematic integrity and more significantly inhibited cell elongation of a loss-of-function insertional mutant specifically under low P conditions, we named the gene HYPERSENSITIVE TO LOW P1 (HYP1).”

29. Line 83/84: “reduces ferric and cupric ions”: Reviewer feels that the protein “can reduce...”. This is more consistent with the conditions under which this was observed in the present manuscript. Line 263: HYP1 mediates ferric and cupric reduction (“can mediate”). Lines 274, 290: “can function as”.

Response: Thank you. We rephrased the sentences as follows: lines 87-89 to: “We show that HYP1 is present in root tips, mediates ascorbate-dependent trans-PM electron transport, and can reduce ferric chelates and cupric [Cu(II)] ions.” ; line 284: HYP1 can mediate ferric and cupric reduction; lines 294-295: “...HYP1 can mediate ferric-chelate reductase activity.”; lines 313-314: “Altogether, our results demonstrate that HYP1 can function as Fe(III) and Cu(II) reductase”.

30. Lines 84/85: critical to prevent Fe overaccumulation in the apoplast – not addressed directly in this manuscript

Response: The new Perls/DAB-stained of semi-thin root sections (Fig. 5g), indicate that stained Fe pools were mainly apoplastic. Therefore, we think the sentence is justified.

31. In Fig. 1d, the phenotype does not appear to be very strong here (and it is hard to relate the photo to the measurements in 1e), and it would be critical for the reader to know where the root tip was positioned in the moment the low-P treatment was initiated. Could the authors please mark this in the plates and also add a scale bar in 1d.

Response: The quantification refers to the length after transfer. This is also indicated in the figure label labels. To facilitate the interpretation of the results, we now marked the position of the primary roots at the day of transfer in all agar plate images. The scale bar was added.

32. In Fig. 1f, overexpressor roots seem longer than wild type, but not in 1g – why? Can the authors clarify this in the figure or manuscript text.

Response: For the sake of space, we only present pictures of plants grown on -P and primary root length quantification of plants grown under +P and -P. As shown by the quantification (Fig. 1g) , HYP1-overexpressing lines have comparable primary root length under sufficient P but significantly longer roots at -P.

33. In Fig. 1h, *hyp1* hardly seems to differ from Col- under -P until 3 d. This contradicts Fig. 1d-g.

Response: This is because measurements for Fig. 1d-g were taken at 6 d after treatment. The experiment shown in Fig. 1h allowed us to determine that primary root of *hyp1* almost completely stops elongating after 2 days on low P, while the primary roots of wild-type plants can still elongate even if very slowly (compare values at 2 and 3 dat). Over a period of 6 days, these differences in length become much more prominent.

34. Line 304: “and decreased cell-cell.communication” – replace by “and decreased cell-to-cell mobility of proteins (add where this was observed because it was a highly localized observation, and it remains unknown what is happening elsewhere, e.g. across the meristem).

Response: To more clearly reflect what we observed, we rephrased the sentences as follows (lines 328-334): “By staining callose with aniline blue, we found that the increased Fe accumulation in *hyp1* meristems was accompanied by increased callose deposition, especially in the stem cell niche and around cortical and endodermal cells in the meristem (Fig. 5e,f). Furthermore, we detected decreased mobility of the transcription factor SHORT ROOT (SHR) from the stele into quiescent center cells, suggesting that cell-to-cell mobility of proteins between these cells was impaired in *hyp1* roots (Suppl. Fig. 11).”

35. Fig. 1i: *hyp1*, but none of the other lines, have a large number of root hairs initiating already very close to the root tip from 2 d after transfer into low-phosphate on. This observation should be referred to in the text. What does it mean (also in relation to Fig. 3k)?

Response: At least in part, this is because the root meristem size and, especially, the cells in the elongation zone were shorter compared to WT (Suppl. Fig. 4b-d). We did not assess whether low P also accelerated root hair differentiation in *hyp1* plants. We now refer to this observation in lines 183-185: “Probably as a result of the smaller meristems and shorter cells, and potentially of an accelerated cell differentiation, root hairs were detected much closer to the primary root apex of *hyp1* than in wild-type plants under low P (Fig. 1i).”

36. Fig. 3k: The image for 35S::HYP1 shows a lot of root hairs already very close to the root tip in seedlings grown in high Cu. This observation should be referred to in the text. What does it mean (also in relation to Fig. 1i)?

Response: As for *hyp1* under low P, this phenotype was probably largely due to the smaller root meristem size and shorter cells of 35S::HYP1 under high Cu compared to WT. To measure the size and number of cells in the meristem and the length of mature cells, we repeated the experiment. In the revised version we present root tip images and quantifications of plants grown under control and high Cu (Fig. 4k and Suppl. Fig. 10b). The new results are mentioned in lines 309-313.

37. Line 386-7: The reviewer would generally expect that intracellular Fe-mediated ROS generation is a much bigger risk than in the apoplast.

Response: This will depend on the activity of Fe detoxification mechanisms in different cells. One can also expect that cells have less control over Fe pools located in the apoplast than intracellular pools. Inside cells, excess Fe can be stored in vacuoles and/or ferritins and the near-to-neutral pH of the cytosol favors Fe²⁺ chelation by nicotianamine. In contrast, ferrous Fe generated in the apoplast by chemical reductants or by light can engage in Fenton chemistry to form the highly toxic hydroxyl radicals. Under circumstances in which apoplastic Fe concentrations become very high, such as in response to malate release, there is a very high risk for Fe-driven Fenton chemistry.

38. 396-7: CYBDOM-dependent ROS formation would be predicted to have the opposite effect, i.e. less root growth inhibition in the mutant. Add one more sentence to explain how this is meant, or else maybe delete this?

Response: We rephrase the sentence (lines 259-532).

39. In some figure legends, n is not formatted in italics.

Response: Thank you. Corrected!

40. Line 44: “very low or only hardly available” – both mean almost the same – please rephrase.

Response: In the sentence, “very low” was referring to total amount of P present in the soil, while “hardly available” to P solubility. In order to make this point clearer, we now rephrased to (lines 49-50): “However, in most soils, total amounts of P are very low or the P that is present is only poorly available to plants.”

41. Line 80/81: “comprised of” rephrase

Response: We rephrased to (lines 85-87): “The protein is a member of the yet poorly characterized CYBDOM family and consists of a plasma membrane-embedded cytochrome b561 domain fused to an apoplastic dopamine β -monooxygenase N-terminal (DOMON) domain.”

42. Line 100: “downregulated by P deficiency” – please rephrase, e.g. “downregulation in response to...”. There are a number of similar instances in the manuscript.

Response: We changed the sentence and corrected the term to “in response to” elsewhere in the text.

43. Line 102: The downregulation of IRT1 transcript levels under P deficiency was reported before and extensively discussed in the literature; authors should mention this and cite the publication that was the first to report this.

Response: Sorry for this oversight. We added the references and rephrased to (lines 108-109): “The expression of major genes involved in root Fe uptake, including *FRO2* and *IRT1*, was indeed down-regulated, as reported earlier (Misson et al., 2005; Thibaud et al., 2010; Li & Lan, 2015; Hoehenwarter et al., 2016).”

Misson J, Raghothama KG, Jain A, Jouhet J, Block MA, Bligny R, Ortet P, Creff A, Somerville S, Rolland N, et al. 2005. A genome-wide transcriptional analysis using Arabidopsis thaliana Affymetrix gene chips determined plant responses to phosphate deprivation. Proc Nat Acad Sci USA 102(33): 11934-11939.

Thibaud MC, Arrighi JF, Bayle V, Chiarenza S, Creff A, Bustos R, Paz-Ares J, Poirier Y, Nussaume L. 2010. Dissection of local and systemic transcriptional responses to phosphate starvation in Arabidopsis. Plant J 64(5): 775-789.

Li W, Lan P. 2015. Genome-wide analysis of overlapping genes regulated by iron deficiency and phosphate starvation reveals new interactions in Arabidopsis roots. BMC Res Notes 8: 555.

Hoehenwarter W, Mönchgesang S, Neumann S, Majovsky P, Abel S, Müller J. 2016. Comparative expression profiling reveals a role of the root apoplast in local phosphate response. *BMC Plant Biol* 16.

44. Line 148: ... “showed more severe” is unclear – rephrase sentence?

Response: We rephrased to (line 163): “...showed significantly shorter primary root length compared to wild type...”

45. Fig. 2a: add higher-resolution images of root tips.

Response: Thank you for the suggestion. We now present high-resolution images of GUS-stained roots in Fig. 2b and mention the results in lines 200-202. Please, not that, since we added new GUS images and quantification for HYP1-GFP fluorescence, the original Fig. 2 is now split in Figs. 2 and 3.

46. Fig. 1: In (c), the label should say “NADPH ferric reductases”, not “NADH...”

Response: Thank you. Corrected!

47. Line 151: replace “play” by “have”

Response: Following the concern of Reviewer #1 that our results are not sufficient to conclude about functional redundancy, we now rephrase the sentence to (lines 164-167): “The simultaneous disruption of *HYP1* and *CRR* further exacerbated the primary root inhibition under low P (Suppl. Fig. 3e,f), suggesting that the activities of these two CYBDOM proteins act in an additive way to sustain root growth under low-P conditions.”

48. Fig. 2 and Legend Fig. 2, FeCN: please replace by correct chemical formula;

Response: FeCN was replaced with the chemical formula $[\text{Fe}(\text{CN})_6]^{3-}$ in all corresponding legends.

49. Amino acids mutated in Fig. 2k, I should be marked in h or j, clarify in h or j the position in the protein of what is shown in i. Legend must define arrows in c, f, k, as well as in each case both the round shapes symbolizing a metal ion (which one?) and all cofactors in h, j and n.

Response: The amino acids are shown in Fig. 3c (former 2l). Arrows and arrowheads are defined in the legend of Fig. 2c, d and g (former 2c, f and k).

50. Fig. 3 and Legend Fig. 3. h: Adjust statistics to correct for multiple comparisons of means.

Response: Since for our conclusions we are only interested in the comparisons to wild type (Col-0), we maintained the pairwise Student’s *t*-tests in Fig. 4h (former 3h).

51. Fig. 4g add size bars.

Response: The size of the scale bars is now indicated in the legend.

52. Fig. S3a: specify technique in legend – assume microarray data? An independent confirmation of HYP1 transcript levels via RT-qPCR would be nice.

Response: Yes, the expression patterns derived from the microarray study. This is now indicated in the legend. Furthermore, following the reviewer's suggestions, we also assessed the expression of these genes by RT-qPCR (new Suppl. Fig. 3b).

53. Suppl. Fig. S4a: define arrow positions in legend text below.

Response: We now indicate in the corresponding legend that the "arrowheads mark the end of the meristem".

54. Suppl. Fig. S6: Replace FeCN (2x) by correct chemical formula, insert spaces as appropriate in legend text and figure. Please specify concentration of ascorbate and GSH used in panel d (e.g. in legend text below or refer to panel c) – 10 mM?

Response: FeCN was replaced with the chemical formula $[\text{Fe}(\text{CN})_6]^{3-}$. The values shown in panel d refer to the current recordings shown in c. The information was added in the legend.

55. Suppl. Fig. S7: FeCN as above, show SD instead of s.e.m., "cyt" is cut off at the bottom in panel b. Legend d: "Acscyt + 10 mM)" very hard to understand, best to add "Asc" again. Can the authors please explain more precisely the meaning of the two vertical arrows in a, b and e, respectively.

Response: Thank you. We now show SD in Suppl. Fig.S7; "cyt" is now visible. We corrected a mistake in figure legend (there was a wrong Asc). To better explain the meaning of the two arrows, we also added the following sentence in the legend of Suppl. Fig. S7: "the left and right arrows indicate the addition and removal of FeCN, respectively".

56. Suppl. Fig. 8: This is a nice figure showing that the H211L/H278L mutant variant is produced and plasma membrane-localized in oocytes, as well as why this mutant was made and not others.

Response: Thank you. To complement the localization in oocytes, we now also shown the localization of the HYP1^{H211L/H278L}:GFP *in planta* in Suppl. Fig. 8g.

57. Suppl. Fig. S9: Please explain precisely the significance of the two vertical arrows in a. Please show s.d. instead of s.e.m. in b.

Response: We added the following sentence in the legend of Suppl. Fig. S9: "the left and right arrows indicate the addition and removal of the ferric chelate, respectively". Now, we use SD in Suppl. Fig. 9.

58. Suppl. Fig. 10: This is a nice figure because it shows the details around the quiescent center.

Response: Thank you for the comment.

59. Suppl. Fig 11: Statistics of a need modification. Does RhoNox-1 move into cells quantitatively – could a useful control be introduced? Please specify in which form, at which concentration and how long Fe(II) was fed to roots and how its oxidation was prevented.

Response: We performed new experiments with RhoNox-1 to demonstrate that it reacts specifically with Fe(II), and provide quantification of RhoNox-derived signals in the RAM of Col-0, *hyp1* and *35S::HYP1* plants grown under +P and -P (Suppl. Fig. 13c-e). RhoNox-1 was assayed by incubating root tips with freshly prepared buffer solution and the dye for 10 min. Roots were imaged immediately with a confocal microscope.

60. Please refer to accompanying manuscript in the manuscript text (just like the accompanying manuscript does).

Response: We now mention the accompanying manuscript of Clúa et al. in lines 167-168 and 498-500.

Reviewer #4 (Remarks to the Author):

The manuscript by Maniero et al. identified a CYBDOM protein named as HYP1 in Arabidopsis, and they found that HYP1 is required for the low P responses of root growth. They found that HYP1 functions as a ferric reductase and mediates electron transfer through plasma membrane in root cells. The story is quite interesting, and this research provides insights into the mechanisms of low P responses. I feel that the identification of this novel gene will facilitate the research in this field. Please see the following for my comments, and I wish the comments are helpful for the authors to improve the manuscript.

Response: We thank the reviewer for the encouraging comments.

1. The electrophysiological data show the larger currents recorded at -20 mM in the oocytes injected with HYP1 cRNA relative to the oocytes injected with water, and the currents become bigger when more ascorbate was injected into the oocytes. My first concern is you added FeCN in the bath solution, and injected ascorbate into the oocytes. That means ascorbate and FeCN have no chance to mix together and bind to each other. Then how they form a complete electron transfer chain, and how electron transferring happens?

Response: We wish to underline that currents were recorded only in HYP1-expressing oocytes and only upon application of external FeCN, as efficient electron acceptor. No FeCN-induced currents were detected in water-injected oocytes. Other electron acceptors induced significant currents, such as ferric-chelates Fe(III)-NTA and Fe(III)-EDTA (Supp. Fig. 9). Very interestingly, our data suggest that ferric malate and Cu(II) salts are possible physiological electron acceptors. Cytosolic ascorbate is the specific electron donor, in line with the HYP1 molecular structure, which has a specific binding site for Asc facing the cytosolic side (updated Fig. 3b). We found that this binding site has an apparent affinity for cytosolic ascorbate of 4 mM (Suppl. Fig. 6). When FeCN was added in the external solution, ascorbate released an electron to the HYP1 heme group in the cytosolic side. The electron was transferred via the other two hemes of HYP1 and reduced the acceptor present at the external side of the membrane. Mutation of the two histidines coordinating the second heme group of HYP1 (H211L/H278L)

completely abolished the FeCN-induced currents (Fig. 2d-e and Suppl. Fig. 8a-f), in agreement with the above mechanism of electron transfer. Therefore, a net negative charge moved across the oocyte membrane, which could be recorded by the voltage-clamp system. Thus, according to our results, a controlled and coordinated redox reaction took place in two cellular compartments: oxidation of ascorbate at the cytosolic site induced reduction of an external acceptor (FeCN/ferric-chelates/Cu(II) salts) in the external solution.

If a small holding potential is enough to drive electron efflux, then how to explain the ascorbate-specificity of the currents. Supp Figure 6 showed another donor, GSH, is not capable of generating currents. What if you add FeCN and ascorbate together in the bath solution, could you record any currents, for example outward cation currents/inward anion currents?

Response: Electrons move from a negative to a positive redox potential. The redox potential inside the oocyte is generated by the MDHA/ascorbate couple and is supposed to be strongly positive due to the low MDHA concentration (see Gradogna et al. *New Phytologist* 2023, doi: 10.1111/nph.18823). The redox potential outside is infinitely positive because the oocyte is subjected to continuous bath perfusion with the acceptor entirely in its oxidised form. The electron currents are therefore always negative regardless of the value of the membrane potential. In most experiments we have chosen a holding potential of -20 mV because the background currents are small and appear to be very stable. The specificity of ascorbate is related to its binding site present in HYP1. If ascorbate and ferricyanide are placed in the same compartment, ascorbate would completely reduce FeCN, which is an undesirable condition due to its instability and difficulty of control.

Another concern is the possibility of Ca²⁺-activated anion currents. The bath solution contains 1 mM CaCl₂, and the currents were recorded at -20 mV. It is known that the influx of Ca²⁺ activates large anion efflux in oocytes when the holding potential applied to the plasma membrane is negative. In my experience, about 80% of the whole-oocyte currents are resulted from the efflux of anion in case Ca²⁺ influx really happens, and only a little portion of the currents is derived from Ca²⁺ and others. Some people add some DIDS, an anion channel blocker in the bath solution, to inhibit the anion efflux, then they claim the currents they recorded are Ca²⁺ currents. That is not correct because DIDS is a quite weak blocker and cannot abolish Ca²⁺-activated anion channel currents. In this research it is possible that the currents could be a mixture of electron efflux from ascorbate to FeCN and some anion efflux activated by Ca²⁺. So I think the authors need to exclude the possibility of anion efflux activated by Ca²⁺ influx. To address this, I guess the authors could try the followings. One is to remove Ca²⁺ at all from the bath solution, and to replace all chloride in both bath and pipette solution with a membrane-impermeable anion. You could use salt bridge containing Cl⁻ to form a closed circle. I think gluconate could be a good choice. If you record similar currents in the oocytes expressing HYP1, and the currents are dependent on ascorbate and FeCN, then the data will be more convincing. Alternatively, the authors also could use a different cell system, such as HEK293. This is because Ca²⁺ activates anion efflux only in oocytes, and this does not happen in many other mammalian cell lines. I guess you need do patch clamping in these other cell lines rather than voltage clamping.

Response: There are several experimental evidences indicating that background currents are negligible and in particular there is no significant contribution of endogenous calcium-activated anion channels: 1) without FeCN in the external solution, at a holding voltage of -20 mV, background currents are small both in water- and HYP1- injected currents; 2) in water-injected oocytes, the application in the external solution of FeCN or of other electron acceptors used in this work, did not induce any currents in any tested experimental condition;

3) the substitution of chloride with the large impermeant anion MES, as suggested by the referee, did not significantly modify the FeCN-induced currents in HYP1-injected oocytes, as shown in Suppl. Fig. 7e,f;

4) the removal of calcium (as suggested by the referee) in the external solution did not modify the FeCN-induced currents in HYP1-injected oocytes (see Suppl. Fig. 7e,f); this was not clearly stated in the previous version of the manuscript, we apologise for this. In the amended version, we modified the following sentence in the Methods section (lines 683-686): “the standard bath solution was modified by setting the pH to 5.5 by substituting 10 mM HEPES with an equimolar concentration of MES, or by replacing NaCl, KCl, CaCl₂ and MgCl₂ with 200 mM MES and 20 mM BTP at pH 5.5 (BTP-MES solution).” We also modified the legend of Suppl. Fig. 7f: “...in oocytes injected with HYP1 cRNA upon exposure to 1 mM FeCN in control solution (red trace) or in modified solutions with pH set to 7.5 (black trace) and with NaCl, KCl, CaCl₂ and MgCl₂ replaced by BTP-MES (blue trace)”.

2. I am also curious what the concentration of ascorbate in root cells, whether the level of ascorbate in root cells is high enough to generate large current, and whether Pi deficiency induces the accumulation of ascorbate in root cells.

Response: Previous studies have recorded up to 10 mM of ascorbate in the cytosol of plant cells (reviewed in e.g., Smirnov 2018 DOI: 10.1016/j.freeradbiomed.2018.03.033). Although we do not have the expertise and equipment to determine cytosolic concentrations to assess ascorbate accumulation in root cells expressing *HYP1*, it has been previously reported that the expression of ascorbate biosynthesis genes, such as *VTC2* and *VTC4* are up-regulated in Arabidopsis roots in response to P deficiency (Mora-Macías et al., 2017). We confirmed the up-regulation of these two genes in P-deficient roots (Fig. 1c). Thus, it is indeed likely that P deficiency increases ascorbate concentration in root cells. We now discuss this point in lines 468-478.

REVIEWERS' COMMENTS

Reviewer #1 (Remarks to the Author):

This manuscript has been substantially improved, and I have no further comment.

Reviewer #2 (Remarks to the Author):

I am satisfied by this revised manuscript and have only a comment and a minor point to make.

Fig.6l (model):

Based solely on the authors' model in Figure 6l, one would think that both the *lpr1*, *lpr2* double mutant and the *35S::HYP1* line should have a short root under -Pi condition, which is the opposite of what is observed.

Couldn't we amend this model by emphasising that there is a competition between two chemical reaction routes: the reactions catalysing ROS (starting by a photo-Fenton reduction of the malate-Fe³⁺ complex), and the reduction of the malate-Fe³⁺ complex by HYP1?

Line 378: « HYP1-depedent »

Reviewer #3 (Remarks to the Author):

All in all, the authors have sufficiently addressed all my comments. This is excellent impressive work and should be published. In the publication, however, the authors should ensure that the full gene lists with transcript levels underlying Figs. 1a and 1b as well as the new data in Figs. 5 h and i are provided as Suppl. Datasets in the form of tab-delimited text or excel files.

Reviewer #4 (Remarks to the Author):

It seems that my concerns were addressed. The authors added new data, more detailed description and discussion in the revised MS in response to my questions. So, I don't have further questions. Good luck to the authors!

REVIEWERS' COMMENTS

Authors' comment: We thank the reviewers for appreciating our work and for providing constructive criticisms and suggestions.

Reviewer #1 (Remarks to the Author):

This manuscript has been substantially improved, and I have no further comment.

Response: We thank the reviewer for the positive feedback.

Reviewer #2 (Remarks to the Author):

I am satisfied by this revised manuscript and have only a comment and a minor point to make.

Response: We thank the reviewer for the positive feedback.

Fig.6l (model):

Based solely on the authors' model in Figure 6l, one would think that both the *lpr1, lpr2* double mutant and the *35S::HYP1* line should have a short root under -Pi condition, which is the opposite of what is observed.

Couldn't we amend this model by emphasizing that there is a competition between two chemical reaction routes: the reactions catalysing ROS (starting by a photo-Fenton reduction of the malate-Fe³⁺ complex), and the reduction of the malate-Fe³⁺ complex by HYP1?

Response: Following the reviewer's suggestion, we amended the model (Fig. 8c, former Fig. 6l) and indicate that the redox cycling in the apoplast can be driven by photo-Fenton chemistry or by apoplastically located chemical reductants (e.g., ascorbate). We now write in the legend: "Depletion of soluble Fe can prevent its participation in apoplastic redox cycling reactions potentially involving Fe³⁺ reduction by blue light or apoplastically-located chemical reductants and Fe²⁺ re-oxidation by the cell-wall-localized multicopper ferroxidases LPR1 and LPR2. Uncontrolled apoplastic redox cycling induces ROS-dependent aberrant callose deposition in the root apical meristem, causing meristem exhaustion. A putative interplay of HYP1 and LPRs remains elusive."

Line 378: « HYP1-depedent »

Response: Thank you for spotting this typo, which we now corrected.

Reviewer #3 (Remarks to the Author):

All in all, the authors have sufficiently addressed all my comments. This is excellent impressive work and should be published. In the publication, however, the authors should ensure that the full gene lists with transcript levels underlying Figs. 1a and 1b as well as the new data in Figs. 5 h and i are provided as Suppl. Datasets in the form of tab-delimited text or excel files.

Response: Thank you for the positive feedback. We now provide the requested information as Excel files in the new Supplementary Data 1-5.

Reviewer #4 (Remarks to the Author):

It seems that my concerns were addressed. The authors added new data, more detailed description

and discussion in the revised MS in response to my questions. So, I don't have further questions.

Good luck to the authors!

Response: Thank you for the positive feedback.